# Diffusion-based Neural Network Weights Generation

**Soro Bedionita**[1*]   **Bruno Andreis**[1*]   **Hayeon Lee**[1]   **Wonyong Jeong**[3]   **Song Chong**[1]
**Frank Hutter**[2]   **Sung Ju Hwang**[1,3]
[1]KAIST   [2]University of Freiburg   [3]DeepAuto.ai

## Abstract

Transfer learning is a cornerstone of modern deep learning, yet it remains constrained by challenges in model selection and the overhead of extensive model storage. In this work, we present Diffusion-based Neural Network Weights Generation, D2NWG, a novel framework that leverages diffusion processes to synthesize task-specific network weights. By modeling the distribution of weights from a diverse ensemble of pretrained models and conditioning the generation process on dataset characteristics, task descriptions, and architectural specifications, D2NWG circumvents the need for storing and searching through massive model repositories. We evaluate D2NWG across multiple experimental settings. On in-distribution tasks, our framework achieves performance that is on par with or superior to conventional pretrained models, while also serving as an effective initialization strategy for novel domains, resulting in faster convergence and a 6% improvement in few-shot learning scenarios. Extensive ablation studies further indicate that our approach scales robustly with increased diversity and volume of pretrained models. Moreover, D2NWG demonstrates significant promise for large language model applications. In evaluations on the OpenLM leaderboard, our method improved LLaMA-3-2-1B-Instruct performance by 3% on challenging mathematical reasoning tasks, with a consistent gain of 0.36% across a range of benchmarks. These findings establish D2NWG as a versatile and powerful framework for neural network weight generation, offering a scalable solution to the limitations of traditional transfer learning.

## 1 Introduction

Diffusion-based generative models have emerged as a breakthrough technology in artificial intelligence, achieving state-of-the-art performance in generating complex, high-dimensional data across domains including natural language, audio, images, and video (Gozalo-Brizuela & Garrido-Merchán, 2023). The success of these models stems from their principled approach to data generation through iterative denoising (Ho et al., 2020b; Rombach et al., 2022; Peebles & Xie, 2023; Gao et al., 2023), which has proven remarkably effective for modeling complex probability distributions and generating high-quality samples (Yang et al., 2024). Despite significant advancements in generative modeling, a fundamental challenge remains largely unexplored: can diffusion models be leveraged to directly generate neural network weights from pretrained models? Successfully addressing this question could reshape core machine learning paradigms, particularly in transfer learning and AutoML (Hutter et al., 2019; Doke & Gaikwad, 2021). By synthesizing task-specific network parameters on demand, we could circumvent the computational inefficiencies of traditional fine-tuning while enhancing adaptability to novel tasks.

Recent efforts in weight generation, such as generative hyper-representation learning (Schürholt et al., 2022a), have only begun to tackle this challenge. Current approaches, including Neural Network Diffusion (Wang et al., 2024) and kernel density estimation based methods (Sch"urholt et al., 2024),

---

*Equal Contribution.  Correspondence to: Soro Bedionita <sorobedio@kaist.ac.kr>, Bruno Andreis <andries@kaist.ac.kr>, Hayeon Lee <hayeon926@kaist.ac.kr>, Wonyong Jeong <wyjeong@kaist.ac.kr>, Song Chong <songchong@kaist.ac.kr>, Frank Hutter <fh@cs.unifreiburg.de>, Sung Ju Hwang <sjhwang82@kaist.ac.kr>

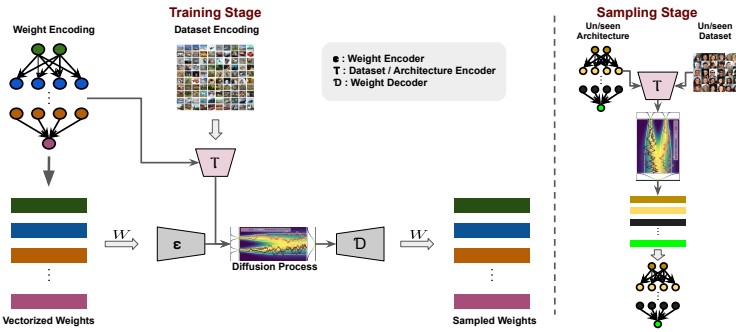

Figure 1: **Stage 1:** VAE Encoder and Decoder training process. **Stage 2**: dataset encoder training stage . **Stage 3:** Dataset conditioned diffusion process.

exhibit significant limitations in both scalability and generalization. These methods are primarily constrained to small architectures and focus on unconditional weight generation within predefined distributions, neglecting the essential problem of generating task-specific weights for novel scenarios from diverse pretrained model distributions. While these techniques yield performance gains on familiar tasks, their inability to generalize effectively to unseen domains diminishes their practical applicability in model selection and transfer learning. Meta-learning approaches (Nava et al., 2023; Zhang et al., 2024) have made strides in addressing weight generation for visual and few-shot learning tasks. However, these methods remain limited in their ability to produce dataset-conditioned solutions, as they often rely on learned priors that do not fully exploit the richness of pretrained model distributions. Consequently, the development of a principled approach to synthesizing neural network weights in a dataset-aware, scalable manner remains an open research question with profound implications for efficient model adaptation and deployment.

To address these challenges, in this work, we introduce Diffusion-based Neural Network Weight Generation (D2NWG ), a novel approach that leverages latent diffusion to generate neural network parameters by learning from diverse pretrained model weights. Our method re-formulates the latent diffusion paradigm for weight generation by incorporating robust dataset and task conditioning capabilities. D2NWG learns the distribution of weights from diverse architectures and pretraining datasets conditioned on dataset or task description enabling dataset/task-specific weights generation during infeence while the generated weights maintain performance comparable to individual pretrained models on in-distribution tasks and fast converge when fine-tuned on unseen dataset/task. Additionally, our analysis reveals that the diversity and size of the pretrained model training set strongly correlates with improved generalization to unseen datasets and tasks. Our empirical evaluation validates the contribution of D2NWG as follows:

- The generated weights match or outperform traditional pretrained models on seen tasks while enabling faster, better learning on new tasks through superior weight initialization.

- D2NWG outperforms recent meta-learning Zhang et al. (2024) approach on few-shot setting as well as recent weights generation methods Schürholt et al. (2022b); Sch"urholt et al. (2024).

- D2NWG enables learning from a distribution of diverse pre-trained models, each trained on different datasets while matching individual pretrained model performance.

- D2NWG scales to small and large datasets, generating weights for architectures with over 400 million parameters including GPT2-Small.

- We demonstrate its effectiveness in improving LLM performance by generating task-specific weights from a single pretrained model and our sampled weights based on LLAMA3-.1-8B and LLAMA3-.2-1B models ranked among the top 2 performing models on the open lm-leaderboard[1]

---

[1] https://huggingface.co/spaces/open-llm-leaderboard/open_llm_leaderboard

## 2 RELATED WORK

**Neural Network Parameters Prediction**: As neural networks expand across domains, transfer learning through pretrained weights has become crucial. While hypernetworks have emerged as a promising approach for weight prediction (Chauhan et al., 2023; Ratzlaff & Fuxin, 2020; Denil et al., 2013; Ha et al., 2016), subsequent Graph Hypernetworks (GHN) methods leverage model architecture graphs to generate weights (Zhang et al., 2019; Knyazev et al., 2021; Zhmoginov et al., 2022; Knyazev et al., 2023). Though recent transformer-based approaches treat weight generation as an autoregressive process (Zhmoginov et al., 2022), these methods remain constrained by their single-task focus, limiting their transfer learning capabilities. Similar to GHNs diffusion models has been used to generate weights in meta learning setting Nava et al. (2023); Zhang et al. (2024). However, the generated parameters are not task-specific and the generator is limited to the classier head.

**Parameters Generation from Pretrained Distribution**: Parameter generation from pretrained distributions has emerged as a promising research direction due to its practical applications. However, existing approaches (Schürholt et al., 2021; Schürholt et al., 2022a; Peebles et al., 2022; Sch"urholt et al., 2024) remain constrained by their focus on single-dataset parameter learning, leaving the broader potential of cross-domain applications largely unexplored.

**Applications of Parameter Generation in LLMs:** Despite minimal exploration around learning from pretrained weight distributions, we show in this work that our approach generates diverse task-specific weights for LLMs (Minaee et al., 2024; Zhao et al., 2023; Dubey et al., 2024). By generating specialized LoRA modules (Tang et al., 2024; Zhao et al., 2024; Gong et al., 2024), we can enhance model flexibility and transfer learning while reducing computational costs.

## 3 APPROACH

### 3.1 PRELIMINARY

Let's consider a collection of neural network models $\{\mathcal{A}_i\}_{i=1}^M$, each pretrained on one of $M$ distinct datasets $\{\mathfrak{D}_1, \mathfrak{D}_2, \ldots, \mathfrak{D}_M\}$. Our primary objective is to characterize and learn the underlying distribution $p(W)$ of the pretrained model weights $W$ across this ensemble. Ultimately, we aim to develop a method for conditional sampling of weights $p(W_T|\mathfrak{D}_T)$ that are optimized for any target dataset or task $\mathfrak{D}_T(x, y)$, regardless of whether it appeared in the training distribution. These sampled weights should either achieve strong performance on $\mathfrak{D}_T$ immediately( or require minimal fine-tuning compared to random initialization). The intuition is that there is a direct relationship between a pretrained network weights and the dataset it was trained on (see Appendix A.1 for a formal argument). We argue that this relationship constrains the high-dimensional weight space $\mathcal{W} \in \mathbb{R}^n$ to a lower-dimensional manifold $\mathcal{M} \subset \mathcal{W}$ with dimension $k \ll n$. This hypothesis is supported by the Lottery Ticket literature (Frankle & Carbin, 2019; Liu et al., 2024), which shows that sparse subnetworks can match full network performance: $\mathcal{L}(\theta; \mathfrak{D}) \approx \mathcal{L}(\theta \odot m; \mathfrak{D})$, where $m \in \{0, 1\}^n$ is a sparse mask. By Whitney's Embedding Theorem (Whitney, 1936), $\mathcal{M}$ can be smoothly embedded in $\mathbb{R}^{2k+1}$ via a diffeomorphism $\phi : \mathcal{M} \to \mathcal{Z}$, where $\mathcal{Z}$ represents a latent space. We approximate this embedding using a variational autoencoder(VAE). Given the differentiability of $\mathcal{Z}$, we can employ latent diffusion to model the distribution of pretrained weights. This enables our proposed D2WNG framework to not only preserve individual model performance but also generalize to unseen datasets as we incorporate more pretrained models, leveraging the smoothness and interpolation properties of the latent space. Later, we investigate some possible way to improve LLMs without fine-tuning through through sampling in latent space with D2NWG. In this paper, we use the terms *seen dataset/task* and *unseen dataset/task* to refer to datasets or tasks that are present in or absent from the training set, respectively and Zero-shot means no finetuning is performed on sampled weights and are directly evaluated.

### 3.2 WEIGHT ENCODING

Let $\{\mathcal{A}_i\}_{i=1}^N$ be a set of pretrained models. For each $\mathcal{A}_i$, we flatten its weights to $W_i \in \mathbb{R}^{d_i}$ where $d_i$ is its parameter count and we define $d_{\max} = \max_i d_i$. We zero-pad each $W_i$ to obtain $\hat{W}_i \in \mathbb{R}^{d_{\max}}$, giving uniform-length representations (Figure 1) to which we refere to as model-wise vectorization.

This setting is suitable for small models and classifier layer adaptation. On the other hand, Layer-wise vectorization keeps each layer's weights separate rather than concatenating them. Each flattened weight vector $\mathbf{w} \in \mathbb{R}^{mn}$ is zero-padded to match a chosen chunk size multiple, then split into $k$ equal-length subvectors $\bar{\mathbf{w}}_i \in \mathbb{R}^l$ where $l = \lceil mn/k \rceil$. This enables independent layer-wise sampling during inference, where each vectorized layer serves as a separate input for subsequent stages. This setting is suitable for large models.

**Parametrs Encoding:** We then train a Variational Autoencoder (VAE) to encode these vectors and minimizing the following objective function:

$$\mathcal{L} = -\mathbb{E}_{q_\phi(z|w)}\left[\log p_\theta(w|z)\right] + \beta \mathrm{KL}\left[q_\phi(z|x) \,\|\, p(z)\right] \tag{1}$$

where $w$ is the vectorized weight, $z$ is the latent representation, $p_\theta$ and $q_\phi$ the reconstruction and approximate posterior terms respectively, $p(z)$ the prior distribution (a Gaussian prior) and $\beta$ is a fixed hyper parameters that regulates the stochasticity of the VAE. Model-wise and layer-wise vectorized parameters are encoded using the same VAE structure, with the only difference being in the input dimensions. In chunk-wise encoding, the original flattened vector $w$ is recovered by reassembling the decoded latent chunks through concatenation. The reconstructed chunks $\hat{w}_i$ from each layer are concatenated to recover $\hat{w} = \hat{w}_1 \oplus \hat{w}_2 \oplus \cdots \oplus \hat{w}_k$, where $\oplus$ denotes concatenation. And reshaping $\hat{w}$ back into the original form $\hat{W}$ yields a close approximation of the original weight $W$. The quality of reconstruction is assessed by evaluating the reconstructed weights on a designated evaluation dataset or task.

### 3.3 DATASET ENCODING

**Image Dataset Encoding:** We adopt a Set Transformer-based encoder (Lee et al., 2019a) $\mathcal{T}$ to encode the pretraining datasets. This approach effectively handles large, multi-class datasets and has been validated in prior dataset-adaptive methods (Jeong et al., 2021; Lee et al., 2021). Figure 5 in the appendix provides an architectural overview of the dataset encoder. Given a dataset with $C$ classes denoted by $\mathfrak{D} = \{(x_i, y_i)\}_{i=1}^C$, where $x_i$, and $y_i$ denote inputs and labels, we use a pretrained clip image encoder to extract the images features and group the data into subsets $s_i$ by class, forming $\mathcal{S} = \{s_i\}_{i=1}^C$ with $s_i \in \mathbb{R}^{C \times K_i \times d_{feat}}$. Here $K_i$ is the number of images belonging to class $i$, and $d_{feat}$ the features dimension. Each subset is transformed into embeddings $z_{s_i} \in \mathbb{R}^{1 \times d}$ using a transformation $\mathcal{T}$, and these embeddings are aggregated into $\tilde{s}_i \in \mathbb{R}^{C \times d}$. Another transformation $\mathcal{T}$ produces the final dataset encoding $z_\mathfrak{D} \in \mathbb{R}^d$, represented as: $z_\mathfrak{D} = \mathcal{T} \circ \mathcal{T}(\mathcal{S})$ This encoding is invariant to the number of classes and dataset size, and it operates without utilizing labels. We train the dataset encoder $\mathcal{T}$ using a contrastive loss to align dataset embeddings $z_{\mathfrak{D}_i}$ with pretrained weight embeddings $z_i$, following the CLIP-style approach introduced in HyperCLIP (Nava et al., 2023). This alignment ensures training stability and computational efficiency during diffusion optimization. Specifically, we optimize the following objective:

$$\mathcal{L}_{\mathrm{CLIP}} = -\log \frac{\exp(z_i \cdot z_{\mathfrak{D}_i}/\tau)}{\sum_{k=1}^N \exp(z_i \cdot z_{\mathfrak{D}_k}/\tau)}, \tag{2}$$

where $z_{\mathfrak{D}_i}$ is the dataset embedding for $\mathfrak{D}_i$, and $z_i$ is the corresponding VAE-encoded weight embedding (Section 3.2). This alignment enables efficient probing and integration into downstream tasks.

**Language Task Encoding:** To enable task-description-based parameter generation for NLP tasks, we first encode each task description using Llama-3-8B-Instruct. The output from the last hidden layer is used as the task's dataset embedding. These embeddings are then directly incorporated into the diffusion process during both training and inference.

### 3.4 DATASET-CONDITIONED PARAMETERS GENERATION

At this stage, we have access to a pretrained VAE for encoding neural network weights and a pretrained Set Transformer module to encode entire datasets. The next stage involves defining a model to generate latent representations of weights conditioned on the dataset embeddings. We achieve this by using diffusion probabilistic models (DDPM) (Ho et al., 2020a; Rombach et al., 2021) trained on the latent representation of the pretrained weights..

**Forward Process:** Given a weight embedding $z$, obtained from the encoder of the pretrained VAE, the forward diffusion process involves successive Gaussian noise perturbations of $z$ over $T$ time steps. At time step $t$,

$$p(z_t|z_{t-1}) = \mathcal{N}(z_t; \mu_t = \sqrt{1 - \beta_t} z_{t-1}, \beta_t I) \tag{3}$$

where $\beta_t \in (0, 1)$ is the noise variance and $p(z_{1:T}|z_0) = \prod_{i=1}^{T} p(z_t|z_{t-1})$.

**Reverse Process:** As in most DDPM approaches the reverse process is approximated by a neural network such that:

$$p_\theta(z_{t-1}|z_t) = \mathcal{N}(z_{t-1}; \mu_\theta(z_t, t), \Sigma_\theta(z_t, t)), \tag{4}$$

where $\mu_\theta$ and $\Sigma_\theta$ are neural networks.

**Dataset-Conditioned Training:** The diffusion model is trained on the VAE embeddings $z$, conditioned on the dataset embeddings concatenated with the latent representations of the weights. To leverage existing architectures, we designed the VAE to generate latent representations that are compatible with standard latent diffusion models with minimal adjustments, optimizing the latent diffusion objective defined in Eq. 5.

$$\mathcal{L}_{LDM} = \mathbb{E}_{z, \varepsilon \sim \mathcal{N}(0,1), Z_\mathfrak{D}, t} \left[ ||\varepsilon - \varepsilon_\psi(z_t, z_\mathfrak{D}, t)||_2^2 \right], \tag{5}$$

where $\varepsilon_\psi(z_t, z_\mathfrak{D}, t)$ is implemented as a UNet.

**Sampling:** New weights are sampled conditionally through the reverse diffusion process as follows:

$$z_t = \frac{1}{\sqrt{a_t}} (z_t - \frac{\beta_t}{\sqrt{1 - \tilde{a}_t}} \varepsilon_\psi(z_t, z_\mathfrak{D}, t,)) + \sigma\xi, \tag{6}$$

where $\xi \sim \mathcal{N}(0, I)$ and, $\sigma_t$ a chosen value. After sampling a latent representation ($\bar{z}$ for a given dataset $\mathfrak{D}_i$). The pretrained VAE decoder is used to transform these latents into a weight vector $\bar{w} = \mathcal{D}(\bar{z})$, which is then used to initialize the target network as shown in Figure 1.

### 3.5 EXPLORING THE OPTIMAL PARAMETERS SPACE OF LLMS

In this section, we extend our method to enhance pretrained LLM performance without fine-tuning by recasting D2NWG as a layer-conditioned parameter generation approach. The key challenge is managing the vast parameter space of LLMs. Drawing from (Hartford et al., 2024), we use the Marchenko-Pastur distribution to identify crucial layers for improving the performance based on weights spectrum. We calculate a signal-to-noise ratio (SNR) to distinguish significant weights from noise as: $SNR = \frac{\sum_{k | |\sigma_k| \geq \varepsilon} \sigma_k}{\sum_{n | |\sigma_n| < \varepsilon} \sigma_n}$, where eigenvalues $\sigma_n$ above threshold $\varepsilon$ represent meaningful signals, while those below are considered noise. For this task, we employ layer-wise chunking to manage large layers. We provide more detailed in Appendix A.3 and A. Additionally, we present a sequential optimal space exploration algorithm, detailed in Algorithm 1.

## 4 EXPERIMENTS

We evaluate our method both with and without finetuning on Few-Shot Learning, Zero-Shot Learning (no fine-tuning), and Model Retrieval tasks. All experiments use a single Titan RTX GPU except experiment with LLMs which used a single A100 GPU. Detailed ablation studies are provided in the Appendix C.

### 4.1 WEIGHT GENERATION WITHOUT FINETUNING ON UNSEEN TASK

We present a set of results where the generated weights are evaluated directly without finetuning for few-shot learning and transferring to unseen Tasks.

#### 4.1.1 WEIGHTS GENERATION FOR FEW-SHOT LEARNING

**Task:** We aim to show that learning the distribution of models pretrained independently on a large set of dataset can enable sampling weights that compete with meta-learning techniques in multi-task few-shot learning, without requiring fine-tuning.

Table 1: Few-Shot Learning. ALL implies generation of the entire parameters and CH denotes generation of classification head only.

| Method | Adaptation | Backbone | mini-ImageNet | | tiered-ImageNet | |
|---|---|---|---|---|---|---|
| | | | 5-way 1-shot | 5-way 5-shot | 5-way 1-shot | 5-way 5-shot |
| iMAML (Rajeswaran et al., 2019) | ALL | Conv4 | $49.30 \pm 1.88\%$ | $59.77 \pm 0.73\%$ | $38.54 \pm 1.37\%$ | $60.24 \pm 0.76\%$ |
| ALFA (Baik et al., 2020) | ALL | Conv4 | $50.58 \pm 0.51\%$ | $69.12 \pm 0.47\%$ | $53.16 \pm 0.49\%$ | $70.54 \pm 0.46\%$ |
| COMLN (Deleu et al., 2022) | CH | Conv4 | $53.01 \pm 0.62\%$ | $70.54 \pm 0.54\%$ | $54.30 \pm 0.69\%$ | $71.35 \pm 0.57\%$ |
| MetaQDA (Zhang et al., 2021) | CH | Conv4 | $56.41 \pm 0.80\%$ | $72.64 \pm 0.62\%$ | $58.11 \pm 0.48\%$ | $74.28 \pm 0.73\%$ |
| MetaDiff (Zhang et al., 2024) | CH | Conv4 | $55.06 \pm 0.81\%$ | $73.18 \pm 0.64\%$ | $57.77 \pm 0.90\%$ | $75.46 \pm 0.69\%$ |
| D2NWG(Ours) | CH | Conv4 | $\mathbf{61.13 \pm 8.50\%}$ | $\mathbf{76.94 \pm 6.04\%}$ | $\mathbf{65.33 \pm 6.50\%}$ | $\mathbf{80.05 \pm 8.25\%}$ |
| ALFA (Baik et al., 2020) | ALL | ResNet12 | $59.74 \pm 0.49\%$ | $77.96 \pm 0.41\%$ | $64.62 \pm 0.49\%$ | $82.48 \pm 0.38\%$ |
| MetaOptNet (Lee et al., 2019b) | CH | ResNet12 | $62.64 \pm 0.61\%$ | $78.63 \pm 0.46\%$ | $65.99 \pm 0.72\%$ | $81.56 \pm 0.53\%$ |
| LEO (Rusu et al., 2019) | CH | WRN-28-10 | $61.76 \pm 0.08\%$ | $77.59 \pm 0.12\%$ | $66.33 \pm 0.05\%$ | $81.44 \pm 0.09\%$ |
| Classifier (Chen et al., 2021) | CH | ResNet12 | $61.22 \pm 0.84\%$ | $78.72 \pm 0.60\%$ | $69.71 \pm 0.88\%$ | $83.87 \pm 0.64\%$ |
| MetaQDA (Zhang et al., 2021) | CH | ResNet18 | $65.12 \pm 0.66\%$ | $80.98 \pm 0.75\%$ | $69.97 \pm 0.52\%$ | $85.51 \pm 0.58\%$ |
| MetaDiff (Zhang et al., 2024) | CH | ResNet12 | $64.99 \pm 0.77\%$ | $81.21 \pm 0.56\%$ | $72.33 \pm 0.92\%$ | $86.31 \pm 0.62\%$ |
| D2NWG(Ours) | CH | ResNet12 | $\mathbf{69.55 \pm 3.77\%}$ | $\mathbf{83.51 \pm 6.21\%}$ | $\mathbf{81.15 \pm 9.70\%}$ | $\mathbf{90.04 \pm 6.10\%}$ |

Table 2: Zero-Shot Transfer Learning. We evaluate on two backbones: Tiny Swin Transformer and ResNet18.

| Model | CIFAR-10 | STL-10 | Aircraft | Pets | CIFAR-100 |
|---|---|---|---|---|---|
| Swin | 7.38 | 8.43 | 5.01 | 2.63 | 1.35 |
| GHN2 (Knyazev et al., 2021) | 48.20 | – | – | – | 12.7 |
| GHN3 (Knyazev et al., 2023) | 51.8 | – | – | – | 11.9 |
| D2NWG(Ours) | $\mathbf{53.12 \pm 0.25}$ | $\mathbf{60.42 \pm 0.14}$ | $\mathbf{24.57 \pm 3.16}$ | $\mathbf{26.47 \pm 1.90}$ | $\mathbf{30.44 \pm 0.15}$ |
| ResNet18 | 10.88 | 6.78 | 3.75 | 2.39 | 1.38 |
| GHN2 (Knyazev et al., 2021) | 19.52 | 13.04 | – | – | – |
| D2NWG | $33.03 \pm 0.04$ | $50.42 \pm 0.13$ | $17.60 \pm 2.13$ | $17.29 \pm 0.13$ | $13.71 \pm 0.63$ |
| D2NWG_CLIP(Ours) | $\mathbf{60.42 \pm 0.75}$ | $\mathbf{82.42 \pm 0.04}$ | $\mathbf{27.70 \pm 3.24}$ | $\mathbf{32.17 \pm 6.30}$ | $\mathbf{51.50 \pm 0.25}$ |

**Dataset:** We utilize the *mini*-ImageNet and *tiered*-ImageNet datasets for this task. For the architectures, we use a four-layer ConvNet and a ResNet12 backbone provided by Chen et al. (2021). We generate the pretrained weights by linear probing a classifier head on each of the 50,000 subsets for 10 epochs and evaluate the performance on 600 subsets from the unseen test split for 1-shot and 5-shot. Analogously to few shot learning, we choose the number of images per class for conditioning to be the same as the support set, while the number of images per class in the query set is fixed to 15 for all methods and 600 tasks are used for testing.

**Baselines:** We benchmark against iMAML (Rajeswaran et al., 2019), ALFA (Baik et al., 2020), COMNL (Deleu et al., 2022), MetaQDA (Zhang et al., 2021), MetaDiff (Zhang et al., 2024), MetaOptNet (Lee et al., 2019b) and a classifier baseline introduced in Chen et al. (2021).

**Results:** Table 1 shows that our approach consistently improves performance on all tasks while utilizing the same backbone as other methods. With the Conv4 backbone, we achieve approximately 6% performance improvement in 1-shot learning and 3 to 4% on 5-shot learning on mini-ImageNet. On Tiered-ImageNet, we achieve more than 8% performance improvement on 1-shot and 5 to 6% average improvement on 5-shots. For the ResNet12 backbone we achieve 4 to 9% performance improvement. These results demonstrate the effectiveness of our method against the existing meta-learning methods.

For evaluation, we perform 50 weight sampling iterations per subset and report the average of the top 3 accuracies. We explore both 1-shot and 5-shot settings, using one and five images per class respectively for conditioning from support set. Our dataset-conditioned weight generation enables efficient task adaptation by producing weights specialized to each dataset's characteristics, achieving superior generalization compared to meta-learning baselines.

### 4.1.2 ZERO-SHOT CLASSIFIER HEAD ADAPTATION

**Task:** We evaluate the performance of the proposed method in adapting the classifier head to unseen datasets. In this experiment, we assess whether our method can conditionally generate the classifier weights, potentially eliminating or significantly speeding up the finetuning process.

Table 3: Model Retrieval via Generative Augmented Weight Sampling

| Domain | Pretrained | D2NWG (Ours) |
|---|---|---|
| Large Animals | $71.11 \pm 11.45$ | $70.33 \pm 12.42$ |
| Small Animals | $54.04 \pm 13.56$ | $54.70 \pm 13.83$ |
| Plants | $63.69 \pm 9.05$ | $71.37 \pm 17.15$ |
| Plant Diseases | $81.69 \pm 19.14$ | $81.98 \pm 19.53$ |
| Microscopy | $55.56 \pm 26.14$ | $55.49 \pm 26.17$ |
| Remote Sensing | $82.20 \pm 7.49$ | $82.68 \pm 8.05$ |
| Vehicles | $57.07 \pm 19.57$ | $58.09 \pm 18.30$ |
| Manufacturing | $84.34 \pm 21.00$ | $84.32 \pm 20.96$ |
| Human Actions | $68.63 \pm 12.45$ | $69.09 \pm 12.73$ |
| OCR | $63.18 \pm 1.75$ | $65.60 \pm 2.00$ |
| Average | $68.32 \pm 13.84$ | $\mathbf{69.47 \pm 14.79}$ |
| Runtime | 6 hours | 40 seconds |

**Dataset:** We partitioned ImageNet-1k into 20k subsets of
10 to 50 classes each with 50 images per class per subset and linear probe a classifier head for 10
epochs using Tiny Swin Transformer (denoted Swin in Table 2), and ResNet18 all pretrained on
ImageNet-1k. For dataset conditioning, we use 5 images per class per subset. The unseen target
datasets are CIFAR-10, STL-10, Aircraft, Pets, and CIFAR-100 . The baseline methods in these
experiments are ResNet18 and Tiny Swin Transformer pretrained on ImageNet-1k.

**Baselines:** We benchmark against the pretrained backbones, and two GHN models (Knyazev et al.,
2021; 2023). Additonally, we provide a powerful variant of our model D2NWG_CLIP where the
dataset encoder encodes the CLIP embedding for each sample in the datasets.

**Results:** Table 2 presents the performance of the sampled weights where it can be seen that the
proposed method achieves better performance compared to the ImageNet pretrained weights and the
GHN family of models. Additionally, the variant of our model that utilizes the CLIP embedding for
dataset encoding significantly improves the performance suggesting that better dataset representation
learning can boost the performance of the generated weights.

### 4.1.3 IN DISTRIBUTION FULL MODELS WEIGHTS GENERATION: MODEL RETRIEVAL

**Task:** We assess the Generative Augmented Retrieval capability of D2NWG , aiming to show that
it can learn the distribution of models pretrained on diverse real-world datasets. This task requires
generation of dataset-conditioned weights that achieve performance comparable to the original
pretrained models and hence provide access to a wide range of pretrained models through efficient
sampling.

**Dataset:** We collected 30 real-world datasets(Ullah et al., 2022), spanning 19 to 706 classes and
organised into 10 domains with 3 datasets per domain, and fine-tuned a MobileNetV3 subnet[2] sampled
from OFA (Cai et al., 2020) for 100 epochs on each dataset. We then learned the distribution of the
combined pretrained models from the last 20 epochs across all datasets.

**Baselines:** For this task, we compare with the original pretrained weights which are finetuned on
each individual dataset. For each dataset, we sample and report the average accuracy of 5 set of
weights sampled with D2NWG .

**Results:** From Table 3 we see that D2NWG conditionally generates high-performing parameters
while enhancing the pretrained model, achieving the best average results across all datasets. This
demonstrates the strong retrieval capability of our method, suggesting it can be used as a neural
network weight retriever in approaches like (Zhao et al., 2024), eliminating the need for pretrained
database. Detailed dataset information is provided in Table 12 and further experiments in Ap-
pendix C.10. Additionally, it is much more efficient to generate weights with our model compared to
pretraining as shown by the runtime in Table 3.

### 4.1.4 TRANSFERRING TO UNSEEN ARCHITECTURE

We investigate weight transferability across ResNet architectures by mod-
eling the distribution of pretrained weights from ResNet32 (trained on
CIFAR-10 and CIFAR-100). We propose a weight initialization method
that leverages pretrained weight distributions from ResNet32 to improve
performance across different ResNet architectures. Our approach sam-
ples and concatenates weights from the source model while preserving
layer-type correspondence, effectively handling varying network dimen-
sions. Experiments on ResNet20/44/56/32 demonstrate consistent im-
provements over random initialization, even without fine-tuning, partic-
ularly on CIFAR-10 classification tasks as shown in Figure 2.

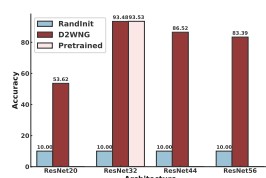

Figure 2: Performance
evaluation with unseen ar-
chitectures on CIFAR-10.

## 4.2 WEIGHTS GENERATION WITH FINE-TUNING

In this section, we evaluate the quality of the generated weights in fine-tuning scenarios to assess
their suitability for transfer learning.

---

[2]https://pytorch.org/hub/pytorch_vision_once_for_all/

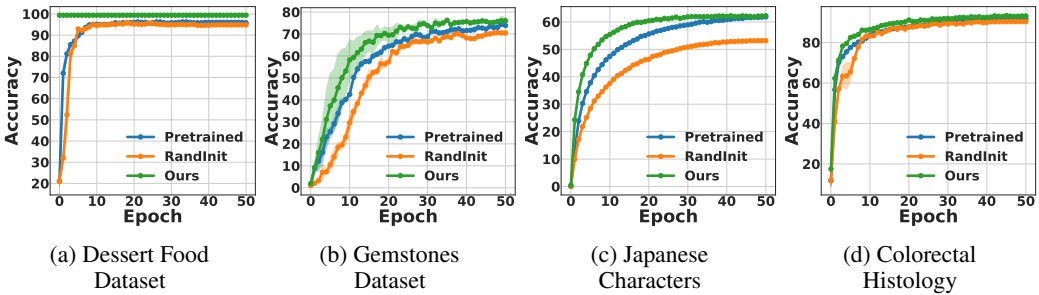

Figure 3: Average accuracy evolution of fine-tuning for 50 epochs with sampled weights for unseen datasets.

### 4.2.1 WEIGHT GENERATION WITH FINE-TUNING ON SEEN TASKS

**Task:** The goal is to assess the behavior of the sampled weights when finetuned on the same dataset and compare convergence speed. This experiment focuses on evaluating whether the sampled weights can be effectively fine-tuned to achieved superior final performance, rather than simply aiming for weights producing high initial accuracy and may not lead to superior performance while fine-tuning.

**Datasets**: We used the modelzoo of Schürholt et al. (2022c) consisting of a ConvNet trained on MNIST, SVHN, CIFAR-10 and STL-10. Our model was trained on the combined pretrained weights from epochs 21 to 25 of all models, consistent with the baseline settings.

Table 4: Finetuning of Generated Weights using the Modelzoo of Schürholt et al. (2022c).

| Epoch | Method | MNIST | SVHN | CIFAR-10 | STL |
|---|---|---|---|---|---|
| 0 | RandomInit | ~10 /% | ~10 /% | ~10 /% | ~10 /% |
| 0 | $S_{KDE30}$ | 68.6±6.7 | 54.5±5.9 | n/a | n/a |
| 0 | $SANE_{KDE30}$ | 84.8±0.8 | 70.7±1.4 | 56.3±0.5 | 39.2±0.8 |
| 0 | $SANE_{SUB}$ | **86.7±0.8** | **72.3±1.6** | 57.9±0.2 | 43.5±1.0 |
| 0 | D2NWG | 80.52±0.82 | 66.6±0.7 | **58.80±0.1** | **44.50±0.1** |
| 1 | RandomInit | 20.6±1.6 | 19.4±0.6 | 37.2±1.4 | 21.3±1.6 |
| 1 | $S_{KDE30}$ | 83.7±1.3 | 69.9±1.6 | n/a | n/a |
| 1 | $SANE_{KDE30}$ | 85.5±0.8 | 71.3±1.4 | 58.2±0.2 | 43.5±0.7 |
| 1 | $SANE_{SUB}$ | 87.5±0.6 | 73.3±1.4 | 59.1±0.3 | 44.3±1.0 |
| 1 | D2NWG | **87.8±0.4** | **73.6±1.3** | **59.2±0.3** | **44.8±0.2** |
| 5 | RandomInit | 36.7±5.2 | 23.5±4.7 | 48.5±1.0 | 31.6±4.2 |
| 5 | $S_{KDE30}$ | 92.4±0.7 | 57.3±12.4 | n/a | n/a |
| 5 | $SANE_{KDE30}$ | 87.5±0.7 | 72.2±1.2 | 58.8±0.4 | 45.2±0.6 |
| 5 | $SANE_{SUB}$ | 89.0±0.4 | 73.6±1.5 | 59.6±0.3 | 45.3±0.9 |
| 5 | D2NWG | **92.5±0.9** | **74.0±0.1** | **60.3±0.1** | **45.4±0.1** |
| 25 | RandomInit | 83.3±2.6 | 66.7±8.5 | 57.2±0.8 | 44.0±1.0 |
| 25 | $S_{KDE30}$ | 93.0±0.7 | 74.2±1.4 | n/a | n/a |
| 25 | $SANE_{KDE30}$ | 92.0±0.3 | 74.7±0.8 | 60.2±0.6 | 48.4±0.5 |
| 25 | $SANE_{SUB}$ | 92.3±0.4 | 75.1±1.0 | 61.2±0.1 | 48.0±0.4 |
| 25 | D2NWG | **96.2±0.3** | **75.7±0.5** | **64.1±1.0** | **48.7±0.5** |
| 50 | RandomInit | 91.1±2.6 | 70.7±8.8 | 61.5±0.7 | 47.4±0.9 |

**Baselines:** We compare against the kernel density estimator approaches from Sch"urholt et al. (2024); Schürholt et al. (2022b), evaluated on the same datasets. Unlike these unconditional methods, we build a model specifically for MNIST and SVHN, and another for CIFAR-10 and STL-10. For each dataset, five sets of weights were sampled to initialize the models, which were fine-tuned for a number of epochs from 0 to 25. We also add RandomInit model trained for 50 epochs and show that our sampled weight finetuned for 25 epochs outperforms this model.

**Results:** As shown in Table 4, D2NWG consistently accelerates convergence across related tasks, surpassing the pretrained model and outperforming both baselines Schürholt et al. (2022a); Sch"urholt et al. (2024). This finding suggests that D2NWG accelerates convergence and improves performance compared to existing methods. This highlights its potential for faster and more efficient model initialization, making it valuable for transfer learning and real-world applications. Interestingly, on MNIST and SVHN, weights with higher initial performance tend to degrade during fine-tuning.

### 4.2.2 FINE-TUNING ON UNSEEN TASKS: MLP CLASSIFIER

**Task:** The objective remains the same as in Section 4.2.1, but here we evaluate the proposed method solely on unseen datasets.

**Datasets:** We assess D2NWG on a real-world dataset of 140 subsets with class counts ranging from 2 to 20, and 10 test sets with up to 1,566 classes. We use a two-layer MLP on top of a CLIP image encoder and fine-tune it on training datasets to collect the pretrained zoo.(see appendix A.5).

**Baselines:** The baseline methods are random initialization and a pretrained MLP previously trained on ImageNet.

**Results:** Figure 3 shows performance on four unseen datasets, where D2NWG achieves 99.04% initial accuracy on the dessert dataset, outperforming the randomly initialized model even after 50 epochs. D2NWG consistently accelerates convergence across all tasks, surpassing both random and pretrained initialization despite no class overlap between training and test datasets, demonstrating strong transferability. Additional results are provided in Table 22 of the Appendix.

Table 5: Task Conditioned LoRA parameters Generation. Adaptations are performed on a Roberta-Base model denoted Rob-B.

| Method | Parameters | SST-2 (Acc) | MRPC (Acc.) | CoLA MCC.) | QNLI (Acc.) | RTE (Acc.) | STS-B (PCC.) | Avg. |
|--------|-----------|-------------|-------------|-----------|-------------|-----------|--------------|------|
| Rob-B | 125M | **94.8** | 90.2 | 63.6 | **92.8** | 78.7 | 91.2 | 85.2 |
| LoRA | 0.9M | 95.1±0.2 | 89.7±0.7 | 63.4±1.2 | 93.3±0.3 | 78.4±0.8 | 91.5±0.2 | 85.2 |
| AdaLoRA | 0.9M | 94.5±0.2 | 88.7±0.5 | 62.0±0.6 | 93.1±0.2 | 81.7±0.6 | 90.5±0.2 | 85.0 |
| DyLoRA | 0.9M | 94.3± 0.5 | 89.5±0.5 | 61.1±0.6 | 92.2±0.1 | 78.7±0.7 | 91.1±0.2 | 84.5 |
| FourierFT | 0.6M | 94.2±0.3 | 90.0 ± 0.8 | 63.8±1.6 | 92.2±0.1 | 79.1±0.5 | 90.80 ± 0.2 | 85.0 |
| D2NWG | 0.6M | 94.3±0.1 +0.2 | **90.3**±0.5(↑0.3) | **64.3**±1.2 (↑0.5) | 92.6±0.2(↑0.5) | **79.6**±0.4(↑0.5 ) | 91.0±0.3(↑0.0.2) | 85.3(↑0.3) |

### 4.2.3 FULL MODEL FINE-TUNING ON UNSEEN TASKS

**Task:** We evaluate each method's generalization on CIFAR-10, STL-10, Pets and Aircrafts, focusing on performance gains in domain-specific tasks. The goal is to identify the best initialization strategy for improving model adaptability across diverse data distributions.

**Baseline:** The baseline in this experiment are the Pretrained model, which uses weights from a model pretrained on ImageNet and RandomInit, a randomly initialized model.

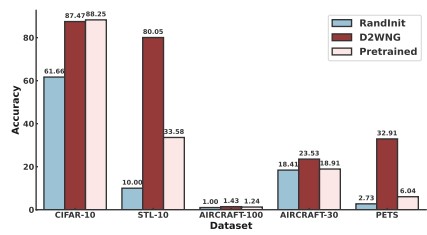

Figure 4: Fine-tuning on Unseen Tasks.

**Datasets:** In this experiment we evaluate the transferability to unseen dataset of D2NWG trained in Section 4.1.3 on unseen datasets CIFAR-10, STL-10, Aircraft100, Aircraft30, and Pets.

**Results:** We evaluated D2NWG by comparing it against 5 pretrained and 5 randomly initialized models, each fine-tuned for 1 epoch across CIFAR-10, STL-10, Aircraft100, Aircraft30, and Pets datasets. As shown in Figure 4, D2NWG consistently outperforms the baselines. Notably, on AIRCRAFT-100, D2NWG achieved 1.43% accuracy, surpassing both randomly initialized (1.0%) and ImageNet-pretrained (1.24%) models. These results demonstrate D2NWG's generalization and fine-tuning capabilities, even on specialized datasets.

### 4.3 TASK CONDITIONED LoRA WEIGHTS GENERATION

**Task:** In this section, we demonstrate that our method can be applied to LLMs by learning the distribution of LoRA matrices conditioned on task-specific textual descriptions.

**Datasets:** We use six tasks from the GLUE benchmark and generate task descriptions using GPT-4 (see Table 14). LoRA weights were generated following the fine-tuning process of Gao et al. (2024). We collected LoRA and classifier head checkpoints from the last 5 epochs, combined the pretrained vectors, and conditionally learned their distribution.

Table 6: Exploration of optimal weight space of some instruct LLMs using a diffusion model sampled weights. ↑ indicates the performance gain

| Methods | Winogrande (5 shot) | Arc-Challenge (25 shot) | Hellaswag (25 shot) |
|---------|---------------------|-------------------------|---------------------|
| LLAMA-3.1-8B-Instruct | 67.17 ± 0.01 | 64.93 ±0.01 | 78.58 ± 0.00 |
| D2NWG | 67.61± 0.02(↑0.44) | 65.74±0.01(↑0.81) | 78.86± 0.02(↑0.28) |
| Mistral-7b-Instruct | 69.93 ± 0.01 | 59.22 ±0.01 | 81.97 ± 0.00 |
| D2NWG | 70.80± 0.02(↑0.80) | 59.80±0.01(↑0.58) | 82.04± 0.00(↑0.07) |
| LLAMA-3.2-1B-Instruct | 56.75± 0.01 | 40.96 ±0.01 | 61.67 ± 0.00 |
| D2NWG | 57.17 ± 0.01(↑0.42) | 41.55 ± 0.01(↑0.59) | 61.70± 0.01(↑0.03) |

**Baselines:** We compare with base Roberta-base, LoRA (Hu et al., 2021), AdaLoRA (Zhang et al., 2023), DyLoRA (Valipour et al., 2022) and FourierFT (Gao et al., 2024) which are all LoRA-based RoBERTa-base models. We sampled and compared the average accuracy of the top 5 performing sets of weights per dataset.

**Results:** As shown in Table 5, D2NWG effectively generates weights that match or surpass the performance of pretrained models. These results align with our findings from the augmented weight retrieval experiments.

### 4.4 ENHANCING LLM PERFORMANCE WITH WEIGHT SAMPLING

**Task:** We aim to demonstrate that D2NWG can enhance existing LLMs by learning the distribution of their pretrained weights, enabling the generation of parameters that improve performance on specific tasks while generalizing to unseen tasks.

Table 7: Performance evaluation on unseen open llms leaderboard v2 benchmark. These results are produced by Huggingface after submission to open LLM leaderdoards. ↑ indicate performance improvement while ↓ indicate a performance decrease

| Method | ifeval (0) | Bbh (3) | Gpqa (0) | MATH-hard (4) | Musr (0) | MMLU-Pro (5) | Avg | Base Model | Fine-tuned |
|---|---|---|---|---|---|---|---|---|---|
| Llama-3.2-1B-Inst. | 56.78 | 8.74 | **3.36** | **2.96** | **2.97** | 7.58 | 13.76 | Llama-3.2-1B | Yes |
| D2NWG | **58.44**(↑1.66) | **8.82**(↑0.08) | 1.68(↓1.68) | 6.04(↑3.08) | 0.66(↓2.31) | **9.09**(↑1.51) | **14.12**(↑0.36) | Llama-3.2-8B-Instruct | No |
| SauerkrautLM-8B-Inst. | **80.17** | 31.00 | **5.37** | 11.18 | 11.52 | **32.12** | 28.56 | Llama-3.1-8B-Inst | Yes |
| D2NWG | 80.33 +0.16 | **31.10**(↑0.10 ) | 5.26(↓0.11) | **11.56**(↑0.38) | 11.52 | 32.07(↓0.05) | **28.64** (↑0.08) | SauerkrautLM-8B-Inst. | No |
| Lexi-Uncensored-V2 | 77.92 | 29.69 | 4.36 | 16.92 | 7.77 | 30.90 | **27.93** | Llama-3.1-8B-Inst. | Yes |
| Llama-3.1-8B-Inst. | **78.56** | 29.89 | 2.35 | **17.60** | 8.41 | 30.68 | 27.91 | Llama-3.1-8B | Yes |
| D2NWG | 77.85(↓0.71) | **30.39**(↑0.5) | **4.47**(↑2.12) | 17.52(↓0.08) | **9.64**(↑1.23) | **31.02**(↑0.34) | **28.50**(↑0.59) | Llama-3.1-8B-Inst. | No |

**Datasets:** We evaluate on several benchmarks(Beeching et al., 2023): AI2 Reasoning Challenge for grade-school science questions, HellaSwag for commonsense inference, Winogrande for commonsense reasoning.

**Baseline:** We evaluate our method against various version of LLAMA3 and Mistral-7B.

For each model, We extract the weights of the top 25% of layer excluding embedding and output layer, learn their distribution using chunk based encoding, We then steer through the optimal space to generate task-specific parameters as shown in Table 6.

**Results:** The results in Table 6 demonstrates that our approach consistently improve the performance of each models demonstrating new application avenues of our proposed method.

## 4.5 EVALUATION ON OPEN LM BENCHMARK

We combine the models frome the previous section following Wortsman et al. (2022) and evaluate them on the OpenLM leaderboard (Fourrier et al., 2024).

**Task:** We evaluate the robustnets of ours best models on the open-lm leaderboard.

**Datasets:** We evaluate models on 6 key benchmarks datasets: IFEval for instruction adherence, BBH (Big Bench Hard, with 23 challenging tasks (arithmetic, reasoning, language understanding), MATH focusing on Level 5 high-school math problems, GPQA with graduate-level Q&A across various fields, MuSR testing complex reasoning with long-range context, and MMLU-Pro for advanced multitask knowledge assessment. These benchmarks assess diverse reasoning and knowledge capabilities in and few-shot settings.

**Baselines:** We compare our method against LLMA3.1-8B-Instruct and its fine-tuned variant, with evaluations conducted on the leaderboard server.

**Results:** As shown in Table 7, our method surpasses baseline models on the leaderboard and performs comparably to models pretrained on task-specific datasets. Despite not being directly calibrated for leaderboard tasks, D2NWG achieves up to a 3% improvement in certain cases. This demonstrates the potential of guided parameter space exploration for task specialization. The consistent gains across benchmarks highlight D2NWG's effectiveness in enhancing model robustness and transferability, with our LLaMA-3.2-1B model ranking among the top LLaMA-3.2-1B entries on the public leaderboard.

**Quality Check:** Our method enhances text generation quality, as shown in Table 13.

## 5 CONCLUSION

In this work, we recast latent diffusion for dataset-conditioned neural network weight generation, enabling quick adaptation to novel datasets and efficient fine-tuning and transfer learning without training. Through extensive experiments on diverse datasets, our method generates high-quality weights for novel tasks and improves generalization. We extend parameter generation to large language models, demonstrating the scalability and versatility of our approach. Our method effectively encodes architectures with up to 1 billion parameters using a single GPU with less than 80GB, including task- or dataset-conditioned generation.

ACKNOWLEDGMENTS

This work was supported by Institute for Information & communications Technology Planning & Evaluation(IITP) grant funded by the Korea government(MSIT) (RS-2019-II190075, Artificial Intelligence Graduate School Program(KAIST)) and (No.RS-2022-II220713, Meta-learning Applicable to Real-world Problems), by Samsung Research Funding Center of Samsung Electronics (No. IO201210-08006-01), Institute of Information & communications Technology Planning & Evaluation (IITP) under Open RAN Education and Training Program (IITP-2024-RS-2024-00429088) grant funded by the Korea government(MSIT) and, by National Research Foundation of Korea (NRF) grant funded by the Korea government (MSIT) (No. RS-2023-00256259) and DeepAuto.ai.

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

## LIMITATION AND ETHICAL STATEMENT

**Limitations**: Our method relies on large collections of pretrained weight tensors and datasets, which require substantial storage and computational resources. However, such pretrained models are becoming more readily available due to the efforts made by open-source communities.

## A   APPROACH

**Broader Impact** D2NWG addresses the resource-intensive nature of deep learning by proposing a method for efficient transfer learning. This has the potential to reduce the computational resources required for training neural networks, making it more accessible to a wider range of researchers and organizations.

**Limitation** In this work, we focus mainly on generalization across datasets. Additionally, while the diffusion model achieves impressive performance on image generation, there are still some challenges to efficiently recast it for weights generation including memory constraint, convergence challenges and considerations of symmetries in the weight spaces of different neural network architectures.

### A.1   RELATIONSHIP BETWEEN DATASETS AND TRAINED WEIGHTS

Gradient descent based optimization is the commonly used technique to generate optimal neural network weights through training by minimizing a loss function, ie. cross-entropy for classification tasks. The weights optimized with gradient descent thus contains some information about the training data. Therefore, understanding the correlation between the training dataset and the optimal weights is important for the generation of weights. During the optimization process with gradient descent the weights of each layer $i$ are updated as $w_i = w_{i-1} - \eta \nabla_{w_i} \mathcal{L}(w_1, w_2, \ldots, w_n)$, where $\nabla_{w_i} \mathcal{L}(w_1, w_2, \ldots, w_n)$ is input dependent. As an example, let's consider a two-layer feedforward neural network:

$$
\begin{aligned}
&x : inputs \\
&l_1 = W_1 x + b_1 \qquad\qquad\qquad h = ReLU(l_1) \\
&h = ReLU(l_1) \qquad\qquad\qquad l_2 = W_2 h + b_2 \\
&\hat{y} = softmax(l_2) \qquad\qquad\quad J = CE(y, \hat{y})
\end{aligned}
$$

Analyzing the weights' update below, we can observe that the optimal weights are noisy perturbation of the inputs feature maps and all together they contain information about the training either related to the raw input or the feature map at a given stage.

$$
\begin{aligned}
\delta_1 &= \frac{\partial J}{\partial l_2} = (y - \hat{y})^T \\
\delta_2 &= \frac{\partial J}{\partial l_1} = \delta_1 W_2 osgn(h) \\
W_1^{(i+1)} &= W_1^{(i)} - \eta \nabla_{w_1} \mathcal{L}(w_1, w_2, b_1, b_2) \\
&= W_1^{(i)} - \eta \delta_2^T x \\
W_2^{(i+1)} &= W_2^{(i)} - \eta \nabla_{w_2} \mathcal{L}(w_1, w_2, b_1, b_2) \\
&= W_2^{(i)} - \eta \delta_1^T h^T
\end{aligned}
$$

### A.2   WEIGHTS VECTORIZATION

[] For a neural network with $L$ layers, the process of vectorizing the weights and biases for both fully connected and convolutional layers is as follows:

- For the $\ell$'th fully connected layer: $W^{(l)} \in \mathbb{R}^{d_{l-1} \times d_l} \rightarrow \text{vec}(W^{(l)}) \in \mathbb{R}^{d_{l-1} \cdot d_l}$ and $b^{(l)} \in \mathbb{R}^{d_l}$, the length of the vectorized weights for this layer, including the bias if it is not null, is given by $d_{l-1} d_l + d_l$.

- For the $\ell$'th convolutional layer: $W^{(l)} \in \mathbb{R}^{k_h \cdot k_w \cdot c_{in} \cdot c_{out}}$ and $b^{(l)} \in \mathbb{R}^{c_{out}}$, the length of the vectorized weights for this layer, including the bias if it is not null, is $k_h \cdot k_w \cdot c_{in} \cdot c_{out} + c_{out}$.

We then concatenate all the flattened weight and bias vectors resulting in a vector $\theta$: $\theta = \bigoplus_{l=1}^{L} \left( \text{vec}(W^{(l)}) \oplus b^{(l)} \right)$ where vec denotes the vectorization operation and $\oplus$ denotes concatenation. The concatenation operation keeps the ordering of weights in the network.

## A.3  LAYER SELECTION STRATEGY

To manage the large number of parameters in LLM architectures, where not all layers are required to be tuned to improve the performance, we propose focusing on the most important layers. These layers are identified using the Marchenko-Pastur (MP) distribution, which serves as a filter to highlight relevant weights while discarding those resembling random noise. The MP law provides a benchmark for distinguishing structured weights from noise by comparing the empirical eigenvalue spectrum of weight matrices to the MP distribution. D2NWG uses this *spectrum method* (Hartford et al., 2024) to learn the distribution of the most informative weights—those corresponding to eigenvalues that significantly exceed the MP upper bound. By focusing on these critical weights, D2NWG captures meaningful patterns in LLMs, leading to enhanced performance in transfer learning.

The spectrum method, grounded in random matrix theory, applies the Marchenko-Pastur (MP) distribution to different types of layers, treating them as rectangular random matrices. In transformer networks, functionally similar layers are grouped, such as a set for all query layers in multi-head attention. The method begins by computing the covariance matrix of each layer's weight matrix, $W \in \mathbb{R}^{m \times n}$, as $\Sigma = \frac{W^T W}{n}$, followed by eigenvalue extraction. Singular value decomposition (SVD), $W = USV^T$, is used to efficiently compute these eigenvalues from the diagonal matrix $S$, which contains the singular values. The resulting eigenvalues describe the variance captured by each principal component of the squared weight matrix and form what is known as the *empirical spectrum*. To analyze this spectrum, we compare it to the theoretical distribution of eigenvalues predicted by the Marchenko-Pastur (MP) distribution. This distribution $p(\lambda)$, in equation 7, characterizes the eigenvalue behavior of random covariance matrices as $m, n \to \infty$, with a fixed aspect ratio $q = \frac{m}{n}$ and variance $\sigma^2$.

$$p(\lambda) = \frac{1}{2\pi\sigma^2 q\lambda}\sqrt{(\lambda_+ - \lambda)(\lambda - \lambda_-)}, \tag{7}$$

where $\lambda \in [\lambda_+, \lambda_-]$, $\lambda_+ = \sigma^2(1 + \sqrt{q})^2$, and $\lambda_- = \sigma^2(1 - \sqrt{q})^2$. From 7, the correspoding bounds for eigen values of $W$ are $\sqrt{\lambda}/\sqrt{n} \in [\varepsilon_+, \varepsilon_-]$, $\varepsilon_+ = \frac{1}{\sqrt{n}}\sigma(1 + \sqrt{q})$, and $\varepsilon_- = \frac{1}{\sqrt{n}}\sigma(1 - \sqrt{q})$.

**Interpretation**: The Marchenko-Pastur (MP) distribution provides insight into the underlying structure of data or layer in our case:

- *Eigenvalues within MP bounds*: Likely represent noise, with their corresponding principal components carrying little meaningful information, indicating the layer's lower importance.
- *Eigenvalues larger than the upper MP bound $\lambda_+$*: Capture more variance than noise, suggesting the presence of true signals or patterns in the data.
- *Eigenvalues smaller than the lower MP bound $\lambda_-$*: May indicate compression or degeneration in the data structure.

Significant deviations, particularly large eigenvalues, indicate meaningful components that capture more variance than random noise, aiding in the identification of important features or signals. This insight is used to compute the signal-to-noise ratio (SNR), where eigenvalues below the upper bound are considered noise. The SNR is calculated as follows:

$$SNR = \frac{\sum_{k \,|\, |\sigma_k| \geq \varepsilon} \sigma_k}{\sum_{n \,|\, |\sigma_n| < \varepsilon} \sigma_n}. \tag{8}$$

## A.4  LEARNING THE DISTRIBUTION OF LLM WEIGHTS

Our method for LLM weight generation employs a layer-wise chunking mechanism that facilitates both layer-wise and chunk-wise sampling. Each layer is divided into independent chunks to form the

training data, and are then encoded with the VAE. During the diffusion process, an index is assigned to each chunk, and the model is trained using class-conditioned diffusion, where chunk indices serve as class labels. At sampling time, the chunk indices corresponding to each layer are grouped into clusters associated with that layer. These clusters are then used to sample new sets of chunks, which are concatenated to reconstruct the sampled weights for each layer.

After selecting the top 25% of the layers, we applied chunking with a size of 2,097,152 for LLaMA 3.2-1B and 4,194,304 for other models. We then performed sequential refinement using Algorithm 1. Unlike in vision tasks, LLM models are conditioned on chunk indices. Here, we refer to neural network operations such as dense layers and layer normalization as *layers*. The spectrum method provides an ordered set of these layers (q, k, v, o, mlp_up, mlp_down, mlp_gate). For architectures like Llama 3.1-8B and Mistral, we only learn the distribution of the top 8 each of these layers, excluding layer normalization. These layers are further divided into two groups: the top 4 and the second top 4, for which we build separate models to learn their distributions. As for the normalization layers, we learn the distribution across all of them. The maximum generated parameters is $\approx 872M$.

---

**Algorithm 1** Sequential Weight Model Improvement

---

1: **Input:** Initial weights $\Theta_{\text{init}} = \{\tilde{\theta}_1, \ldots, \tilde{\theta}_L\}$, Hypernetwork $\mathcal{H}_i$ for each layer $i$, Validation dataset $\mathcal{D}_{\text{val}}$, $K$ candidates per layer
2: **Output:** Final weights $\Theta^* = \{\theta_1^*, \ldots, \theta_L^*\}$
3: Initialize $\Theta^* = \Theta_{\text{init}}$
4: Compute initial validation accuracy: current_accuracy $= \mathcal{A}(\Theta_{\text{init}}, \mathcal{D}_{\text{val}})$
5: **for** each layer $i = 1$ to $L$ **do**
6:     Generate $K$ candidates $\{\theta_i^{(1)}, \ldots, \theta_i^{(K)}\}$ using $\mathcal{H}_i$
7:     **for** each candidate $k = 1$ to $K$ **do**
8:         Replace $\tilde{\theta}_i$ with $\theta_i^{(k)}$ in $\Theta^*$ to form $\Theta^{(k)}$
9:         Compute validation accuracy: $\mathcal{A}(\Theta^{(k)}, \mathcal{D}_{\text{val}})$
10:     **end for**
11:     Choose $\theta_i^* = \arg\max_k \mathcal{A}(\Theta^{(k)}, \mathcal{D}_{\text{val}})$
12:     **if** $\mathcal{A}(\Theta^{(k)}, \mathcal{D}_{\text{val}}) >$ current_accuracy **then**
13:         Update $\Theta^* = \Theta^{(k)}$
14:         Update current_accuracy $= \mathcal{A}(\Theta^*, \mathcal{D}_{\text{val}})$
15:     **else**
16:         Retain $\tilde{\theta}_i$ in $\Theta^*$
17:     **end if**
18: **end for**
19:
20: **return** $\Theta^*$

---

### A.5 MODELZOO AND PRETRAINED DATASETS

**Model zoo** We use the pretrained datasets from Schürholt et al. (2022c) as structured in Schürholt et al. (2022a). This dataset consists of 4 different datasets with 5000 pretrained weights per architectures and datasets. The details of the architecture used to generate the pretrained weights are available in Schürholt et al. (2022c).

**KaggleZoo** This modelzoo is generated using the dataset provided by Jeong et al. (2021). To efficiently generate the pretrained weights, we first compute the features of each image then use a MLP with two layers with input size 512, hidden size 256 and leaky ReLU activation functions. We train the MLP on clip features as it allows us to quickly generate high performing weights. For each datasets we used the last 10 checkpoints which results in 1400 pretrained weights for training.

**ImageNet zoo** To generate the pretrained modelzoo on ImageNet, we sample 1000, 5000, 10000 and 20000 subsets with 10 classes each with 100 images per class in the training set and 50 per class in the test set. For the 1000 and 5000 subsets we used the same MLP architecture as the KaggleZoo. For the 10000 subset, we reduce the hidden dimension to 128 and, for the 20000 subset we use a single linear probing layer. On the other datasets linear probing shows similar generalization performance

as the two-layer MLP. We use Adam optimizer with a learning rate of $1e-3$ and all models are trained for 30 epochs.

**Zoo for Few-shot learning**: The few-shot learning pretrained zoo is generated by fine-tuning the classifier head for 10 epochs on each of the 50,000 subsets.

**LLMs zoo**: We collected the pretrained LLM model from their original HugginFace repositories with no further pertaining on specific tasks or datasets.

**Meta-album datasets**: We split the meta-album dataset into a training set (70%) and a test set (30%). Next, we trained the MobileNetV3 OFA subnet with parameters $d = 2$, $k = 3$, and $e = 3$ for 100 epochs. Checkpoints from the last 20 epochs were collected as training data. A detailed breakdown of the dataset can be found in Table 12.

## A.6  DETAILS OF THE PROPOSED MODEL

We build our dataset conditioned weight generation model using latent diffusion (Rombach et al., 2021).

**AutoEncoder**: We use the same VAE modules of latent diffusion and use the same architecture for all experiments except adaptation of the inputs and output dimensions. We insert a linear layer before the first layer of the encoder such that we can reshape its output to a representation for the convolution layers. Similarly, a linear layer is placed at the last layer of the decoder adapting the output to the vectorized weights representations. For the VAE loss function we removed the discriminator in the original latent diffusion VAE loss function.

**Diffusion Model**: We utilize same UNet architecture as in latent diffusion with the same training procedure.

**Dataset Encoding Mechanisms** We investigated three different mechanisms of dataset encoding. Firstly, we use Set Transformer (Lee et al., 2019a) which can be difficult to train when optimized together with the diffusion using the weights encoder from the VAE and the Set Transformer.

Table 8: Models seting, $n$ and $c$ in the dataset configuration represent respectively the number of samples per class n=5 for training and c the total number of classes per dataset. The VAE and the diffusion models share similar configuration and architectures as (Rombach et al., 2021)

In addition to the Set Transformer, we explored a two-layer MLP model as the dataset encoder. The first layer is a dynamic linear layer with a maximum input feature size set to $n_{\max} \cdot c_{\max}$, where $n_{\max}$ is the maximum number of images per class and $c_{\max}$ is the maximum number of classes among all subsets of the pretrained datasets. The shape of the image features in each dataset obtained with the CLIP image encoder is $x \in \mathbf{R}^{c \times n \times d}$, where $d$ is the feature dimension for each corresponding pretrained weight vector. While the Set Transformer-based encoder uses these inputs directly, the MLP encoder reshapes each input from $x \in \mathbf{R}^{c \times n \times d}$ to $x \in \mathbf{R}^{d \times (n \cdot d)}$ and then applies the dynamic linear layer. If a dataset has more classes or samples than $c_{\max}$ and $n_{\max}$ respectively, we only consider the first $c_{\max}$ classes and $n_{\max}$ samples per class. If the dataset has fewer classes or samples, we adjust the dynamic linear layer dimensions accordingly. The output of the dynamic linear layer is $z \in \mathbf{R}^{d \times h}$, where $h$ is an arbitrarily chosen number greater than zero. We

| Parameters | Values |
|---|---|
| Epochs | [50, 2000] |
| VAE | |
| Optimizer | Adam |
| Learning Rate | 1e-3 |
| Latent Dimiension | 1024 |
| KL-Divergence Weight | 1e-6 |
| Dataset Encoder | |
| Architecture | Set Transformer |
| Input Dimension | $c \times n \times 512$(min) |
| Output Dimension | 1024 (min) |
| Depth of Set Transformer | 2 |
| Diffusion | |
| Optimizer | AdamW |
| Learning Rate | 1e-4 |
| Scheduler | Linear |
| Time step | 1024 |
| Network | Unet |
| UNet Input Size | $(c \times 32 \times 32)$ |

then reshape $z$ from $\mathbf{R}^{d \times h}$ to $\mathbf{R}^{1 \times (h \cdot d)}$ (with $h \cdot d$ fixed) and apply the final linear layer to obtain the desired output. This model can be jointly optimized with the diffusion model while achieving good performance.

**Dataset Encoding with Set Transformer** We use the Set Transformer for dataset encoding, pretrained as described in Lee et al. (2021). The approach involves using the frozen Set Transformer and adding

---

**Algorithm 2** Datasets Encoder Training

---

**Input:** pretrained weights $x$, image features $y$, batch_num $m$
Instanciate $\mathcal{T} = $ Set Transformer, Load pretrained Encoder ($\mathcal{E}$).
**repeat**
    Initialize $loss = 0.0$
    **for** $i = 1$ **to** $m - 1$ **do**
        $x_i \sim x, \mathfrak{D}_i \sim \mathfrak{D}$
        $z_i = \text{Encoder}_{\text{VAE}}(x_i)$
        $z_{\mathfrak{D}_i} = \mathcal{T}(\mathfrak{D}_i)$
        $loss = loss + \mathcal{L}_{CLIP}(z_i, z_{\mathfrak{D}_i})$ (Equation **??**)
    **end for**
    Update weights of $\mathcal{T}$
**until** convergence

---

a single linear layer to adapt its output to our specific problem, utilizing it as the dataset encoder. This method reduces the computational cost of training the Set Transformer and enables joint optimization of the dataset encoder and the diffusion model. The results of these data set encoding schemes are presented in Table 21 for the Hyperzoo dataset.

## B  TRAINING DETAILS

In this section, we describe the training steps used to train our method.

- **Pretrained Zoo Generation:** For classifier head adaptation, we first compute the features for all datasets. Then, we train the classifier head to generate the pretrained zoo.

- **VAE Training:** We train the VAE to encode the pretrained weights following Equation 1. Additionally, a pretrained performance predictor can be used to predict the performance of the reconstructed weights and guide the VAE training as described in Equation 9.

- **Dataset Alignment:** If using dataset alignment, we pretrain the Set Transformer to align the pretrained weights' latent representations. This is done using the frozen encoder of the VAE and the dataset embeddings. The inputs to the Set Transformer are image features, with five image features per class.

- **Diffusion Process Training:** We train the diffusion model while keeping the Set Transformer and the VAE models frozen. If an MLP is used for dataset encoding, we jointly optimize the diffusion process with the MLP dataset encoder.

Although the dataset encoder can be optimized together with diffusion model, we train them separately to speed up the training process and reduce memory requirements. The VAE and the dataset encoder are trained using the Adam optimizer with a learning rate of $1e-4$. The diffusion model in each experiment is trained with a linear scheduler, a base learning rate of 1e-4, and the AdamW optimizer (Rombach et al., 2021). During the training process of the diffusion model, the output of the dataset encoder is concatenated with the latent representation of the input weights, forming the input to the UNet model. Additionally, we investigate joint training of the diffusion process in the ablation study and Appendix C.5 and A.6. Further details can be found in Table 8.

### B.1  PREDICTOR TRAINING

To improve the reconstruction and sampling efficiency, we trained an accuracy predictor $g$ from pretrained weights $w$ then use the frozen predictor during the training of the VAE as a regularizer as shown below:

$$\min_{\theta,\sigma} \frac{w - f_\theta(w)}{\sigma^2} + \log \sigma^2 + ||g(w) - g(f_\theta(w))||^2, \tag{9}$$

where $g(w)$ is the embedding of the original input and $g(f_\theta(w))$ is the predictor embedding of the reconstructed weights. The predictor can be either dataset-conditioned or unconditioned. In general we found that dataset-conditioned predictor works only well for large number of samples per dataset.

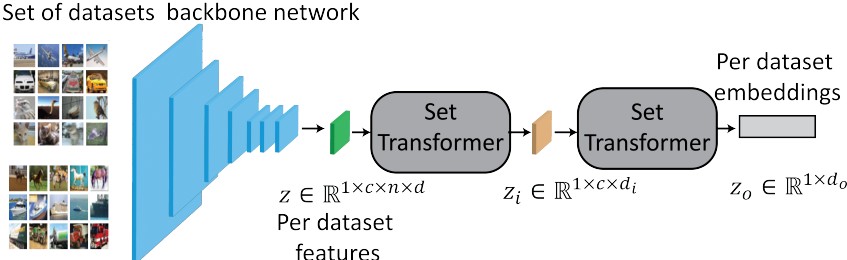

Figure 5: Overview structure of the set-transformer-based dataset encoder. For each pretrained dataset we use $n = 5$ images per class and the embedding dimension $d_0 = 1024$.

---

**Algorithm 3** Predictor-Guided VAE

---

**Input:** Pretrained weights $x$, accuracy $y$, batch_num $m$
Instantiate $f$ = Set Transformer, and load pretrained predictor $g$).
**repeat**
    Initialize $loss = 0.0$
    **for** $i = 1$ **to** $m - 1$ **do**
      $\bar{x} = f_\theta(x), \bar{y} = g(\bar{x})\ \hat{y} = g(x)$
      $L_\theta \frac{x-\bar{x}}{\sigma^2} + \log \sigma^2 + ||\hat{y} - \bar{y}||^2$
    **end for**
    Update weights of $f$
**until** Convergence

---

After the AutoEncoder is trained, we train the dataset-conditioned module which requires a dataset encoder.

## C    ABLATION STUDY

### C.1    CAN THE PROPOSED METHOD HANDLE MULTIPLE ARCHITECTURES?

This section provides a simple way to handle the case where the pretrained zoo contains multiple architectures per task or dataset. Since the number of architecture and dataset are predefined, it is possible to build a set of unique index for each combination of dataset-architecture pairs. An alternative will be to encode the graph representation of the architectures then used that as conditioning. In this ablation study we use the simple class indexing approach to demonstrate the versatility of our method. We use CIFAR10 and CIFAR100 as the dataset and as target architectures we utilze a ResNet44 trained on CIFAR-100 with 667,188 parameters and a ResNet44 trained on CIFAR-10 with 661,338 parameters and finally, a MobileNetV2 trained on CIFAR-10 with 700,490 parameters. All models were zero-padded to 700,490 parameters, combined into a unified dataset, and trained without chunking. The results in Table 9 demonstrate that the proposed method is capable of simultaneously learning the distributions of diverse architectures trained on diverse datasets.

| Model | ResNet44 (CIFAR-10) | ResNet44 (CIFAR-100) | MobileNetV2 (CIFAR-10) |
|---|---|---|---|
| Pretrained | 94.01 | 71.63 | 92.88 |
| D2NWG | 94.10 $\pm$0.09 | 71.64$\pm$0.02 | 93.11$\pm$0.20 |

Table 9: Performance evaluation on mixed architectures.

## C.2 TRANSFERABILITY

As demonstrated in Table **??**, our approach achieves performance comparable to existing methods while relying on a **single generative model** instead of 38 task-specific pretrained models. Notably, the pretrained model architecture and parameter counts used in this study are publicly available on a non-affiliated GitHub repository: `https://github.com/chenyaofo/pytorch-cifar-models`.

### EVALUATING SAMPLING FOR TRANSFER LEARNING

We compared sampling from a distribution of diverse pretrained models against traditional single-model transfer learning, using **ResNet-56** and our generative model trained on weights from 19 diverse architectures pretrained on CIFAR-10 and CIFAR-100. We tested three experimental setups:

1. **Direct evaluation** of the pretrained models.

2. **Sampling conditioned on training sets** (e.g., STL-10, CIFAR-10).

**Results** show that our approach consistently outperforms single-model transfer learning. Notably, there is no significant difference between training- and test-conditioned sampling when drawn from the same distribution, demonstrating the robustness of our method. This highlights the practicality of leveraging diverse pretrained model distributions for improved generalization.

Table 10: Performance on CIFAR10.1 and STL10 of D2NWG trained on diverse architectures

| Model | CIFAR10.1 | STL10 |
|---|---|---|
| Pret-cifar10 | 75.20 | 32.37 |
| Pret-cifar100 | 0.25 | 0.12 |
| Ours | $83.10 \pm 0.06$ | $35.41 \pm 0.13$ |
| Ours(test) | $83.04 \pm 0.06$ | $35.47 \pm 0.12$ |

## C.3 EFFECT OF MODELZOO SIZE GENERALIZATION

Here we investigates the impact of increasing the number of pretrained datasets on performance with experiments that use model zoos of sizes 5000, 10,000, and 20,000, derived from ImageNet subsets. Unseen target datasets CIFAR-10 and STL-10 are used. Sampling 50 weights, the average performance of the top 5 performing weights is shown in Figure 6a.

**Results:** On CIFAR-10 and STL-10, we obtain accuracies of $39.60 \pm 1.31\%$ and $44.66 \pm 0.55\%$ for 5000 subsets, $42.15 \pm 2.12$ and $64.83 \pm 2.83\%$ for 10000 subsets, and $52.64 \pm 3.12\%$ and, $80.49 \pm 1.77\%$ for 20000 subsets. The maximum accuracies with random initialization are $12.11\%$ and $17.12\%$ on CIFAR-10 and STL-10 without fine-tuning. This experiment demonstrated that increasing the number of datasets enhances the generalizability of the proposed method.

## C.4 SAMPLING WITHOUT LATENT REPRESENTATION

This section explores a model variant that directly learns the diffusion model on weights, bypassing the AutoEncoder stage, and compares it to the standard approach. Both variants are trained on 1000 subsets of ImageNet, and evaluated in in-distribution sampling setting on three randomly selected subsets from the 1000 subsets. The results, presented in Figure 6b, indicate that learning the distribution of pretrained weights in the latent space is notably successful in generating high-performing weights. The failure of the DDPM process on raw pretrained weights may stem from their higher model capacity requirement.

## C.5 CLIP-BASED DATASET ENCODING

In this section, the comparison between the CLIP-based dataset encoding scheme trained at an intermediate stage and the Set Transformer encoder jointly trained with the diffusion process is explored. Experiments are conducted on 140 Kaggle datasets and their respective model zoos. The

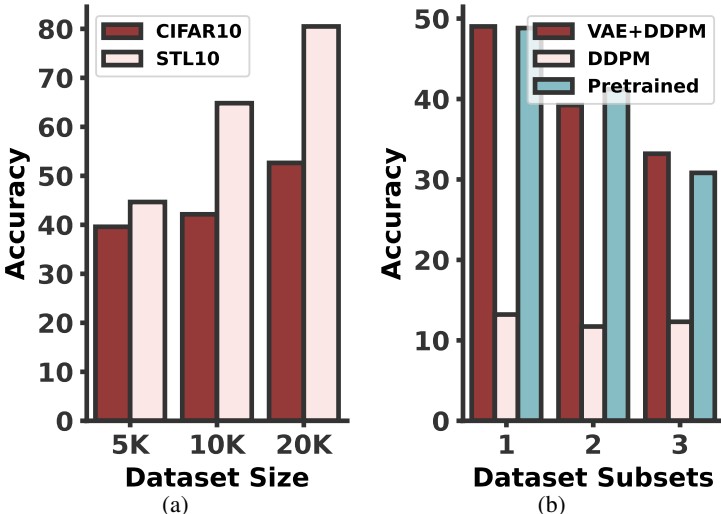

Figure 6: (a) Effect of the number of pretrained datasets on sampling weights performance on unseen datasets. (b) Performance comparison on in-distribution sampling of methods with VAE+DDPM vs DDPM

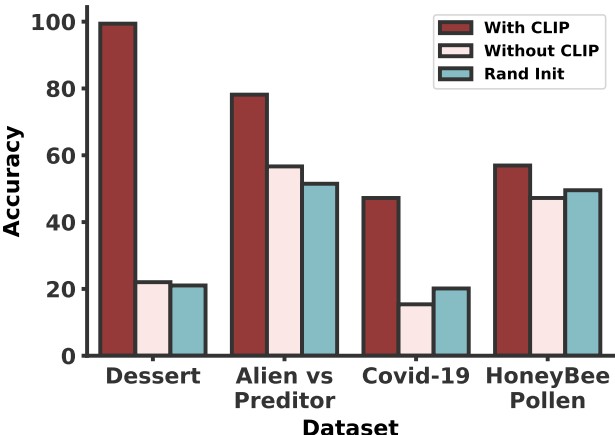

Figure 7: Performance comparison at initialization of method with jointly trained set-transformer (Without CLIP) and method clip-based dataset encoder.

results depicted in Figure 7 indicate that both methods achieve similar results for small numbers of datasets during the in-distribution sampling. However, as the number of datasets increases, the Set Transformer jointly trained with the diffusion approach faces challenges in convergence and requires more computational resources, as demonstrated in Figure 7.

## C.6 UNCONDITIONAL SAMPLING

We conduct the experiment using ResNet18 pretrained on CIFA-100 and CIFAR-10. For all datasets, the weight vector length is 2048 and we compare with pdiff (Wang et al., 2024). While pdiff requires a separate model for each dataset, our method combines the pretrained weights into a single dataset and conditionally learns their distribution. The sample size for each dataset in our method is 200, with a combined total of 400 parameters. The results are provided in Table 11 for 100 sampled weights. Two separate models for are trained for pdiff, CIFA10-pdiff and CIFAR100-pdiff while our method consists of a single model trained once for both datasets. It can be seen that our method outperformance the baseline (Wang et al., 2024) in Table 11.

Table 11: Unconditional Sampling Evaluation against Wang et al. (2024) on ResNet18.

| Dataset | CIFAR-10 | | | | CIFAR-100 | | | | Runtime |
|---------|----------|--------|------|---------------------|-----------|--------|------|---------------------|---------|
| Method | Avg | Median | Max | #Epochs for VAE,DDPM | Avg | Median | Max | #Epochs for VAE,DDPM | |
| pdiff | 94.46 | 94.46 | **94.52** | 8999,47999 | 76.1028 | 76.13 | 76.21 | 32999,38999 | $\approx 3h$ |
| D2NWG | 94.46 | **94.47** | 94.50 | 100,200 | **76.1796** | **76.18** | 76.24 | 100,200 | $\approx 1h30$ |

## C.7 COUPLING WITH AN ACCURACY PREDICTOR

This section reports the extended results of Table 19 in which we compared our method in-distribution and out-of distribution with and without accuracy predictor.

**Results.**: The full results of Table 19 are reported in Table 20. Using an accuracy predictor enable easily selecting highly performing when sampling in-distribution. However, in our case the accuracy predictor struggles to generalize well for unseen dataset as shown in Table 20

## C.8 SAMPLED WEIGHTS ANALYSIS

In this section, we analyze the characteristics of the sampled weights and compare them to the pre-trained ones based on experiments with the model zoo and a model pre-trained on a subset of ImageNet. The proposed method samples weights with a large variance, as shown in Figure 10, providing a broad range of initialization choices, from weights with low initial performance to those with higher initial performance.

Table 12: Details description of meta-album datasets

| **Domain** | **Original Dataset** | **# Classes** |
|------------|----------------------|---------------|
| Large Animals | Animals with Attributes, Dogs, Birds | 50, 120, 315 |
| Small Animals | Insects, Insects 2, Plankton | 117, 102, 102 |
| Plants | Fungi, PlantNet, Flowers | 25, 25, 102 |
| Plant Diseases | PlantDoc, Medicinal Leaf, Plant Village | 27, 26, 38 |
| Microscopy | Subcel. Human Protein, Pan-Nuke, Bacteria | 21, 19, 33 |
| Remote Sensing | RSD, RSICB, RESISC | 43, 45, 45 |
| Vehicles | Boats, Airplanes, Cars | 26, 21, 196 |
| Manufacturing | Textures ALOT, Textures DTD, Textures | 250, 47, 64 |
| Human Actions | MPII Human Pose, Stanford 40 Actions, 73 Sports | 29, 40, 73 |
| OCR | Omniprint-MD-6, Omniprint-MD-5-bis, Omniprint-MD-mix | 703, 706, 706 |

## C.9 EVALUTAION ON LARGE DATASETS

We investigate how our method perform for combined large and small dataset as well for mixed architectures. For this experiment we collect the pretrained weights from PyTorch hub with one checkpoints per datasets(CIFAR-10, CIFAR-100, and ImageNet-1k). After conditionally learning the combined weights distribution, we sampled 10 weights for each datasets and report the average accuracy of the top-3 datasets in Table 15. As shown in Table 15, D2NWG consistently produced high performing weights for each dataset from a single pretrained checkpoint.

Table 13: Comparison between Base Model and Improved Model Responses

| Instruction | Base Model Response | Improved Model Response |
|---|---|---|
| *Prompt: {"role": "system", "content": "You are an AI chatbot"}, {"role": "user", "content": "Explain what AI is and its importance"},* | **What is AI?** Artificial Intelligence (AI) refers to the development of computer systems that can perform tasks that would typically require human intelligence, such as:

1. **Learning**: AI systems can learn from data, experiences, and environments to improve their performance.
2. **Reasoning**: AI systems can reason and make decisions based on the data they have learned.
3. **Problem-solving**: AI systems can solve complex problems and adapt to new situations.
4. **Perception**: AI systems can perceive and understand the world around them through sensors, cameras, and other devices.

AI systems use various techniques, such as machine learning, deep learning, natural language processing, and computer vision, to achieve these capabilities.
**Types of AI:**

1. **Narrow or Weak AI**: Designed to perform a specific task, such as facial recognition or language translation.
2. **General or Strong AI**: Designed to perform any intellectual task that a human can, such as reasoning, problem-solving, and learning.
3. **Superintelligence**: Significantly more intelligent than the best human minds, potentially leading to exponential growth in technological advancements.

**Importance of AI:**

1. **Automation**: AI can automate repetitive and mundane tasks, freeing up human resources for more strategic work. | **What is Artificial Intelligence (AI)?** Artificial Intelligence (AI) is a branch of computer science that focuses on creating intelligent machines that can think, learn, and behave like humans. AI involves the development of algorithms, statistical models, and computer programs that enable machines to perform tasks that typically require human intelligence, such as:

1. **Reasoning and problem-solving**: AI systems can analyze data, identify patterns, and make decisions based on that data.
2. **Learning**: AI systems can learn from experience, adapt to new situations, and improve their performance over time.
3. **Perception**: AI systems can interpret and understand data from sensors, such as images, speech, and text.
4. **Natural Language Processing (NLP)**: AI systems can understand and generate human language, enabling them to communicate with humans.

**Importance of AI** AI has become increasingly important in various aspects of our lives, including:

1. **Automation**: AI can automate repetitive and mundane tasks, freeing up human time for more creative and strategic work.
2. **Improving decision-making**: AI can analyze large datasets, identify patterns, and provide insights that inform business decisions.
3. Enhancing customer experience: AI-powered chatbots and virtual assistants can provide 24/7 support and personalized services. |
| *prompt; {"role": "system", "content": "You work at a coffee shop"}, {"role": "user", "content": "what title do you give to someone who serves the coffee to customers"},* | At our coffee shop, we call the person who serves the coffee to customers a 'Barista'. However, some of our team members are also referred to as 'Cafe Attendants' or 'Coffee Servers', depending on their specific role and responsibilities. But generally, the title 'Barista' is the most commonly used and well-known term in the industry. | In a coffee shop, the person who serves the coffee to customers is typically referred to as a: Barista: This is the most common title, and it refers to a skilled person who prepares and serves coffee drinks, as well as other beverages and food items. Coffee Server: This title is also used to describe someone who serves coffee and other beverages to customers. Coffee Barista or Coffee Specialist: These titles are often used in specialty coffee shops or high-end establishments to emphasize the barista's expertise and knowledge of coffee. In our coffee shop, I'm a barista, and I'm happy to serve you your favorite coffee drink. |

## C.10 Generating the Full Weights for ResNet18

We investigate how our method performs when used to generate the full parameters for a ResNet18 model pretrained on MNIST, CIFAR-10, and CIFAR-100. In total, we use 100 pretrained weights per dataset and conditionally learn their distribution. The modelzoo generation follows the same setting as Wang et al. (2024). Table 16 demonstrates the effectiveness of our method for generating the entire weights of a network.

## C.11 Generating Weights for MobileNetV3

So far, our focus has been on model zoos populated by relatively simple classifier heads. In this section, we evaluate our method using MobileNetV3, a subnetwork sampled from OFA (Cai et al., 2020), consisting of 2.8 million parameters fine-tuned on CIFAR-10, STL-10, SVHN and MNIST for 15

Table 14: Glue benchmark tasks descriptor used in the experiment on glue datasets.

| Task Name | Description |
|---|---|
| **SSTB** | Predict the similarity score between two sentences. Rate their similarity on a scale from 0 to 5, where 0 indicates no meaning overlap, 1 indicates very little overlap, and 5 indicates complete overlap in meaning. |
| **MRCP** | Determine the semantic equivalence of two given sentences (Sentence 1 and Sentence 2). If the sentences are semantically equivalent, return 1. If they are not, return 0. |
| **SST2** | Determine the sentiment of a given sentence. Respond with 0 if the sentiment is negative and 1 if the sentiment is positive. |
| **COLA** | Evaluate whether the given sentence is both syntactically and semantically correct. If it is, respond with "1"; otherwise, respond with "0". |
| **QNLI** | Evaluate whether the given response properly answers the provided question. If the response answers the question correctly, return 0; otherwise, return 1. |
| **RTE** | Determine if a given hypothesis is true (entailment), false (contradiction), or undetermined (neutral) based on a provided premise. |

Table 15: Evaluation on Large Datasets

| Datasets | CIFAR10 (ShuffleNet) | | CIFAR100 (ShuffleNet) | | ImageNet-1k (SqueezeNet) | |
|---|---|---|---|---|---|---|
| Methods | Top1 | Top5 | ToP1 | Top5 | Top1 | Top5 |
| Pretrained | 92.98 | 99.73 | 72.39 | 91.46 | 58.178 | 80.624 |
| Ours(sampling) | **93.14 $\pm$ 0.25** | **99.76$\pm$ 0.22** | **72.60 $\pm$ 0.15** | **91.29 $\pm$ 0.13** | **58.257 $\pm$ 1.022** | **81.01$\pm$ 1.251** |

epochs. We collect the last 10 checkpoints per dataset and utilize our method to learn the distribution of pretrained weights. Furthermore, we combine the pretrained weights of MNIST and CIFAR-10, learn their distribution, and then evaluate our method on SVHN and STL-10. Subsequently, we reverse this process by combining the pretrained weights of SVHN and STL-10, and evaluate our method on MNIST and CIFAR-10.

As shown in Table 23 our method enhances the performance of the pretrained model. Furthermore, we note that learning the full model weights does not compromise performance. Although learning the distribution of the classifier head is computationally efficient, it can result in lower performance.

## C.12 GENERATING WEIGHTS FOR VISION TRANSFORMERS

Our method shows the ability to learn the distribution of all parameters within a vision transformer, including convolutional and linear layers. We present in-distribution evaluation results in plot Figure 9, highlighting the learning of combined weight distributions conditioned on individual datasets. The model zoo for ViTs is collected based on models proposed by Gani et al. (2022).

Table 16: **Zero-Shot Transfer Learning** This Table represent results of zero-shot evaluation against the pretrained model on Resnet18 full model architecture.

| Model | MNIST | CIFAR-10 | CIFAR-100 |
|---|---|---|---|
| Pretrained | 99.61 | 94.56 | 75.86 |
| D2NWG(ours) | $99.62 \pm 0.07$ | $94.57 \pm 0.00$ | $75.83 \pm 0.02$ |

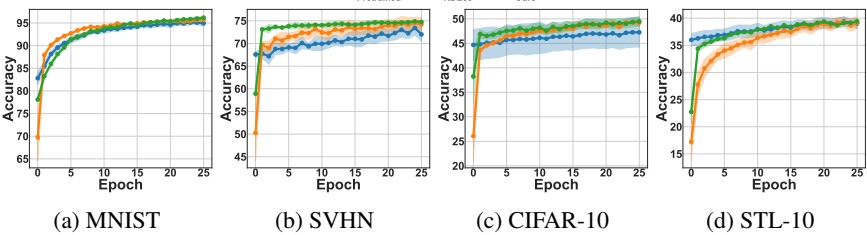

| (a) MNIST | (b) SVHN | (c) CIFAR-10 | (d) STL-10 |

Figure 8: **Convergence Plots on Finetuning Generated Weigths:** Weights generated by the competing methods are finetuned for 25 epochs on the training set. We utilize the modelzoos of Schürholt et al. (2022c).

# D  APPLICATION TO LARGE LANGUAGE MODEL (LLM) OUTPUT LAYER GENERATION

**Phi-3-MINI-4K-Instruct:**We conduct experiments on the Microsoft Phi-3-MINI-4K-Instruct model to demonstrate the scalability of our method for generating output layers in large language models (LLMs). The model's 98.5 million-parameter output layer was split into 96 chunks, each of size 1,026,048, and used as training data for a Variational Autoencoder (VAE) with an embedding size of 1,024. Lacking access to original training data, we used a class-conditional diffusion process, with chunk embeddings as conditioning data. Post-training, conditioned chunks were sampled and concatenated to reconstruct the original output vector. We evaluate our method using the Open-LLM LeadearBoard-1. As shown in Table 17, our approach effectively scales to the LLMs head generation demonstrating adaptability across diverse domains with minimal adjustments to conditioning data.

Table 17: Generating weights for the Microsoft Phi-3 language model output head.

| Methods | ARC Challenge (25-shots) | ARC Easy (25-shots) | HellaSwag (10-shots) | Winogrande (5-shots) |
|---|---|---|---|---|
| Pretrained | $87.16 \pm 0.00$ | $63.23 \pm 0.01$ | $73.65 \pm 0.01$ | $76.64 \pm 0.01$ |
| D2NWG | $87.36 \pm 0.01$ | $63.74 \pm 0.01$ | $73.65 \pm 0.00$ | $76.72 \pm 0.01$ |

**GPT2:** In this experiment, we show that our method can learn the distribution of any layer in an LLM by modeling the full distribution of GPT-2 small (164M parameters). We use a chunk size of 1,523,712 and, unlike Llama architectures, concatenated all vectorized layer weights before chunking them uniformly. Table 18 highlights the method's effectiveness on the Open LM-Leaderboard benchmark. While it did not outperform the base model overall, it significantly improved performance on certain tasks and maintained average accuracy comparable to the pretrained model.

## D.1  FAST CONVERGENCE PERFORMANCE EVALUATION

In this section we report supplementary results for experiment on tiny model zoo dataset. The pretrained weights used here are from epochs 21 to 25 for each dataset where 70% of the resulting modelzoo is used for training and 15% for validation and testing respectively. The number of pretrained weights in the modelzoos are 3500 for MNIST, CIFAR-10, and STL-10, and 2864 for SVHN. The flattened network weights' length is 2864 for CIFAR-10 and STL-10 and, 2464 for MNIST and SVHN. We pad all the weights with zero to 2864.

Table 18: Performance evaluation on unseen open llms leaderboard v2 benchmark base on full gpt2-164M small. These results are produced by Huggingface after submission to open LLM leaderdoards. ↑ indicate performance improvement while ↓ indicate a performance decrease

| Method | ifeval (0) | Bbh (3) | Gpqa (0) | MATH-hard (4) | Musr (0) | MMLU-Pro (5) | Avg | Base Model | Fine-tuned |
|---|---|---|---|---|---|---|---|---|---|
| openai-community-gpt2 | 17.8 | 2.83 | **1.12** | 0.3 | **13.91** | 1.84 | 6.3 | na | Yes |
| D2NWG | **19.16**(↑1.36) | **2.85**(↑0.02) | 1.01(↓0.11) | **0.38**(↑0.08) | 12.68(↓1.23) | **1.68**(↓0.16) | 6.29(↓0.01) | openai-community-gpt2 | No |

Table 19: **No Fine-tuning Initialization on Unseen Datasets** We transfer from one dataset, or combinations of datasets, to unseen datasets at test time.

| Source | Target | Accuracy | Methods |
|---|---|---|---|
| MNIST | SVHN | 13.25 | |
| SVHN | MNIST | 29.30 | $S_{KDE30}$ |
| CIFAR-10 | STL-10 | 15.20 | |
| STL-10 | CIFAR-10 | 15.40 | |
| Sampling from Combined Weights Distribution | | | |
| MNIST+CIFAR-10 | SVHN | 18.80 | |
| MNist+CIFAR-10 | STL-10 | 16.21 | Ours |
| SVHN + STL-10 | MNIST | 36.64 | |
| SVHN + STL-10 | CIFAR-10 | 18.00 | |

## D.2 SAMPLING WEIGHTS FOR UNSEEN DATASETS

**Task:** We evaluate the transferability of the models on unseen datasets. We create disjoint modelzoos by combining MNIST and CIFAR-10 into a single modelzoo and combining the SVHN and STL-10 modelzoos. When we train on the MNIST plus CIFAR-10 modelzoos, we test on the SVHN and STL-10 modelzoos and vice-versa.

**Results:** As shown in Table 19, D2NWG is able to sample weights with higher accuracy on unseen datasets as well as for in distribution. Through these experiments our method does not only outperform the baseline it also demonstrates promising results for dataset-conditioned sampling for unseen datasets.

## E MISCELLANEA

In Table 24 we present the parameter count for the model used to learn the distribution of the 25% of llama-3.2-1B transformer blocks. In Table 25 we showcase the set of experiments and the corresponding number of parameters generated by D2NWG. Although D2NWG is capable of generating up to 1 billion parameters, all our experiments were limited to a maximum of 872 million, achieved using the Llama 3.1-8B model with 4 transformer layers, excluding layer normalization, for which we constructed a separate model. This parameter count makes D2NWG the only method, to the best of our knowledge, capable of generating nearly a billion parameters, significantly enabling large architecture weights generation including GPT-2 and most existing image classification models in terms of parameter scale. For non-LLM models, we utilize joint distribution learning, enabling task

| Model | MNIST | SVHN | CIFAR-10 | STL-10 |
|---|---|---|---|---|
| Pretrained | 99.42 ± 0.05 | 94.62 ± 0.18 | 93.51 ± 0.16 | 94.01 ± 0.10 |
| Linear_prob | 96.88 ± 0.45 | 57.23 ± 0.28 | 82.85 ± 0.25 | 95.63 ± 1.23 |
| D2NWG(ful) | **99.55 ± 0.02** | **95.13 ± 0.10** | **94.23 ± 0.27** | 94.02 ± 0.10 |
| D2NWG(rob) | 97.56 ± 0.26 | 57.41 ± 0.17 | 83.64 ± 0.47 | **95.74 ± 0.74** |
| Cross datasets transfer learning | | | | |
| OFA (Pretrained)Cai et al. (2020) | 13.34 | 8.90 | 13.34 | 8.90 |
| D2NWG(full) | **66.82 ± 0.65** | **35.20 ± 0.65** | **36.70 ± 0.18** | **51.50 ± 0.37** |
| D2NWG(prob) | 42.86 ± 0.62 | 20.974 ± 0.78 | 26.56 ± 1.22 | 47.33 ± 0.32 |

Table 23: MobileNet Weight Generation.

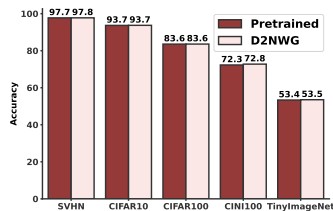

Figure 9: Experiment with ViT

Table 20: Performance evaluation at initialization without fine-tuning. For the baseline we use weights of SVHN for MNIST and vice versa similarly for CIFAR-10 and STL-10

| Datasets | MNIST | SVHN | CIFAR10 | STL10 |
|---|---|---|---|---|
| Random | 10.23±0.56 | 12.21±3.76 | 9.98±1.47 | 9.56±1.02 |
| Pretrained models | 82.82± 1.38 | **67.57± 0.59** | 44.68± 3.15 | **35.99± 1.15** |
| $S_{kde30}$Schürholt et al. (2022a) | 69.73± 5.12 | 50.25± 6.12 | 26.06± 3.01 | 17.20± 3.43 |
| seen (D2NWG) | 83.92±1.92 | 61.81 ± 3.13 | 43.08±0.55 | 31.45±0.35 |
| seen(D2NWG)(with Pred) | 84.85±0.83 | 66.03 ± 1.36 | 43.89±0.15 | 34.29±0.13 |
| $S_{kde30}$Schürholt et al. (2022a)(cross) | 29.30± 3.46 | 13.25± 1.12 | 15.40± 0.51 | 15.20±1.24 |
| not seen(D2NWG) | 36.64±4.69 | 18.80±0.58 | 18.00±0.22 | 16.21±0.52 |
| not seen(D2NWG)(with Pred) | 30.15±5.09 | 15.76±1.43 | 17.10±1.12 | 15.37±0.52 |

Table 21: In-distribution performance comparison of different image dataset encoding schemes on model zoo dataset

| Datasets | MNIST | SVHN | CIFAR10 | STL10 |
|---|---|---|---|---|
| Pretrained models | 82.82± 1.38 | **67.57± 0.59** | **44.68± 3.15** | **35.99± 1.15** |
| $S_{kde30}$Schürholt et al. (2022a) | 69.73± 5.12 | 50.25± 6.12 | 26.06± 3.01 | 17.20± 3.43 |
| MLP_Encoder | 67.04±17.73 | 35.65 ± 13.03 | 17.41±3.02 | 20.36±7.38 |
| Set_transf(pret) | 78.21±1.76 | 60.90 ± 1.08 | 28.68±1.84 | 34.75±00.38 |
| seen (D2NWG) | 83.92±1.92 | 61.81 ± 3.13 | 43.08±0.55 | 31.45±0.35 |
| seen(D2NWG)(with Pred) | 84.85±0.83 | 66.03 ± 1.36 | 43.89±0.15 | 34.29±0.13 |

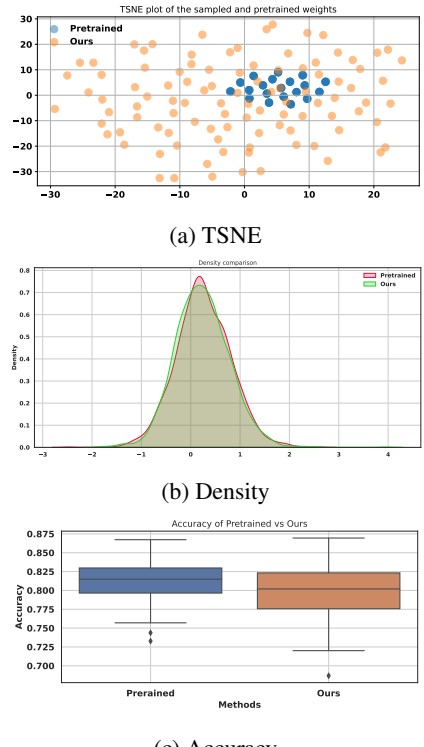

(a) TSNE

(b) Density

(c) Accuracy

Figure 10: **Analysis of relationship between the pretrained weights and the sampled weights for MNIST dataset**

or dataset-conditioned sampling. For example, CIFAR-10 and ImageNet are considered two separate datasets, while SST-2 and CoLA in the GLUE benchmark are treated as two distinct tasks, regardless of differences in the number of classes or subtasks within each dataset or task. Table 25 highlights

Table 22: Performance of the datasets conditional sampling on 10 unseen real-world datasets. We report the averaged accuracy on ten unseen test datasets over 3 different runs fine-tuned for 50 epochs. pret(imnet): pretrained on imagenet1k

| Datasets | No-fine-tuning | | | 50 epochs Fine-Tuning | | | # of classes |
|---|---|---|---|---|---|---|---|
| | Random init. | pret(imnet) | D2NWG(ours) | Random init. | pret(imnet) | D2NWG(ours) | |
| Gemstones | $1.13 \pm 0.52$ | $0.62 \pm 0.00$ | **$1.86 \pm 0.25$** | $70.59 \pm 0.91$ | $67.49 \pm 0.43$ | **$76.06 \pm 0.88$** | 87 |
| Dog Breeds | $0.55 \pm 0.22$ | $0.69 \pm 0.00$ | **$1.87 \pm 0.39$** | $80.78 \pm 0.28$ | $78.13 \pm 0.49$ | **$80.88 \pm 0.88$** | 133 |
| Dessert | $21.03 \pm 2.44$ | $12.50 \pm 0.00$ | **$99.40 \pm 0.02$** | $95.83 \pm 0.34$ | $94.64 \pm 0.00$ | **$99.40 \pm 0.02$** | 5 |
| Colorectal Histology | $11.77 \pm 2.88$ | $11.00 \pm 0.00$ | **$18.12 \pm 0.25$** | $90.34 \pm 0.33$ | $89.75 \pm 0.19$ | **$93.65 \pm 0.10$** | 8 |
| Drawing | $10.86 \pm 1.22$ | $11.00 \pm 0.00$ | **$11.87 \pm 0.93$** | **$90.20 \pm 0.16$** | $90.00 \pm 0.16$ | $89.00 \pm 0.16$ | 10 |
| Alien vs Predator | $51.48 \pm 2.09$ | $28.88 \pm 0.00$ | **$78.15 \pm 0.52$** | $98.52 \pm 0.52$ | **$98.89 \pm 1.42$** | $97.77 \pm 0.00$ | 2 |
| COVID-19 | $20.13 \pm 18.66$ | $46.53 \pm 0.00$ | **$47.22 \pm 0.00$** | $93.86 \pm 0.16$ | $93.40 \pm 0.49$ | **$94.56 \pm 0.71$** | 3 |
| honey-bee-pollen | $49.54 \pm 1.30$ | $50.00 \pm 0.00$ | **$56.94 \pm 4.53$** | $93.05 \pm 0.00$ | $88.89 \pm 0.00$ | **$93.55 \pm 4.53$** | 2 |
| Speed Limit Signs | $30.55 \pm 2.27$ | $25.00 \pm 0.00$ | **$31.48 \pm 10.23$** | $83.33 \pm 0.00$ | $86.11 \pm 0.00$ | **$90.74 \pm 1.31$** | 4 |
| Japanese Characters | $0.03 \pm 0.00$ | $0.08 \pm 0.00$ | **$0.50 \pm 0.22$** | $53.17 \pm 0.15$ | **$62.33 \pm 0.16$** | $62.16 \pm 0.47\ 0.45$ | 1566 |

Table 24: Model components and their configuration modes for llma3.2.1B

| ID | Name | Type | Params | Mode |
|---|---|---|---|---|
| 0 | Model | DiffusionWrapper | 102 M | Train |
| 1 | Model Ema | LitEma | 0 | Train |
| 2 | First stage Model | VAENoDiscModel | 553 M | Eval |
| 3 | Cond Stage Model | IdentityCondStage | 0 | Eval |

that the proposed method supports text and image conditioning, as well as layer- or chunk-wise conditional sampling. D2NWG is one of the first weight generation methods to produce over 800 million parameters in a single instance without tiling. Additionally, it is among the first to effectively explore weight generation across various domains, learning the distribution of combined models pretrained on diverse tasks or datasets.

Table 25: Summary of Experiments for Figures and Tables presented. Min #cls and Max #cls correspond to the minimum and maximum number of classes respectively.

| Object | # Datasets | Min #cls | Max #cls | #Params | Trainset Size | Conditioning |
|---|---|---|---|---|---|---|
| Table 1 | 10 | 1 | 5 | 2565/8005 | 50k | Dataset |
| Table 2 | 5 | 10 | 50 | 25600 | 20k | Dataset |
| Table 3 | 30 | 19 | 706 | 3 M | 30 | Dataset |
| Table 4 | 4 | 10 | 10 | 10853 | 4 | Dataset |
| Table 5 | 6 | 2 | 3 | 0.6M | 6 | Text Description |
| Table 6 | NA | NA | NA | 872M | NA | Chunk Indices |
| Table 7 | NA | NA | NA | 872M | NA | Chunk Indices |
| Table 9 | 2 | 10 | 100 | 0.7M | 2 | Dataset |
| Table 11 | 2 | 10 | 100 | 2048 | 2 | Dataset |
| Table 15 | 3 | 10 | 1000 | 1.4M | 3 | Dataset |
| Table 16 | 3 | 10 | 100 | 11M | 2 | Dataset |
| Table 16 | 4 | 10 | 10 | 2.8M | 4 | Dataset |
| Table 17 | NA | NA | NA | 96M | NA | Chunk Indices |
| Table 18 | NA | NA | NA | 164M | NA | Chunk Indices |
| Figure 3 | 10 | 2 | 1566 | 136468 | 140 | Dataset |
| Figure 2 | 2 | 10 | 100 | 0.47M | 2 | Dataset |
| Figure 6a | 2 | 10 | 10 | 5310 | 2 | Dataset |
| Figure 7 | 2 | 10 | 10 | 5310 | 2 | Dataset |
| Figure 9 | 5 | 10 | 200 | 2.8M | 5 | Dataset |

