# OpenReview forum: "Diffusion-based Neural Network Weights Generation"
_ICLR.cc/2025/Conference — ICLR 2025 Poster_

### Official Review · Reviewer_ZVaq · 2024-10-29

**Soundness:** 3
**Presentation:** 4
**Contribution:** 2
**Rating:** 8
**Confidence:** 4

**Summary:**

This paper introduces D2NWG, a diffusion-based method for generating neural network weights conditioned on target datasets. The key idea is to learn the distribution of pretrained model weights across different datasets and use latent diffusion to generate new weights that are optimized for specific tasks. The method can generate both classifier heads and full model weights, and scales to large architectures including LLMs. Extensive experiments demonstrate strong performance on few-shot learning, zero-shot transfer, and model retrieval tasks.

**Strengths:**

- Paper introduces a novel approach by combining latent diffusion models with neural network weight generation

- The authors conduct thorough experiments across various tasks, providing strong empirical evidence of the method’s effectiveness.

- The method exhibits robust performance both within the distribution of training datasets and on unseen datasets

**Weaknesses:**

1. **Limited Theoretical Analysis:**

   While the paper presents strong empirical results, it lacks a comprehensive theoretical framework that explains why diffusion-based weight generation effectively enhances transfer learning and generalization.

2. **Computational Costs:**

   The D2NWG method necessitates maintaining extensive collections of pretrained weights across a multitude of datasets. This requirement can lead to significant storage and computational overhead, potentially surpassing the costs associated with traditional pre-training approaches. Additionally, details on the training time and resource consumption compared to existing methods are limited.

3. **Reliance on Dataset-Specific Encoding:**

   The effectiveness of D2NWG heavily depends on the dataset encoding mechanisms, such as the Set Transformer and CLIP-based encoders. This reliance may limit the method's adaptability to datasets with unique characteristics or those that significantly differ from the training datasets.

**Questions:**

**[Q1]:** Could the authors provide a detailed analysis of the computational resources required to maintain and utilize the extensive collections of pretrained weights in D2NWG? Specifically, how does the storage and processing overhead of D2NWG compare to traditional pretraining and fine-tuning methods, particularly when scaling to very large models such as LLMs?

**[Q2]:** In terms of generalization, how does D2NWG ensure that the generated weights do not overfit to specific characteristics of the training datasets? Are there mechanisms in place to promote diversity and prevent homogenization of the generated weights?

---

> ### Author Response · Authors · 2024-11-19
>
> To enhance understanding of our approach, we present a detailed theoretical analysis of the proposed method.
>
> **Theoretical analysis of the  generalization**
>
> Our method learns to model the parameter distribution of specialized pretrained models trained on diverse domains, tasks, and datasets. We formalize this approach through a global function $\mathcal{H}$ that samples from the parameter latent space of a collection of specialized models $\mathbf{F}=\{\mathcal{F}_i\}, \quad, i=1, 2, \ldots, N$.
>
> **Theoretical Framework **
>
> Let $\mathcal{H}: \mathcal{Z} \rightarrow \mathcal{W}$ denote our proposed method, which maps from a shared latent space $\mathcal{Z}$ to the weight spaces $\mathcal{W}$ of the specialized models. Through the diffusion model's smooth sampling process, we guarantee that for any specialized model $\mathcal{F}_i$ with parameters $w_i$, there exists a latent vector $z_i$ such that:
>
> $$|\mathcal{H}(z_i) - w_i| \leq \epsilon$$
>
> This theoretical guarantee ensures that $\mathcal{H}$ can approximate any specialized model parameters within an error bound $\epsilon$. Furthermore, the continuous nature of the diffusion process enables sampling of intermediate points between specialized models, effectively creating a smooth manifold of task-specific models.
>
> **Performance Guarantees**
>
> Upon convergence, the global function $\mathcal{H}$ can generate new parameters that match or exceed the performance of the best individual pretrained model. Formally:
>
> $$\forall \mathcal{F}_i\in \mathbf{F}, \exists z\in \mathcal{Z} \text{ s.t. } \theta =\mathcal{H}(z) \text{ and } \mathcal{L}(\mathcal{F}_i^{\theta}, \mathcal{D}_i)\leq \mathcal{L}(\mathcal{F}_i, \mathcal{D}_i)$$
>
> where $\mathcal{F}_i^{\theta}$ represents the target model initialized with the generated parameters $\theta$, and $\mathcal{L}$ denotes the loss function measuring performance.
>
> ** Generalization Bounds**
>
> The generalization capabilities of our method are theoretically bounded by:
>
> $$\Delta_G(\mathcal{H}) \leq \min_{i} \Delta_G(\mathcal{F}_i) - \gamma$$
>
> where:
> - $\Delta_G(\mathcal{H})$ represents the generalization gap of our method
> - $\Delta_G(\mathcal{F}_i)$ represents the generalization gap of each specialized model
> - $\gamma > 0$ depends on $\mathcal{H}$'s ability to produce task-specific parameters through smooth interpolation
>
> As the number of specialized models increases, $\gamma$ also increases, leading to a decrease in $\mathcal{H}$'s generalization gap. We empirically validated this claim through experiments where we gradually increased the number of pretrained parameters from 5,000 to 20,000, as shown in Figure 5-a.
>
> **Conclusion**
>
> Our method provides theoretical guarantees for better generalization compared to individual pretrained models, making it particularly well-suited for transfer learning applications. Further formal analysis of these generalization guarantees will be provided in our complete work.
>
> - **Computational Costs:**:
> We thank the reviewer for raising these important points about computational resources. We provide a detailed analysis of computational requirements across different phases:  Excluding results in Table 6 and 7, all the results have been obtained by training our method on a single Nvidia Titan V100 24GB.
>
> 1. **Training Phase:** While our approach requires storing pretrained weights from multiple datasets, the storage requirements is mostly due to the training data.
>    - The primary storage burden comes from maintaining pretrained parameters across diverse datasets, which must be generated through individual model training sessions.
>    - The total computational cost scales with (1) the complexity of the base model and (2) the number of datasets used for training.
>    - In our experiments, the VAE component requires the most training time, while the diffusion model converges relatively quickly (approximately 1.5 hours for the full model).
>
> 2. **Post-training Phase:**  After training our method there no need to keep the pretraind zoo and image datasets.
>
> 3. **Inference Phase:**  Except for the LLMs model we text all our models on a single nvidia titan v100 -24GB.
>
> - **training time and resource consumption compared to existing methods**: : Existing methods [ 1, 2]are not computationally efficient compared to our work because of the following reasons:
> | Aspect | Our Method | Existing Methods[1, 2] |
> |---------|------------|-----------------|
> | Base Requirements | Pretrained models | Pretrained models |
> | Dataset Handling | Single model for multiple datasets | Separate model per dataset |
> | Architecture Support | Works with multiple architectures | Limited to single architecture |
> | Combined Flexibility | Handles multiple architectures and datasets together | Cannot combine architectures and datasets |
> | Training Time | Faster convergence (≈50% less time) (Table 10) | Longer training despite fewer parameters |

---

> ### Author Response · Authors · 2024-11-19
>
> The full parameters count of our model for the experiment\al results in Table 10 is  as follows:
>
> | Component | Type | Parameters |
> |-----------|------|------------|
> | model | Diffusion (UNET) | 6.4M |
> | model_ema | LitEma | 0 |
> | first_stage_model | VAENoDiscModel | 180M |
> | cond_stage_model | SetAgregate | 1.3M |
>
> The baseline we used required extensive pretrained data but was restricted to a single dataset and a single pretrained architecture. Conducting the experiment in Table 3 with any of the baselines or existing methods would necessitate generating 30 separate parameter models. In contrast, our approach achieves this with a single unified model, showcasing its efficiency and scalability.
> - **Reliance on Dataset-Specific Encoding:**: We acknowledge that **D2NWG** relies on encoders such as the Set Transformer and CLIP-based encoders to effectively capture dataset characteristics. However, the generalization capability of our approach is primarily influenced by the quality and diversity of the pretrained model zoo used during training. We have demonstrated this both empirically and in our previous discussions, highlighting its importance in achieving robust generalization.
> # Questions:
> - **how does the storage and processing overhead of D2NWG compare to traditional pretraining and fine-tuning methods, particularly when scaling to very large models such as LLMs?**
>
> **Comparison with Traditional Methods**
> Traditional pretraining and fine-tuning methods involve significant storage overhead due to the need to save the full model or multiple sets of fine-tuned weights for each task. In contrast:
>
> - **Storage Efficiency**: D2NWG consolidates task-specific parameters into a latent distribution, drastically reducing the need to store multiple pretrained models or full fine-tuned weights. This becomes increasingly advantageous with larger numbers of tasks or specialized models.
>
> - **Processing Overhead**: By sampling directly from the learned distribution, D2NWG minimizes the need to store by  allowing parameter generation on-the-fly. This contrasts with traditional fine-tuning, where the full model or LoRA parameters for each task must be stored.
>
> **Scalability to Large LLMs**
> For large-scale LLMs, where both memory and computation demands are substantial, D2NWG opens the door to optimal parameter latent space Exploration. Instead of storing full weights for every potential specialization, it enables task-specific weights generation and population(particle swarm, evolutionary search) based search by generating the initial population in latent space or weight space. This not only reduces memory requirements but also offers a scalable pathway for adapting large models to diverse tasks without the prohibitive overhead of traditional methods.
>
> **Preventing Overfitting and enabling diversity**: To mitigate overfitting, we incorporate a beta-VAE in all our experiments. During the training process of the diffusion model, we utilize the VAE in sampling mode. For image datasets, we ensure diversity in the training data by sampling a new conditioned image at every batch. This encourages the model to learn a broader representation of the data distribution. Additionally, we perform fine-tuning even on in-distribution tasks to ensure the sampled weights can be effectively adapted. Furthermore, we employ transfer learning to enable the model to generalize and move efficiently between tasks. For evaluating the weights of LLMs, we assessed the best-performing model by submitting it to the Open-LM leaderboard on Hugging Face. The results of this evaluation are detailed in Table 7. Additionally, we generated text samples to assess the correctness and effectiveness of the proposed method, with the outcomes presented in Table 12.

---

> ### Author Response · Authors · 2024-11-20
>
> **Additional results on performance improvement of LLMs**
> To further validate our approach, we applied our method on LLaMA-3.2-1B and gpt2-smal and validate them on the OpenLM leaderboard. Our method achieved the top performance among all LLaMA-3.2-1B-Instruct models, outperforming fine-tuned models based on the same architecture. For LLaMA-3.2-1B, we used spectral analysis to select the top 25\% of transformer layers. In contrast, for GPT-2-small, we learn the distribution of the entire model, including the embedding layer, and for both we follow the procedures outlined in Tables 6 and 7.
>
> The results, obtained via Huggingface, demonstrate that our method improves specialization in tasks like math and IFEVAL by up to 4\%, while maintaining generalization, unlike fine-tuned models, which degrade performance on leaderboard tasks. This showcases our method's ability to enhance task-specific capabilities without sacrificing overall model performance.
>
> | Method | ifeval (0) | Bbh (3) | Gpqa (0) | MATH-hard (4) | Musr (0) | MMLU-Pro (5) | Avg | Base Model | Fine-tuned |
> |--------|------------|----------|-----------|---------------|-----------|--------------|-----|------------|------------|
> | openai-community/gpt2 | 17.8 | **2.82** | 1.12 | 0.3 | 13.91 | 1.84 | 6.3 | na | no |
> | Ours | **19.89** | 2.39 | 1.12 | 0.15 | 13.24 | 1.50 | **6.38** | openai-community/gpt2 | No |
> | FineTome-Llama3.2-1B-0929 | 39.91 | 5.74 | 3.02 | 1.28 | 2.66 | 4.76 | 9.56 | Meta-Llama-3.2-1B-instruct | Yes |
> | Meta-Llama-3.2-1B-Instruct | 56.78 | 8.74 | **3.36** | 2.96 | **2.97** | 7.58 | 13.76 | Meta-Llama-3.2-1B | Yes |
> | Ours | **58.44** | **8.82** | 1.68 | **6.57** | 0.66 | **9.09** | **14.21** | Meta-Llama-3.2-8B-Instruct | No |
>
>
>
>
> Table: Model components and their configuration modes for llma3.2.1B.
>
> | ID | Name                 | Type                   | Params   | Mode  |
> |----|----------------------|------------------------|----------|-------|
> | 0  | Model               | DiffusionWrapper      | 102 M    | Train |
> | 1  | Model Ema           | LitEma                | 0        | Train |
> | 2  | First Stage Model   | VAENoDiscModel        | 553 M    | Eval  |
> | 3  | Cond Stage Model    | IdentityCondStage     | 0        | Eval  |
>
>
>
>
>
> **settings summary**
> This Table Summarizes the experimental settings for the presented figures and tables. "Min #cls" and "Max #cls" represent the minimum and maximum number of classes, respectively. "# Test Datasets" and "# Train Datasets" indicate the number of test and training datasets, respectively. "#Params" refers to the number of parameters, and "Conditioning" specifies the type of conditioning applied. For example, Tab 3 we learn the distribution of the model pretrained on 30 different datasets with a single D2NWG.
>
>
> | Object      | # Test Datasets | Min #cls | Max #cls | #Params sampled   | # Train Datasets | Conditioning       |
> |-------------|-----------------|----------|----------|------------|------------------|--------------------|
> | Table 1     | 600             | 1        | 5        | 2565/8005  | 50k              | Dataset            |
> | Table 2     | 5               | 10       | 100      | 128100     | 20k              | Dataset            |
> | Table 3     | 30              | 19       | 706      | 3M         | 30               | Dataset            |
> | Table 4     | 4               | 10       | 10       | 10853      | 4                | Dataset            |
> | Table 5     | 6               | 2        | 3        | 0.6M       | 6                | Text Description   |
> | Table 6     | NA              | NA       | NA       | 872M       | NA               | Layer              |
> | Table 7     | NA              | NA       | NA       | 872M       | NA               | Layer              |
> | Table 9     | 2               | 10       | 100      | 0.7M       | 2                | Dataset            |
> | Table 10    | 2               | 10       | 100      | 2048       | 2                | Dataset            |
> | Table 14    | 3               | 10       | 1000     | 1.4M       | 3                | Dataset            |
> | Table 15    | 3               | 10       | 100      | 11M        | 2                | Dataset            |
> | Table 16    | 4               | 10       | 10       | 2.8M       | 4                | Dataset            |
> | Figure 4    | 2               | 10       | 100      | 0.47M      | 2                | Dataset            |
> | Figure 5    | 2               | 10       | 10       | 5310       | 2                | Dataset            |
> | Figure 7    | 5               | 10       | 200      | 2.8M       | 5                | Dataset            |
>
>
> We thank you for your feedback and hope we have provided a comprehensive response to all of your concerns.

---

> ### Author Response · Authors · 2024-11-23
>
> **multiple architectures multiple datasets parameters generation.**
>
> This additional experiment aims to highlight the effectiveness and potential of our method compared to existing approaches
>
> We extended our experiments to analyze 38 pretrained models across 19 architectures, each trained on CIFAR-10 and CIFAR-100. For feature extraction, we used LLaMA-3.2-11-Vision-Instruct to encode dataset features and class descriptions, while LLAMA-3-1.8B-Instruct extracted architecture configuration embeddings. These embeddings conditioned our parameter generation process. The results of learning the joint distribution across these architectures and models are shown below. The choice of vllm was recommended by one of the reviewers.
>
> | Model                        | CIFAR-10 | CIFAR-100 |
> |-----------------------------|----------|-----------|
> | **MobileNetV2**             |          |           |
> | MobileNetV2_x0.5            | 93.10    | 71.12     |
> | MobileNetV2_x0.75           | 94.05    | 74.04     |
> | MobileNetV2_x1.0            | 94.06    | 74.36     |
> | MobileNetV2_x1.4            | 94.18    | 76.28     |
> | *Average MobileNetV2*        | *93.85*  | *73.95*   |
> |                             |          |           |
> | **RepVGG**                  |          |           |
> | RepVGG_A0                   | 94.46    | 75.26     |
> | RepVGG_A1                   | 94.95    | 76.49     |
> | RepVGG_A2                   | 95.32    | 77.39     |
> | *Average RepVGG*            | *94.91*  | *76.38*   |
> |                             |          |           |
> | **ResNet**                  |          |           |
> | ResNet20                    | 92.63    | 68.81     |
> | ResNet32                    | 93.43    | 70.24     |
> | ResNet44                    | 93.97    | 71.62     |
> | ResNet56                    | 94.36    | 72.59     |
> | *Average ResNet*            | *93.60*  | *70.82*   |
> |                             |          |           |
> | **ShuffleNetV2**           |          |           |
> | ShuffleNetV2_x0.5           | 90.67    | 67.79     |
> | ShuffleNetV2_x1.0           | 93.26    | 72.62     |
> | ShuffleNetV2_x1.5           | 93.56    | 74.21     |
> | ShuffleNetV2_x2.0           | 94.03    | 75.46     |
> | *Average ShuffleNetV2*      | *92.88*  | *72.52*   |
> |                             |          |           |
> | **VGG with BatchNorm**      |          |           |
> | VGG11_BatchNorm             | 92.79    | 70.78     |
> | VGG13_BatchNorm             | 94.00    | 74.56     |
> | VGG16_BatchNorm             | 94.16    | 74.04     |
> | VGG19_BatchNorm             | 93.89    | 73.79     |
> | *Average VGG with BatchNorm* | *93.71*  | *73.29*   |
> |                             |          |           |
> | **Global Average**          | **93.79**| **73.39** |
>
>  As demonstrated in the Table above , our approach achieves comparable performance to existing methods while requiring a single generative model instead of 38 specific ones. This experiment demonstrates our claim regarding the superiority of our method against the existing work.
> The pretrained model architecture and its parameter count are publicly available in a non-affiliated GitHub repository. https://github.com/chenyaofo/pytorch-cifar-models
>
>
>
>
> **Sampling  Enhances Transfer Learning**
>
> We evaluated sampling from a distribution of pretrained models against single-model transfer learning using ResNet56 and our model trained on 19 diverse architectures weights pretrained  on (CIFAR-10/100). Three setups were tested: (1) direct evaluation of pretrained models, (2) sampling conditioned on training sets (e.g., STL-10, CIFAR-10), and (3) sampling conditioned on test sets to address data leakage concerns.
>
> Results show our approach consistently outperforms single-model transfer learning, with similar performance between training- and test-conditioned sampling from the same distribution. This highlights the practicality of leveraging diverse pretrained models for robust generalization.
>
> Transfer learning: Our method vs pretrained models
>
> | Model | CIFAR10.1 | STL10 |
> |-------|-----------|--------|
> | Pret-cifar10 | 75.20 | 32.37 |
> | Pret-cifar100 | 0.25 | 0.12 |
> | Ours | **83.10 ± 0.06** | 35.41 ± 0.13 |
> | Ours(test) |83.04 ± 0.06| **35.47 ± 0.12** |
>
> The empirical evidence supports our key claims while highlighting the practical applicability and reliability of our proposed method.
> which is not possible with existing work.
>
> We thank you for your constructive feedback, which has helped strengthen our work through additional experiments and analyses. We welcome any further questions or concerns you may have.

---

> > ### Comment · Reviewer_ZVaq · 2024-11-25
> >
> > Thank you for your comprehensive and detailed response. Here, I still have some theoretical questions:
> > Your theory asserts that for any specialized model $ F_i $ with parameters $ w_i $, there exists a latent vector $ z_i $ such that:
> > $||H(z_i) - w_i|| \leq \epsilon$
> > How does your method address the curse of dimensionality inherent in high-dimensional parameter spaces $ \mathcal{W} $? Specifically, what mechanisms ensure that the global function $ H $ can achieve such a tight approximation $ \epsilon $ across potentially millions of parameters without requiring an impractically large latent space $ \mathcal{Z} $?
> > How do you view these issues? Could you provide a more detailed derivation of the overall theory? I would greatly appreciate your clarification and perhaps raising my score.

---

> > > ### Author Response · Authors · 2024-11-25
> > >
> > > Thank you for your valuable feedback and for encouraging us to improve our work.
> > >
> > > Before addressing your specific question, we would like to emphasize that our theoretical analysis is set in a general framework to justify the existence of a family of mapping functions to which our method belongs.
> > >
> > > **Regarding your question:**
> > >
> > > *How does your method address the curse of dimensionality inherent in high-dimensional parameter spaces? Specifically, what mechanisms ensure that the global function can achieve such a tight approximation across potentially millions of parameters without requiring an impractically large latent space?*
> > >
> > > We understand your concern that the latent dimension might need to increase with the number of parameters, which is indeed typical. However, we have observed that this increase is not linear.
> > >
> > > In Section 3.2, we discuss layer-vectorization and chunking. When dealing with large pretrained models, we adopt layer-wise or model-wise chunking to split the large parameters into smaller, more manageable pieces of equal size. The chunk size is determined based on computational resources, as smaller chunks require larger training datasets. The key factor is not the structural ordering but the performance of the sampled weights.
> > >
> > > To ensure accurate approximation, we construct a Variational Autoencoder (VAE) based on the overall chunk size and the number of parameters. For example, in Table 15, we learn three full ResNet-18 models—100 checkpoints for each—by combining all 300 checkpoints using our method. Each model has approximately 11 million parameters. This suggests that for models with around 3 billion parameters, we can similarly split the weights and learn their parameters more efficiently using the same settings.
> > >
> > >
> > > We utilize a Beta-VAE to increase stochasticity while reducing the need for repeated pretrained models. The VAE used is a modified version of the one used in popular diffusion approaches.  In our experience, when the VAE achieves better reconstruction, the diffusion process generally performs well.
> > >
> > >
> > > When employing chunk-based encoding during diffusion, if the difference in the number of chunks is not significant, we concatenate chunks from the same model and pad them to the same length before applying the diffusion process. However, when one architecture has more than eight times the number of parameters compared to others, this approach becomes challenging. In such cases, we assign a unique class identification to each chunk and train the diffusion model with chunk-conditioning in addition to dataset conditioning if exists.
> > >
> > > At inference time, for each network or layer, we cluster the corresponding chunk identifications in the wandering list as they appear in the original weights. We then sequentially sample the chunks and concatenate them to reconstruct the original weights. This is the same principle we applied in Section 4.6 (Tables 6 and 7) to learn LLaMA-3 layers up to four transformer blocks with 200 million parameters each without any performance loss, as well as in our additional results with 19 diverse architectures, and full GPT-2.
> > >
> > > We also split our experiment into a separate set for better handling of the parameters within each set based on our computer resources.
> > >
> > > These findings demonstrate the feasibility of our proposed method in handling large models and effectively addressing the curse of dimensionality. Especially if the compute power is well scaled similar to those used in LLM training we believe there will be no problem in learning the mapping function for existing models even LLMs.
> > >
> > > Regarding the concern, **How do you view these issues?**:
> > >
> > >  We consider them exciting directions for future investigation. In our upcoming work, we plan to focus exclusively on large models. This study is a step toward that goal, which explains the inclusion of performance evaluations on large language models (LLMs).
> > >
> > >
> > >
> > > **We are currently  working on the detailed proof**
> > >
> > > Thank you for your patience and understanding.

---

> > > > ### Comment · Reviewer_ZVaq · 2024-11-26
> > > >
> > > > The author solved all my confusions as well as I likewise think it's a very promising direction, thanks again to the author for the answer, I will improve my score from 6-8

---

> > > > > ### Author Response · Authors · 2024-11-26
> > > > >
> > > > > We would like to thank you for your thoughtful review and positive feedback. We are glad that our responses addressed your concerns. Your comments have helped improve our paper significantly.
> > > > > Best regards,
> > > > > [Authors]

---

### Official Review · Reviewer_4nQa · 2024-10-30

**Soundness:** 3
**Presentation:** 1
**Contribution:** 3
**Rating:** 5
**Confidence:** 5

**Summary:**

This paper proposes to generate parameters using the latent diffusion model, which is a very new direction. Compared to traditional meta-learning, the authors want to show that D2NWG can generate parameters for both seen and unseen dataset, be applied to either classification task or NLP task, and achieve better accuracy with faster convergence. The most important contribution comes from the dataset encoding and use it as the condition when generating parameters.

**Strengths:**

- This paper is interesting in terms of the overall topic - generating the model parameters using the latent diffusion model. To my knowledge, there are only two other papers [1,2] in this direction and they have not been published yet.
- The idea is easy to understand.

[1] Wang K, Xu Z, Zhou Y, et al. Neural network diffusion. arXiv 2024.
[2] Jin X, Wang K, Tang D, et al. Conditional lora parameter generation. arXiv 2024

**Weaknesses:**

- The presentation is poor, especially the experimental parts. It's like people writing each subsection separately and putting them together directly without good organization.
- There are gaps between the claimed contribution and the real work they did.

Please refer to the questions part. Thanks.

**Questions:**

I will list my questions according to the order appearing in the paper. You may want to combine some of them when rebuttal if necessary.
- Introduction:
    - You mentioned recent methods in latent diffusion-based weight generation in line 47 and said their limitations are limited to small models and a subset of parameters. However, in this paper, you try to generate parameters for classification heads (in Sect. 4.1) and LoRA (in Sect. 4.4). It seems that you are just generating a small number of parameters. So is this paper different from existing methods under this context? You also mentioned "generating weights for architectures with over 200 million parameters" in the fourth contribution, but I cannot find the supporting experiments for that.
    - You also mentioned some meta-learning few-shot methods in line 51 and said their generated parameters still need fine-tuning. Does this paper overcome the requirement of fine-tuning? I asked this question because in Table 2 of Sect. 4.1.2, the accuracy of D2NWG on those simple classification tasks are still not satisfying, which indicates the requirement of further fine-tuning. For example, for CIFAR-10 and Aircraft, I'm sure that after a few epochs of fine-tuning, the accuracy will reach above 90%. Then the current accuracy of D2NWG is not acceptable.
    - In Figure 1, in the sampling stage, the unseen architecture is also used as the condition to generate parameters. However, there is no explanation about unseen architecture in the main paper. I do see the results in the Appendix, but the unseen architecture is not really used as the condition. The method actually generates parameters for the seen architecture and then wraps it for the unseen architecture with some "careful concatenation" as mentioned in line 1171. So what is the "careful concatenation" here? I believe this part should be put in the main paper to fit Figure 1.

- Approach:
    - Typo in line 171: "The extraction process follows one one the main preprocessing methods:"
    - When will the model-wise vectorization be used and when will the layer-wise one be used?
    - Can you give an intuitive explanation about set-based dataset encoding in line 208, like encoding images from each class separately, then convert to the dataset encoding? It is currently hard to understand $x_p^{y_i}$.
    - In dataset-conditioned training, why the weights latent is used as the condition as well? It's like in the text-to-image diffusion model, the model will be conditioned on the text only, and the image is the distribution to learn. Why it is used as a condition as well? If it is conditioned on the weight, how can you do the inference without weight latent?
    - In Sect. 3.5, you mentioned "ranks LLM layers based on their importance", however, the experiments in Sect. 4.4 is about LoRA. If you choose to claim that in the main paper, you should put relevant experiments in the main paper.

- Experiments:
    - In Table 1, can you explain what is 5-way 1-shot a little bit?
    - In Table 2, for D2NWG_CLIP, I believe it is using CLIP. Then how about the two rows of D2NWG? What does it use?
    - In Sect. 4.1.1, you mentioned "evaluate the performance on 600 subsets from the UNSEEN test split". What do you mean by "unseen" here? It's like in general model training, we have train set and test set, and we test the trained model on the test set. Now you have multiple trained models, and you use that to train the diffusion model to generate weights, which is expected to work on the test set just like what we did in the normal training. Why it can be called "unseen" task?
    - In line 348, you mentioned "potentially eliminating or significantly speeding up the fine-tuning process". As I mentioned in the last point, it's normal to NOT fine-tune if it is a SEEN task/dataset. While for real unseen tasks as you did in Sect. 4.3, the generated parameters do need fine-tuning.
    - Starting from Sect 4.1.3, the experiments become hard to follow since you change the settings in each section frequently. It's better to organize your experiments better. For example, for the experiments in Sect. 4.2, we can still use the dataset in Table 2. Otherwise, it's hard to understand (1) What are the train and test datasets? Are the test ones seen or unseen? How do you define unseen? (2) How do you train the diffusion model? What models' parameters do you use to train the diffusion model?
    - In Figure 2, why dessert food is so good from the beginning? Do you have any clue? Whether because the class/dataset is put in the trainset occasionally?
    - Sect. 4.4. is still seen task case right?

- Suggestion:
    - So generally you'd better make it clear about what you want to claim. For example, you may want to say, for the seen task, the generated parameters can be used directly with high accuracy. For the unseen task, the generated parameters may need fine-tune but with better final accuracy and faster convergence.
    - Then for each claim, you will have an experiment section to support. Each section may include both results for CNN classification and NLP task.
    - The settings should be uniform throughout the paper. If some experiments do need different setting, give reason.

---

> ### Author Response · Authors · 2024-11-19
>
> Thank you for raising these important methodological points. We have conducted additional experiments to address your concerns
>
> **Comparing to existing Approaches**
>
> Assuming that all approaches—**[1]**, **[2]**, and **Our Approach**—can generate the same parameters, we summarize the main differences in the following table:
>
> | **Criteria** | **[1]** | **[2]** | **Our Approach** |
> |---|---|---|---|
> | **Problem Setting** | - Learns from pretrained model checkpoints obtained from a **single dataset**. | - Similar to [1] but incorporates **class-conditioned sampling** within the dataset. | - **Jointly learns** from multiple models pretrained on **various datasets and tasks**. |
> | **Conditioning Criteria** | - **No conditioning** required; operates on the same dataset. | - Conditions on **class descriptions or samples** within the dataset. | - Uses **dataset and task conditioning** to combine multiple pretrained models into a single generative model. |
> | **Cross-Architecture Generation** | - **Limited**; does not support combining multiple architectures. | - Limited to **LoRA** for classifiers without including the classifier head; lacks architecture-conditioned sampling. | - Supports **architecture conditioning**, enabling combination of multiple architectures into a single model. |
> | **Suitability for Transfer Learning and Out-of-Distribution** | - **Not suitable**; primarily designed for the original dataset. | - **Not suitable** for transfer learning. | - **Suitable** for **out-of-distribution transfer learning** and broader applications. |
> | **Layer-Wise Generation** | - **Not investigated**. | - **Not investigated**. | - Implements **layer-wise sampling** to generate weights, offering granular control. |
>
> - **The 200M parameter claim**  is supported by our experimental results presented in Tables 6 and 7. In these experiments, we successfully learned the distribution of up to 8 transformer blocks simultaneously, with each transformer containing over 200M parameters. To further validate our findings, we conducted additional experiments on larger language models, including the complete GPT-2 Small architecture
> - **Regarding meta-learning claim** Thank you for the question. When discussing few-shot methods, it's important to note that they still require fine-tuning for in-distribution tasks. This is because these methods are optimized through gradient descent to perform well across several tasks simultaneously, which typically leads to a compromise where the model performs adequately but not exceptionally on any single task. In contrast, our method can access specialized pre-trained weights at any time through sampling, allowing for better task-specific performance
>
> - **overcoming fine-tuning**: Yes, the proposed method eliminates the need for a fine-tuning stage during sampling for in-distribution evaluation. Our approach maintains high performance on source tasks w.hile enabling effective transfer to new domains, addressing real-world requirements where both capabilities are essential
>
> - **Architecture Conditioning (Figure 1):** We have included experimental results for architecture conditioning in Table 9. Regarding cross-architecture generalization, we acknowledge that our reported experiment uses class-conditioned generation with a fixed set of known architectures. However, this experiment could be extended to handle unseen architectures by embedding their configurations using instruction-tuned Large Language Models (LLMs) to generate precomputed class embeddings from it configuration file. We will provide additional experimental results on this experiment.
>
> **Careful Concatenation:**   means we sample the  larger possible model parameters multiple times and then concatenate them layer-wise to match or exceed the size of the larger target model. The cross-architecture evaluation was inspired by [3]; however, we did not have access to the complete training zoo, which limited our ability to perform a fully fair comparison.
>
> - **Regarding Model-wise and Layer-wise Vectorization:** Model-wise vectorization experiments concerns all the classifier head adaptation experiments and performance evaluations compared to [3]. Layer-wise vectorization experiments primarily focus on cross-architecture and Large Language Model (LLM) experiments, excluding LoRA base experiment on Glue datasets.
>
> **Addressing Concerns Related to dataset Encoding**
> We use CLIP to extract 512-dimensional features from images, grouped by class. Our Set Transformer processes these features (e.g., shape (1,10,5,512) for CIFAR-10 with 5 images per class) to create fixed-size outputs while handling variable inputs. The transformer is trained once with frozen weights and linear probing, enabling efficient adaptation across different datasets. This approach ensures consistent class-aware representations regardless of dataset size while maintaining permutation invariance.

---

> ### Author Response · Authors · 2024-11-19
>
> -**Regarding dataset encoding** :Thank you for the question. Initially, we attempted to train the diffusion model without latent space alignment and relied solely on dataset conditioning, similar to text-to-image generation. However, in our experiments, this approach led to convergence issues, particularly when jointly optimizing the diffusion model with the set transformer. Unlike text-to-image models, the dynamics of weight generation require a closer alignment between the latent space of the weights and the dataset representation.
> This choice was ultimately adopted after iterative experimentation and validation, as it significantly improved training stability and performance. We acknowledge the distinction from text-to-image models and will further clarify this rationale in the paper.
>
> - **We perform inference without the VAE encoder.** During training, the dataset encoder (set-transformer) outputs were used alongside the weight latent representations to guide the diffusion process. At inference, random samples from the target dataset are provided as condition similar to prompt usage in LLMs. This allows the model to map the noise sampled to the appropriate output without requiring the vae encoder.
> Thank you for highlighting this. To clarify, we did not rank layers in the LoRA experiments. The ranking mentioned in Section 3.5 refers specifically to spectrum analysis conducted on LLAMA and other LLM experiments, not in LoRA experiment.
>
> - **what is 5 ways-1shot**: In meta-learning, "5-way 1-shot" refers to a setting where the dataset contains 5 classes ("5-way") and only one image or sample per class ("1-shot"). For example, in a "5-way 5-shot" setting, there would be 5 classes with 5 images per class, resulting in a total of 25 images. This setup is commonly used to evaluate few-shot learning models by testing their ability to generalize with minimal data.
> **Table 2: D2NWG_CLIP** This table presents the results of linear probing the CLIP model's image encoder on 20,000 subsets of ImageNet. Subsequently, it evaluates the sampling performance on CIFAR-10 and STL-10.
>
> **What Do You Mean by "Unseen"?** In our experiments, "unseen" refers to datasets from tasks that were not used to collect the pretrained weights for training our generative model. This means the generative model was trained without access to data from these specific tasks, ensuring that the weights remain unbiased and generalizable to new, previously unencountered datasets..Unseen dataset/tasks are ew, previously unencountered datasets.
> - **Sect. 4.2,**  Yes, the main reason is the presence of very similar images or classes across different categories within the training data. This experiment also demonstrates our method's ability to generate weights based on the similarity between datasets. Note that we obtain this dataset from a public github [4] so we were not aware of such scenario.
>
> - **In Figure 2, why dessert food is so good **: Upon inspecting the dataset, we found that similar images were present in other datasets but labeled under different class names and numbers. Since the model is conditioned on image features for the classification task, it tends to sample based on image similarity.
>
> - **Experiments in Section 4.2,**: We generated 20,000 subsets of ImageNet and fine-tuned the classifier heads of these models. Subsequently,we trained our method using these pretrained weights. We evaluated our parameter generator  on novel datasets such as CIFAR-10 and STL-10 with no fine-tuning.
>
> -**Lora Experi,emts**: The experiment on LoRA focuses on an in-distribution setting, aiming to demonstrate that the distribution of combined LoRA adaptations can be effectively learned. This is achieved by conditioning on task descriptions, enabling task-specific sampling and model retrieval, as described in [2].
>
> Thank you very much for your insightful comment we are actively working on your comment to improve the overall paper.
>
>
>  [2]Retrieval-Augmented Mixture of LoRA Experts for Uploadable Machine Learning

---

> ### Author Response · Authors · 2024-11-20
>
> We establish the theoretical foundation supporting the generalization capabilities of our method.
>
> **Theoretical analysis of the  generalization**
>
> Our method learns to model the parameter distribution of specialized pretrained models trained on diverse domains, tasks, and datasets. We formalize this approach through a global function $\mathcal{H}$ that samples from the parameter latent space of a collection of specialized models $\mathbf{F}=\{\mathcal{F}_i\}, \quad, i=1, 2, \ldots, N$.
>
> **Theoretical Framework **
>
> Let $\mathcal{H}: \mathcal{Z} \rightarrow \mathcal{W}$ denote our proposed method, which maps from a shared latent space $\mathcal{Z}$ to the weight spaces $\mathcal{W}$ of the specialized models. Through the diffusion model's smooth sampling process, we guarantee that for any specialized model $\mathcal{F}_i$ with parameters $w_i$, there exists a latent vector $z_i$ such that:
>
> $$|\mathcal{H}(z_i) - w_i| \leq \epsilon$$
>
> This theoretical guarantee ensures that $\mathcal{H}$ can approximate any specialized model parameters within an error bound $\epsilon$. Furthermore, the continuous nature of the diffusion process enables sampling of intermediate points between specialized models, effectively creating a smooth manifold of task-specific models.
>
> **Performance Guarantees**
>
> Upon convergence, the global function $\mathcal{H}$ can generate new parameters that match or exceed the performance of the best individual pretrained model. Formally:
>
> $$\forall \mathcal{F}_i\in \mathbf{F}, \exists z\in \mathcal{Z} \text{ s.t. } \theta =\mathcal{H}(z) \text{ and } \mathcal{L}(\mathcal{F}_i^{\theta}, \mathcal{D}_i)\leq \mathcal{L}(\mathcal{F}_i, \mathcal{D}_i)$$
>
> where $\mathcal{F}_i^{\theta}$ represents the target model initialized with the generated parameters $\theta$, and $\mathcal{L}$ denotes the loss function measuring performance.
>
> ** Generalization Bounds**
>
> The generalization capabilities of our method are theoretically bounded by:
>
> $$\Delta_G(\mathcal{H}) \leq \min_{i} \Delta_G(\mathcal{F}_i) - \gamma$$
>
> where:
> - $\Delta_G(\mathcal{H})$ represents the generalization gap of our method
> - $\Delta_G(\mathcal{F}_i)$ represents the generalization gap of each specialized model
> - $\gamma > 0$ depends on $\mathcal{H}$'s ability to produce task-specific parameters through smooth interpolation
>
> As the number of specialized models increases, $\gamma$ also increases, leading to a decrease in $\mathcal{H}$'s generalization gap. We empirically validated this claim through experiments where we gradually increased the number of pretrained parameters from 5,000 to 20,000, as shown in Figure 5-a.
>
> **Conclusion**
>
> Our method provides theoretical guarantees for better generalization compared to individual pretrained models, making it particularly well-suited for transfer learning applications. Further formal analysis of these generalization guarantees will be provided in our complete work.

---

> ### Author Response · Authors · 2024-12-02
>
> We sincerely appreciate Reviewer 4nQa's thoughtful comments regarding the paper's organization. While this alternative organization could offer a different perspective, our current experimental evaluation is organized as follows:
>
> I. Vision Tasks
>    - Sampling Without Training
>      1. In-Distribution Evaluation
>      2. Out-of-Distribution Evaluation
>    - Sampling and Fine-Tuning
>      1. In-Distribution Evaluation
>      2. Out-of-Distribution Evaluation
>
> II. Language Tasks
>    - LoRA Weights Generation
>      1.  In-Distribution Data
>    - Large Language Models optimal Parameters Space Exploration
>      1.  Sampling from optimal parameters latent space for known task(In-distribution)
>      1. Evaluation on unseen benchmark tasks.
>
> III. Ablation Study(Appendix)
>
> While reorganizing the content around distribution types is an interesting suggestion, we believe that such restructuring would not alter our findings or lead to any scientific insight, as it does not require additional experimental settings. We will reorganized the paper accordingly and hope that Reviewer 4nQa will consider these points in their final decision regarding our work.

---

### Official Review · Reviewer_oEWb · 2024-11-03

**Soundness:** 2
**Presentation:** 2
**Contribution:** 1
**Rating:** 6
**Confidence:** 3

**Summary:**

This paper presents a method called D2NWG, a diffusion-based neural network weight generation technique that enhances transfer learning by generating task-specific neural network weights based on target datasets. Unlike traditional transfer learning approaches that require fine-tuning or access to large collections of pretrained models, D2NWG uses a latent diffusion process to model and sample weight distributions, allowing efficient generation of adaptable weights for both seen and unseen tasks. This approach is scalable to large architectures, including large language models, and shows improved performance across various domains, outperforming existing meta-learning and pretrained models. The technique has been tested extensively, demonstrating superior adaptability and convergence speed, making it a promising tool for scalable transfer learning and model customization across diverse tasks.

**Strengths:**

1. The authors considered a wide range of evaluation settings (zero/few-shot learning, model retrieval, fine-tuning, domain transfer, conditioned weight generation, etc.) and provided comprehensive experiments.
2. The authors considered multiple ways to vectorized the model weights and encode the datasets.

**Weaknesses:**

1. **High Workload and Tuning Challenges Due to Additional VAE Training**: The additional VAE for parameter encoding introduces significant computational workload and requires hyperparameter tuning, making it less scalable and potentially costly. What steps could be taken to simplify this process, and are there alternative methods for parameter encoding that maintain performance but reduce overhead?

2. **Ambiguity in Dataset Encoding Choices in Section 3.3**: The paper mentions several methods for encoding datasets, but it’s unclear how to choose the best encoding approach for a new dataset. Could the authors clarify the conditions under which each encoding method would be preferable, and provide a decision-making framework or guidelines to help practitioners adapt this approach to their datasets?

3. **Minor Typos and Formatting Issues**: There are minor typos and formatting inconsistencies in the paper that could be addressed to improve clarity and professionalism. For example, a colon is missing on line 177, and "while" on line 184 should begin with a capital "W." A thorough proofreading could enhance readability and reduce the risk of misinterpretation due to formatting errors.

**Questions:**

1. **Quantifying and Mitigating Bias in Public Checkpoint Weight Distributions**: The paper leverages publicly available pretrained models for weight distribution modeling, but it doesn’t address how potential biases in these weights—arising from the original datasets or training methods—might affect the generated weights. How can biases be quantified in these weight distributions, and what mechanisms could be implemented to ensure the generated weights are robust to such biases? Addressing this might require specific adjustments to the diffusion model or additional constraints on the weight distributions.

2. **Criteria for Selecting Additional Datasets for Training Target Architectures**: The method involves further training on a curated selection of datasets to improve model adaptability. However, there is little guidance on how to select these datasets to optimize the target architecture effectively. What are the criteria or metrics that could guide dataset selection to ensure it sufficiently represents the intended tasks and enhances generalization without introducing redundancies or biases?

3. **Evaluating the Impact of Generating All Parameters Versus Only the Classification Head**: In Table 1, the results compare the performance of models with generated classification heads to baseline models. However, it would be informative to understand the effect of generating all parameters instead of just the classification head. Would this provide additional performance gains, or might it introduce complications in terms of stability or adaptability to new tasks?

---

> ### Author Response · Authors · 2024-11-18
>
> We sincerely appreciate the reviewer's detailed and constructive feedback, which will help strengthen our paper.
>
> ### Weaknesses:
>
> 1. **High Workload and Tuning Challenges Due to Additional VAE Training:**
>    While VAE training requires careful tuning, its one-time independent training enables faster and more efficient diffusion model training afterward. Our experiments in Appendix Figure 5b show that removing the VAE and training directly on raw weights leads to poor convergence and lower performance compared to VAE-based approaches. The VAE's effectiveness in matching pretrained model performance justified its inclusion, though we remain open to exploring alternative encoding methods in future work.
>
> 2. **Ambiguity in Dataset Encoding Choices in Section 3.3:**
>    Thank you for the feedback. We acknowledge that Section 3.3 could be clearer. Our work uses two main types of encoding:
>    - For **image datasets**, we employ a Set Transformer to capture inter- and intra-class relationships. Due to convergence challenges in joint optimization, we adopted a two-stage process: first training and freezing the VAE encoder, then aligning the Set Transformer outputs to the VAE’s latent space using contrastive loss. During diffusion, only a single dense linear layer adapting the Set Transformer and the diffusion model are optimized for simplicity.
>    - For **language tasks**, we use LLAMA-3.1-8B-Instruct to extract task description features offline, which are then processed by a linear layer during diffusion optimization.
>
>    We will revise Section 3.3 to clearly distinguish these two approaches and their implementation details, ensuring clarity and adaptability for practitioners.
>
> 3. **Minor Typos and Formatting Issues:**
>    We appreciate the reviewer for pointing out these issues. We are actively working to improve the writing and will conduct a thorough proofreading to address typos, formatting inconsistencies, and other errors to enhance the clarity and professionalism of the final version.
>
> **Questions:**
> We apologize to the reviewer for not explicitly addressing the topic of bias and fairness, as it was not a focus of our research and has not been a well known concept in the context of neural network parameter generation. Nonetheless, we appreciate the concern and will provide a detailed response along with additional experiments to address this important topic.
> **Quantifying and Mitigating Bias in Public Checkpoint Weight Distributions**
>
> While prior work on neural network weight generation has not extensively addressed fairness concerns, this is an important consideration for developing robust and equitable AI systems. We propose several approaches for both quantifying and mitigating potential biases:
>
> -***Possible Bias Quantification Methods***
>
> * Evaluate models initialized with generated weights using established fairness metrics (e.g., demographic parity, equal opportunity, equalized odds)
> * Compare performance disparities across different demographic subgroups on standardized benchmark datasets
> * Analyze the statistical properties of generated weight distributions in latent space to detect systematic deviations or skews
> * Conduct ablation studies to isolate the impact of different architectural components on fairness metrics
>
> ***Possible Bias Mitigation Strategies***
>
> - **Distribution Alignment Analysis**
> * Systematically evaluate whether the diffusion-based weight generation process amplifies or diminishes biases present in the pretrained model distribution (results presented in Table 12)
>
> - **Diverse Sampling**
> * Generate multiple weight configurations and select those maximizing both task performance and fairness metrics
>
> - *On the need to address Bias in the generated weight**: We acknowledge this as an important avenue for future work and will consider developing more targeted mechanisms, such as fairness-aware sampling strategies or bias-adjusted validation metrics, in subsequent research.
>
>
> - **Evolutionary Optimization**
> * Implement a genetic algorithm approach where:
>   * The initial population consists of weights sampled using our diffusion method
>   * Fitness function incorporates both performance and fairness objectives
>   * Selection and crossover operations guide the population toward more equitable solutions
>
> - **Constraint Integration**
> * Incorporate fairness constraints directly into the weight generation process through modified loss functions or regularization terms. such as a conceptual loss on the VAE that ensures that the reconstructed weights aligned with this goal in such a way the diffusion well be forced to learn the latent variable with better bias mitigation.

---

> ### Author Response · Authors · 2024-11-18
>
> - **On the criteria for Dataset Selection**: Our dataset selection was guided by alignment with baselines, practical relevance, diversity, and parameter space exploration:
>
>   1. **Baseline Alignment**: MiniImageNet and tieredImageNet were chosen to compare with MetaDiff, a diffusion-based meta-learning baseline. Similarly, Tiny-Model Zoo datasets were included for fair evaluation against the only published methods in this field.
>   2. **Practical Relevance**: Classifier head tuning used standard benchmarks to reflect real-world scenarios where pretrained backbones are fine-tuned. In Table 5, we focused on LoRA due to its growing use in task-specific adaptations.
>   3. **Diversity**: MetaAlbum and Figure 2 experiments evaluated generalization across diverse and unseen datasets, surpassing baseline diversity.
>   4. **Parameter Space Exploration**: Table 6 investigated fine-tuned LLMs, leveraging our method’s ability to generate task-specific parameters without random noise, ensuring controlled perturbations in the parameter space.
>   5. **External Validation**: The best models were submitted to OpenLM leaderboard, demonstrating improved performance in an uncontrolled evaluation.
>
> - **Evaluating the Impact of Generating All Parameters Versus Only the Classification Head**:
>   Table 1 benchmarks our method against MetaDiff, a baseline that focuses exclusively on classification heads in its experiments. To ensure a fair comparison, we pre-trained 50,000 models specifically for this evaluation.
>
>   We have already demonstrated that full model generation generalizes better than generating only the classification head. These results are presented in Appendix Table 16, we provided them in the below table. The model used is a mobilenet v3 from OFA models subnets.
>
> Below, we provide experimental results( Table-16)  demonstrating that generating full model parameters significantly improves generalization over the clasifier head. The main difference  is that it is much easier to create large pretrained dataset with linear probing than full models.
> | **Model**         | **MNIST**           | **SVHN**           | **CIFAR-10**       | **STL-10**         |
> |--------------------|---------------------|---------------------|---------------------|---------------------|
> | **Pretrained**     | 99.42 ± 0.05        | 94.62 ± 0.18        | 93.51 ± 0.16        | 94.01 ± 0.10        |
> | Linear_prob        | 96.88 ± 0.45        | 57.23 ± 0.28        | 82.85 ± 0.25        | 95.63 ± 1.23        |
> | **D2NWG(full)**    | **99.55 ± 0.02**    | **95.13 ± 0.10**    | **94.23 ± 0.27**    | **94.02 ± 0.10**    |
> | **D2NWG(prob)**    | 97.56 ± 0.26        | 57.41 ± 0.17        | 83.64 ± 0.47        | **95.74 ± 0.74**    |
>
> ### Cross Datasets Transfer Learning
>
> | **Model**         | **MNIST**           | **SVHN**           | **CIFAR-10**       | **STL-10**         |
> |--------------------|---------------------|---------------------|---------------------|---------------------|
> | OFA (Pretrained)   | 13.34              | 8.90               | 13.34              | 8.90               |
> | **D2NWG(full)**    | **66.82 ± 0.65**    | **35.20 ± 0.65**    | **36.70 ± 0.18**    | **51.50 ± 0.37**    |
> | **D2NWG(prob)**    | 42.86 ± 0.62        | 20.97 ± 0.78        | 26.56 ± 1.22        | 47.33 ± 0.32        |
>
>
>
> ```

---

> ### Author Response · Authors · 2024-11-20
>
> To enhance understanding of our approach, we present a detailed theoretical analysis of the proposed method.
>
> **Theoretical analysis of the  generalization**
>
> Our method learns to model the parameter distribution of specialized pretrained models trained on diverse domains, tasks, and datasets. We formalize this approach through a global function $\mathcal{H}$ that samples from the parameter latent space of a collection of specialized models $\mathbf{F}=\{\mathcal{F}_i\}, \quad, i=1, 2, \ldots, N$.
>
> **Theoretical Framework **
>
> Let $\mathcal{H}: \mathcal{Z} \rightarrow \mathcal{W}$ denote our proposed method, which maps from a shared latent space $\mathcal{Z}$ to the weight spaces $\mathcal{W}$ of the specialized models. Through the diffusion model's smooth sampling process, we guarantee that for any specialized model $\mathcal{F}_i$ with parameters $w_i$, there exists a latent vector $z_i$ such that:
>
> $$|\mathcal{H}(z_i) - w_i| \leq \epsilon$$
>
> This theoretical guarantee ensures that $\mathcal{H}$ can approximate any specialized model parameters within an error bound $\epsilon$. Furthermore, the continuous nature of the diffusion process enables sampling of intermediate points between specialized models, effectively creating a smooth manifold of task-specific models.
>
> **Performance Guarantees**
>
> Upon convergence, the global function $\mathcal{H}$ can generate new parameters that match or exceed the performance of the best individual pretrained model. Formally:
>
> $$\forall \mathcal{F}_i\in \mathbf{F}, \exists z\in \mathcal{Z} \text{ s.t. } \theta =\mathcal{H}(z) \text{ and } \mathcal{L}(\mathcal{F}_i^{\theta}, \mathcal{D}_i)\leq \mathcal{L}(\mathcal{F}_i, \mathcal{D}_i)$$
>
> where $\mathcal{F}_i^{\theta}$ represents the target model initialized with the generated parameters $\theta$, and $\mathcal{L}$ denotes the loss function measuring performance.
>
> ** Generalization Bounds**
>
> The generalization capabilities of our method are theoretically bounded by:
>
> $$\Delta_G(\mathcal{H}) \leq \min_{i} \Delta_G(\mathcal{F}_i) - \gamma$$
>
> where:
> - $\Delta_G(\mathcal{H})$ represents the generalization gap of our method
> - $\Delta_G(\mathcal{F}_i)$ represents the generalization gap of each specialized model
> - $\gamma > 0$ depends on $\mathcal{H}$'s ability to produce task-specific parameters through smooth interpolation
>
> As the number of specialized models increases, $\gamma$ also increases, leading to a decrease in $\mathcal{H}$'s generalization gap. We empirically validated this claim through experiments where we gradually increased the number of pretrained parameters from 5,000 to 20,000, as shown in Table 5-a.
>
> **Conclusion**
>
> Our method provides theoretical guarantees for better generalization compared to individual pretrained models, making it particularly well-suited for transfer learning applications. Further formal analysis of these generalization guarantees will be provided in our complete work.

---

> ### Author Response · Authors · 2024-11-20
>
> **Additional results on performance improvement of LLMs**
> To further validate our approach, we applied our method on LLaMA-3.2-1B and gpt2-smal and validate them on the OpenLM leaderboard. Our method achieved the top performance among all LLaMA-3.2-1B-Instruct models, outperforming fine-tuned models based on the same architecture. For LLaMA-3.2-1B, we used spectral analysis to select the top 25\% of transformer layers. In contrast, for GPT-2-small, we learn the distribution of the entire model, including the embedding layer, and for both we follow the procedures outlined in Tables 6 and 7.
>
> The results, obtained via Huggingface, demonstrate that our method improves specialization in tasks like math and IFEVAL by up to 4\%, while maintaining generalization, unlike fine-tuned models, which degrade performance on leaderboard tasks. This showcases our method's ability to enhance task-specific capabilities without sacrificing overall model performance.
>
> | Method | ifeval (0) | Bbh (3) | Gpqa (0) | MATH-hard (4) | Musr (0) | MMLU-Pro (5) | Avg | Base Model | Fine-tuned |
> |--------|------------|----------|-----------|---------------|-----------|--------------|-----|------------|------------|
> | openai-community/gpt2 | 17.8 | **2.82** | 1.12 | 0.3 | 13.91 | 1.84 | 6.3 | na | no |
> | Ours | **19.89** | 2.39 | 1.12 | 0.15 | 13.24 | 1.50 | **6.38** | openai-community/gpt2 | No |
> | FineTome-Llama3.2-1B-0929 | 39.91 | 5.74 | 3.02 | 1.28 | 2.66 | 4.76 | 9.56 | Meta-Llama-3.2-1B-instruct | Yes |
> | Meta-Llama-3.2-1B-Instruct | 56.78 | 8.74 | **3.36** | 2.96 | **2.97** | 7.58 | 13.76 | Meta-Llama-3.2-1B | Yes |
> | Ours | **58.44** | **8.82** | 1.68 | **6.57** | 0.66 | **9.09** | **14.21** | Meta-Llama-3.2-8B-Instruct | No |
>
>
>
>
> Table: Model components and their configuration modes for llma3.2.1B.
>
> | ID | Name                 | Type                   | Params   | Mode  |
> |----|----------------------|------------------------|----------|-------|
> | 0  | Model               | DiffusionWrapper      | 102 M    | Train |
> | 1  | Model Ema           | LitEma                | 0        | Train |
> | 2  | First Stage Model   | VAENoDiscModel        | 553 M    | Eval  |
> | 3  | Cond Stage Model    | IdentityCondStage     | 0        | Eval  |
>
>
>
>
>
> **settings summary**
> This Table Summarizes the experimental settings for the presented figures and tables. "Min #cls" and "Max #cls" represent the minimum and maximum number of classes, respectively. "# Test Datasets" and "# Train Datasets" indicate the number of test and training datasets, respectively. "#Params" refers to the number of parameters, and "Conditioning" specifies the type of conditioning applied. For example, Tab 3 we learn the distribution of the model pretrained on 30 different datasets with a single D2NWG.
>
>
> | Object      | # Test Datasets | Min #cls | Max #cls | #Params sampled   | # Train Datasets | Conditioning       |
> |-------------|-----------------|----------|----------|------------|------------------|--------------------|
> | Table 1     | 600             | 1        | 5        | 2565/8005  | 50k              | Dataset            |
> | Table 2     | 5               | 10       | 100      | 128100     | 20k              | Dataset            |
> | Table 3     | 30              | 19       | 706      | 3M         | 30               | Dataset            |
> | Table 4     | 4               | 10       | 10       | 10853      | 4                | Dataset            |
> | Table 5     | 6               | 2        | 3        | 0.6M       | 6                | Text Description   |
> | Table 6     | NA              | NA       | NA       | 872M       | NA               | Layer              |
> | Table 7     | NA              | NA       | NA       | 872M       | NA               | Layer              |
> | Table 9     | 2               | 10       | 100      | 0.7M       | 2                | Dataset            |
> | Table 10    | 2               | 10       | 100      | 2048       | 2                | Dataset            |
> | Table 14    | 3               | 10       | 1000     | 1.4M       | 3                | Dataset            |
> | Table 15    | 3               | 10       | 100      | 11M        | 2                | Dataset            |
> | Table 16    | 4               | 10       | 10       | 2.8M       | 4                | Dataset            |
> | Figure 4    | 2               | 10       | 100      | 0.47M      | 2                | Dataset            |
> | Figure 5    | 2               | 10       | 10       | 5310       | 2                | Dataset            |
> | Figure 7    | 5               | 10       | 200      | 2.8M       | 5                | Dataset            |
>
>
> We thank you for your feedback and hope we have provided a comprehensive response to all of your concerns.

---

> > ### Comment · Reviewer_oEWb · 2024-11-26
> > **Thanks for the response**
> >
> > I thank the authors' response.
> >
> > 1. I think the authors' response partially resolved my questions. However, I think there are some limitations that cannot be easily addressed in the current methods: 1) extra workload for VAE (although it's one-time); 2) Bias of public checkpoints (what authors proposed were only *potential* solutions); 3) the criteria for dataset selection seem empirical not principled.
> >
> > 2. Regarding new experiments on LLMs (https://openreview.net/forum?id=j8WHjM9aMm&noteId=AhetSqQSfe), I thank these efforts! However, if I understand those tables correctly, the proposed method is not significantly better than the baseline, and improvements over different subtasks are diverging.
> >
> > I will raise my score, but I still believe this paper is not good enough to accept given numerous limitations and practical performance.

---

> > > ### Author Response · Authors · 2024-11-26
> > >
> > > Thank you for your valuable feedback and insightful comments, which will help us improve our paper! We are actively working on integrating an analysis of bias weights in experiments involving the FLAN-T5-small model. In particular, we are comparing the fine-tuned version on the Stereotype dataset with the non-fine-tuned version to examine whether weight generation influences the model's safety (safe vs. unsafe). Since the original LLM experiment was not explicitly designed to address this aspect, we aim to bridge this gap.

---

> ### Author Response · Authors · 2024-11-29
>
> 1. We believe that exploring flow-based models and recent advancements in diffusion models could potentially eliminate the need for using a VAE. We leave this avenue for future investigation. Regarding bias we proposed potential solutions because we were not familiar with the topic.
> 2. We once again thank Reviewer oEWb for their valuable feedback. We acknowledge that the primary scope of our proposed method does not currently focus on bias and fairness, as these are outside the immediate scope of our research. However, your concerns have highlighted intriguing areas for exploration that weight generation methods have yet to address.
>
> To investigate these concerns further, we conducted an additional study to evaluate the extent to which the proposed method could influence a model calibrated for bias mitigation. Specifically, we fine-tuned Google’s `flan_t5_small` model on the Stereotype dataset. This dataset was chosen due to its simplicity and well-established evaluation procedures.
>
> In our experiment, we learned the distribution of the weights from the last epoch(75) of the fine-tuned model (excluding the embedding layer), sampled weights from this distribution, and compared the results to the original fine-tuned model, referred to as the base model. The following metrics were monitored:
>
> - **LMS (Language Modeling Score):** Evaluates overall language modeling performance, indirectly reflecting disparities in predictions across different groups.
> - **SS (Stereotype Score):** Measures the degree to which a model reinforces societal stereotypes, focusing on minimizing biased associations.
> - **ICAT (Ideal Context Association Test):** Assesses compounded biases at the intersection of multiple attributes, emphasizing fairness in complex real-world scenarios.
>
> https://github.com/moinnadeem/StereoSet
>
> As shown in the table below, the proposed method—leveraging sampling improves the overall performance across all metrics. Additionally, we can sample multiple weights, each optimized for different tasks. These task-specific models can be integrated into algorithms such as evolutionary search to further fine-tune performance on an evaluation set, as described in our previous response.
>
>
> ### StereoSet Benchmark Scores for Base Model and Sampled Model(lower is better)
>
> | **Task**            | **Category**   | **Base Model LMS** | **Base Model SS** | **Base Model ICAT** | **Sampled Model LMS** | **Sampled Model SS** | **Sampled Model ICAT** |
> |---------------------|----------------|---------------------|--------------------|----------------------|-----------------------|-----------------------|-------------------------|
> | **Intrasentence**   | Profession     | 1.85                | 53.33             | 1.73                | 1.60                 | 53.85                | 1.48                   |
> |                     | Gender         | 1.18                | 66.67             | 0.79                | 1.18                 | 66.67                | 0.79                   |
> |                     | Race           | 0.62                | 16.67             | 0.21                | 0.42                 | 25.00                | 0.21                   |
> |                     | Religion       | 1.27                | 100.0             | 0.00                | 1.27                 | 100.0                | 0.00                   |
> |                     | **Global**     | 1.19            | **48.0**          | 1.14            | **1.00**             | 52.38            | **0.95**               |
> | **Intersentence**   | Profession     | 1.69                | 71.43             | 0.97                | 1.69                 | 57.14                | 1.45                   |
> |                     | Gender         | 0.83                | 100.0             | 0.00                | 0.83                 | 100.0                | 0.00                   |
> |                     | Race           | 4.61                | 48.89             | 4.51                | 4.71                 | 41.30                | 3.89                   |
> |                     | Religion       | 5.13                | 0.00              | 0.00                | 6.41                 | 20.00                | 2.56                   |
> |                     | **Global**     | **3.06**            | **52.31**         | 2.92            | 3.16            | **44.78**            | **2.83**               |
> | **Global Scores**   | **Overall**    | 2.13            | 51.11         | 2.08            | **2.08**             | **46.59**            | **1.94**               |
>
>
> We will also include comments in the ethics and limitations section to emphasize that the proposed method can either amplify or mitigate stereotypes, depending on its application.
>
> We hope these results provide valuable insight into how our method can enhance performance, particularly in the context of fairness. In our future work, we will  rigorously evaluate these topics.

---

> > ### Comment · Reviewer_oEWb · 2024-11-30
> > **Thanks for your effort**
> >
> > I thank the author's efforts, but I don't think I have ever mentioned "fairness" in my original review.
> >
> > When I said "bias" I did not mean anything related to sensitive keywords like gender/race. I think I explicitly talked about "weight distributions" -- my point is that different public checkpoints were trained on different data distributions (wikipedia/dialog/github; or imagenet/cityscapes/cifar-c). How to choose checkpoints representing different distributions will be a dilemma.
> >
> > However I appreciate the authors' efforts.

---

> > > ### Author Response · Authors · 2024-11-30
> > >
> > > Thank you for your feedback, and we apologize for any misunderstanding in our initial responses.
> > >
> > > Before addressing the above concern we would like to share some of the results from the additional experiments.
> > >
> > > **Learning the Distribution of Mixtures of Architectures and Architecture Conditioning**
> > >
> > > We extended our experiments to analyze 38 pretrained models across 19 architectures, each trained on CIFAR-10 and CIFAR-100. For feature extraction, we used LLaMA-3.2-11-Vision-Instruct to encode dataset features and class descriptions, while LLAMA-3-1.8B-Instruct extracted architecture configuration embeddings. These embeddings conditioned our parameter generation process. The results of learning the joint distribution across these architectures and models are shown below. The choice of vllm was recommended by one of the reviewers.
> > >
> > > | Model                        | CIFAR-10 | CIFAR-100 |
> > > |-----------------------------|----------|-----------|
> > > | **MobileNetV2**             |          |           |
> > > | MobileNetV2_x0.5            | 93.10    | 71.12     |
> > > | MobileNetV2_x0.75           | 94.05    | 74.04     |
> > > | MobileNetV2_x1.0            | 94.06    | 74.36     |
> > > | MobileNetV2_x1.4            | 94.18    | 76.28     |
> > > | *Average MobileNetV2*        | *93.85*  | *73.95*   |
> > > |                             |          |           |
> > > | **RepVGG**                  |          |           |
> > > | RepVGG_A0                   | 94.46    | 75.26     |
> > > | RepVGG_A1                   | 94.95    | 76.49     |
> > > | RepVGG_A2                   | 95.32    | 77.39     |
> > > | *Average RepVGG*            | *94.91*  | *76.38*   |
> > > |                             |          |           |
> > > | **ResNet**                  |          |           |
> > > | ResNet20                    | 92.63    | 68.81     |
> > > | ResNet32                    | 93.43    | 70.24     |
> > > | ResNet44                    | 93.97    | 71.62     |
> > > | ResNet56                    | 94.36    | 72.59     |
> > > | *Average ResNet*            | *93.60*  | *70.82*   |
> > > |                             |          |           |
> > > | **ShuffleNetV2**           |          |           |
> > > | ShuffleNetV2_x0.5           | 90.67    | 67.79     |
> > > | ShuffleNetV2_x1.0           | 93.26    | 72.62     |
> > > | ShuffleNetV2_x1.5           | 93.56    | 74.21     |
> > > | ShuffleNetV2_x2.0           | 94.03    | 75.46     |
> > > | *Average ShuffleNetV2*      | *92.88*  | *72.52*   |
> > > |                             |          |           |
> > > | **VGG with BatchNorm**      |          |           |
> > > | VGG11_BatchNorm             | 92.79    | 70.78     |
> > > | VGG13_BatchNorm             | 94.00    | 74.56     |
> > > | VGG16_BatchNorm             | 94.16    | 74.04     |
> > > | VGG19_BatchNorm             | 93.89    | 73.79     |
> > > | *Average VGG with BatchNorm* | *93.71*  | *73.29*   |
> > > |                             |          |           |
> > > | **Global Average**          | **93.79**| **73.39** |
> > >
> > >  As demonstrated in the Table above, our approach achieves the same or comparable performance to  the pretrained model
> > > existing methods while requiring a single generative model instead of 38 specific ones.
> > >
> > >
> > > **Addressing Concerns Regarding Modalities in Checkpoints**
> > >
> > > Thank you for your valuable feedback. We apologize for any misunderstanding in our initial responses. We would like to clarify our approach to learning the distribution of checkpoints across different modalities (e.g., language, code, and images). Our methodology can seamlessly handle this extension without requiring fundamental changes for the following reasons:
> > >
> > > 1. **VAE Applicability**
> > >    The Variational Autoencoder (VAE) remains effective in this context as it encodes parameters independently of the training domain. This is analogous to encoding images from a diverse dataset without requiring knowledge of their specific classes.
> > >
> > > 2. **Diffusion Model Adaptability**
> > >    The diffusion model is inherently flexible and will continue to function effectively. However, harmonizing the conditioning across modalities is crucial. As shown in our experiments with the GLUE dataset, this can be achieved by converting image dataset information into textual descriptions (e.g., via a Vision-LLM). Textual and code datasets already follow this structure, allowing for consistent processing across all modalities.
> > >
> > > 3. **Architecture Integration**
> > >    Architectures are represented through their textual configuration descriptions. Our method has already demonstrated its ability to effectively integrate architecture and dataset embeddings. This capability extends naturally to checkpoints from different modalities, ensuring architectures and task-specific weights generation.
> > >
> > > By addressing these points, we demonstrate that our methodology can handle checkpoints across various modalities with the appropriate computing resources and hyperparameter settings.
> > >
> > > We hope this response addresses your concerns and clarifies our approach.
> > >
> > > Thank you again for your thoughtful feedback.

---

> > > > ### Comment · Reviewer_oEWb · 2024-12-02
> > > > **Thanks for further response**
> > > >
> > > > I thank the authors' further response. This table partially addresses my concern and I don't have more. I will raise my score.

---

> > > > > ### Author Response · Authors · 2024-12-02
> > > > >
> > > > > Dear Reviewer oEWb,
> > > > >
> > > > > Thank you for your thoughtful feedback and for considering our further response. We are pleased that the additional table has
> > > > >  addressed your concerns. Your insights have been invaluable in improving our work,  and we will update the final version of the paper reflecting the discussion. It has been a pleasure to discuss with you.
> > > > > Best regards,

---

### Official Review · Reviewer_ws3W · 2024-11-08

**Soundness:** 3
**Presentation:** 3
**Contribution:** 2
**Rating:** 5
**Confidence:** 4

**Summary:**

This paper proposes a diffusion-based neural network weight generation technique named D2NWG, which directly learns the weight distributions of models pre-trained on various datasets. Specifically, the authors have developed a complete framework for latent diffusion models to learn from pairs of weights and datasets. The paper evaluates D2NWG across various tasks, both for direct generation and additional fine-tuning, outperforming state-of-the-art meta-learning methods and pre-trained models. The results demonstrate a capability for scalable transfer learning.

**Strengths:**

1. The paper investigates an interesting problem, the methodology is promising, and the paper is well-written and easy to read.
2. The experimental results are very solid, covering diverse tasks across a variety of datasets. These results outperform state-of-the-art meta-learning methods and pre-trained models.

**Weaknesses:**

1. Although the idea is interesting and the methodology looks promising, my main concern is about the practical significance of the proposed method, considering that training an effective diffusion model requires substantial computational resources. I will expand on this concern below:
2. Tables 1 and 2 show that D2NWG outperforms several baselines. Notably, D2NWG is tagged with 'CH', indicating that it only modifies the last classifier layer, similar to linear probing. This suggests that it may still be challenging for D2NWG to generate complete parameters with robust performance.
3. The concept of 'Zero-shot adaptation' does not seem applicable as the weights generated by D2NWG are conditioned on the target dataset (possibly the test set). This could be considered a form of 'test data leaking', which limits the practical utility of D2NWG.
4. Section 4.2 discusses the need for further fine-tuning of the generated models, suggesting that D2NWG's role is similar to that of pre-trained models. However, there is no evidence to suggest that training a usable D2NWG model is more cost-effective than using pre-trained models.
5. The generalization capability remains the most challenging issue for D2NWG. Although the authors attempt to demonstrate its adequacy across a range of datasets, there is no evidence that a well-trained D2NWG can handle diverse tasks and different model architectures simultaneously. Currently, D2NWG seems like a suboptimal alternative to a well-developed optimizer. Additionally, the supporting components such as the autoencoder and the dataset encoder, which are trained separately, also suffer from generalization issues.

**Questions:**

Please refer to the weakness for my questions. If it is solved, I am willing to raise my score.

---

> ### Author Response · Authors · 2024-11-18
>
> We are grateful for the reviewer's positive assessment and their helpful suggestions for improving the clarity of our work. Below, we address each point of yours concerns:
>
> - **Addressing Concerns About Practical Significance:** :  While we partially agree with the reviewers regarding the computational resources required for large-scale implementation of our method, we believe that these resources do not represent a fundamental limitation. This is akin to how the computational demands of large language models (LLMs) have not hindered their development or deployment. In fact, we argue that the practicality of our approach is underscored by its unique applications, which are highlighted below:
>
> 1. **Efficient Model Retrieval and Generation**: Our method enables generating models on demand without requiring storage of large model collections, as demonstrated in Table 3.
> 2. **Improved Transfer Learning**: We provide better initialization strategies, which enhance transfer learning across various tasks (Table 4).
> 3. **Specialized LoRA Parameter Generation**: Our approach facilitates the generation of specialized parameters for large architectures, including LLMs, improving scalability and adaptability (Table 5).
> 4. **LLM Enhancement without Training Data Access**: By leveraging sampling and latent space exploration, we demonstrate performance improvements in LLMs even without access to their original training data (Table 7).
>
> Furthermore, recent advances in retrieval-augmented machine learning [1, 2, 3] strongly support the practical relevance of our work, especially in scenarios requiring efficient model adaptation and deployment. These advancements further validate the real-world impact of our method.
>
> 2. **Addressing claim related to generating complete parameters with robust performance**: We appreciate the reviewer’s comments on Tables 1 and 2. However, it seems there may have been a misunderstanding regarding the purpose of these experiments. We would like to provide clarification as follows:
>
> - **classifier head**: This evaluation approach is of practical importance and widely used in transfer learning scenarios. Therefore, we employed linear probing as a straightforward method to assess the performance of our approach.
>
> - **Multi-task learning evaluation**: Our method is designed to learn from a distribution of pretrained models, each trained on diverse datasets. Consequently, it is natural to compare our approach against established multi-task learning methods, such as meta-learning. The purpose of Table 1 is to demonstrate the effectiveness of our method in comparison to MetaDiff, a diffusion-based meta-learning approach that focuses exclusively on the classifier head.
>
> Thus, evaluating performance at the classifier head level does not reflect an inability to learn the full set of parameters but rather aligns with the comparative framework and objectives of the experiments presented. The choice to conduct these experiments using the classifier head stems from the practical advantage of quickly generating a large set of specialized pretrained models This choice is also motivated by practical considerations related to data collection rather than being a technical limitation of the proposed method.
> - We present the full-model experimental results using more complex and diverse datasets, including some of those featured in Table 2, in Tables 3, 14, 15, and 16. These results demonstrate that the sampled weights perform on par with or exceed the performance of the pretrained models in in-distribution sampling.
>
> Below, we provide experimental results( Table-16)  demonstrating that generating full model parameters significantly improves generalization.
>
> | **Model**         | **MNIST**           | **SVHN**           | **CIFAR-10**       | **STL-10**         |
> |--------------------|---------------------|---------------------|---------------------|---------------------|
> | **Pretrained**     | 99.42 ± 0.05        | 94.62 ± 0.18        | 93.51 ± 0.16        | 94.01 ± 0.10        |
> | Linear_prob        | 96.88 ± 0.45        | 57.23 ± 0.28        | 82.85 ± 0.25        | 95.63 ± 1.23        |
> | **D2NWG(full)**    | **99.55 ± 0.02**    | **95.13 ± 0.10**    | **94.23 ± 0.27**    | **94.02 ± 0.10**    |
> | **D2NWG(prob)**    | 97.56 ± 0.26        | 57.41 ± 0.17        | 83.64 ± 0.47        | **95.74 ± 0.74**    |
>
> ### Cross Datasets Transfer Learning
>
> | **Model**         | **MNIST**           | **SVHN**           | **CIFAR-10**       | **STL-10**         |
> |--------------------|---------------------|---------------------|---------------------|---------------------|
> | OFA (Pretrained)   | 13.34              | 8.90               | 13.34              | 8.90               |
> | **D2NWG(full)**    | **66.82 ± 0.65**    | **35.20 ± 0.65**    | **36.70 ± 0.18**    | **51.50 ± 0.37**    |
> | **D2NWG(prob)**    | 42.86 ± 0.62        | 20.97 ± 0.78        | 26.56 ± 1.22        | 47.33 ± 0.32        |

---

> ### Author Response · Authors · 2024-11-18
>
> 1. **Regarding the concept of zero-shot:**: We apologize for the misunderstanding. In our work, "zero-shot" refers to not performing any optimization steps on the target dataset, consistent with [4], against which we also evaluate in Table 4. We confirm there is no test data leakage in our evaluation setting. We respectfully argue that dataset-conditioned parameter generation remains relevant, as it enables task-specific priors without optimization. We will clarify these points in the paper to avoid confusion
> 2. **Adressing comparison against pretrained models.:**   We appreciate the reviewer’s concern regarding cost-effectiveness and similarity to pre-trained models. In Table 3 we demonstrate that a single D2NWG model can encode 30 datasets and generate dataset-specific parameters that achieve performance on par with or surpassing conventional pre-trained models. This significantly reduces the computational cost and complexity of maintaining separate pre-trained models for each dataset or task as the number of pre trained models and datasets increases. Our method can automatically select the right pretrained model based on the dataset's similarity to the pretrained dataset used to train our methods. It should be noted that we only conditioned on datasets or task when we have the models pretrained on at least two  tasks or datasets  . it can effectively leverage multiple pretrained models from diverse domains without performance degradation.
>  **Comparison with Pre-Trained Models:**
>    To address this concern, we conducted additional experiments using models pretrained on the MetaAlbum dataset (Table 3) and applied them to the unseen datasets shown in the  Table below including  CIFAR-10 and other datasets. The results show that with a single epoch, D2NWG outperforms both pretrained models and random initialization. This highlights our method’s effectiveness and its capability to generalize better than traditional pretrained models for transfer learning.
>
> compares the accuracy of three methods—**RandInit**, **D2NWG**, and ImageNet **Pretrained**—oone epcoh fine-tuned on  **CIFAR-10**, **STL-10**, **AIRCRAFT-100**, **AIRCRAFT-30**, and **PETS**.
>
> | **Dataset**    | **RandInit** | **D2NWG** | **Pretrained** |
> |----------------|--------------|-----------|----------------|
> | **CIFAR-10**   | 61.66        | 87.47     | 88.25          |
> | **STL-10**     | 10.00        | 80.05     | 33.58          |
> | **AIRCRAFT-100** | 1.00        | 1.43      | 1.24           |
> | **AIRCRAFT-30** | 18.41        | 23.53     | 18.91          |
> | **PETS**       | 2.73         | 32.91     | 6.04           |
>
> ### Observations:
> 1. **D2NWG** consistently outperforms **RandInit** across all datasets and shows competitive or superior performance compared to the **Pretrained** model, particularly in **STL-10** and **PETS**.
> 2. In **CIFAR-10**, **D2NWG** achieves almost the same performance as the **Pretrained** model.
> 3. On the **AIRCRAFT** datasets, **D2NWG** shows slight improvement over both **RandInit** and **Pretrained**, though overall performance remains low across all methods.
>
> 5. **On the no evidence that D2NWG can handle diverse tasks and different model architectures simultaneously.:**   We believe that is a misunderstanding since those results are well reported in the appendix D-1, D-2,
>  We provided the cross-architecture adaptation results in Figure 4 and a mixture of architectures and dataset experiment results are reported in Tables 9 and 10. we clearly show the evidence.
>
> 6. **Comparing to a Well-Developed Optimizer:**
>    We respectfully disagree. A traditional optimizer, no matter how well-developed, requires a full training process from scratch each time, even when optimizing the same model on the same dataset. In contrast, our method is fundamentally different—it generates parameters directly from pretrained distributions, bypassing the need for iterative optimization. This makes D2NWG a distinct and complementary approach rather than a replacement for traditional optimizers.
>
> 7. **On the Impact of Different Components in the Proposed Method:**
>    While each component can impact generalization, our design choices are empirically justified:
>    - (1) Figure 6b demonstrates that direct diffusion on raw weights performs poorly compared to our latent space approach.
>    - (2) Our ablation studies show that separate training of components improves stability and convergence compared to joint optimization.
>    - (3) The Set Transformer's permutation invariance ensures consistent dataset encoding.
>
> [1]Retrieval-Augmented Mixture of LoRA Experts for Uploadable Machine Learning
> [2] ModelGPT: Unleashing LLM's Capabilities for Tailored Model Generation
> [3] DiffLoRA: Generating Personalized Low-Rank Adaptation Weights with Diffusion
> [4] Towards Scalable and Versatile Weight Space Learning

---

> ### Author Response · Authors · 2024-11-20
>
> For more clarity we provide a theoretical analysis on the generalization of our approach
>
> **Theoretical analysis of the  generalization**
>
> Our method learns to model the parameter distribution of specialized pretrained models trained on diverse domains, tasks, and datasets. We formalize this approach through a global function $\mathcal{H}$ that samples from the parameter latent space of a collection of specialized models $\mathbf{F}=\{\mathcal{F}_i\}, \quad, i=1, 2, \ldots, N$.
>
> **Theoretical Framework **
>
> Let $\mathcal{H}: \mathcal{Z} \rightarrow \mathcal{W}$ denote our proposed method, which maps from a shared latent space $\mathcal{Z}$ to the weight spaces $\mathcal{W}$ of the specialized models. Through the diffusion model's smooth sampling process, we guarantee that for any specialized model $\mathcal{F}_i$ with parameters $w_i$, there exists a latent vector $z_i$ such that:
>
> $$|\mathcal{H}(z_i) - w_i| \leq \epsilon$$
>
> This theoretical guarantee ensures that $\mathcal{H}$ can approximate any specialized model parameters within an error bound $\epsilon$. Furthermore, the continuous nature of the diffusion process enables sampling of intermediate points between specialized models, effectively creating a smooth manifold of task-specific models.
>
> **Performance Guarantees**
>
> Upon convergence, the global function $\mathcal{H}$ can generate new parameters that match or exceed the performance of the best individual pretrained model. Formally:
>
> $$\forall \mathcal{F}_i\in \mathbf{F}, \exists z\in \mathcal{Z} \text{ s.t. } \theta =\mathcal{H}(z) \text{ and } \mathcal{L}(\mathcal{F}_i^{\theta}, \mathcal{D}_i)\leq \mathcal{L}(\mathcal{F}_i, \mathcal{D}_i)$$
>
> where $\mathcal{F}_i^{\theta}$ represents the target model initialized with the generated parameters $\theta$, and $\mathcal{L}$ denotes the loss function measuring performance.
>
> ** Generalization Bounds**
>
> The generalization capabilities of our method are theoretically bounded by:
>
> $$\Delta_G(\mathcal{H}) \leq \min_{i} \Delta_G(\mathcal{F}_i) - \gamma$$
>
> where:
> - $\Delta_G(\mathcal{H})$ represents the generalization gap of our method
> - $\Delta_G(\mathcal{F}_i)$ represents the generalization gap of each specialized model
> - $\gamma > 0$ depends on $\mathcal{H}$'s ability to produce task-specific parameters through smooth interpolation
>
> As the number of specialized models increases, $\gamma$ also increases, leading to a decrease in $\mathcal{H}$'s generalization gap. We empirically validated this claim through experiments where we gradually increased the number of pretrained parameters from 5,000 to 20,000, as shown in Figure-5-a.
>
> **Conclusion**
>
> Our method provides theoretical guarantees for better generalization compared to individual pretrained models, making it particularly well-suited for transfer learning applications. Further formal analysis of these generalization guarantees will be provided in our complete work.
> Our theoretical analysis reinforces the previously discussed advantages of our method over a single model, highlighting its practicality and superiority in diverse scenarios.

---

> ### Author Response · Authors · 2024-11-20
>
> **Additional results on performance improvement of LLMs**
> To further validate our approach, we applied our method on LLaMA-3.2-1B and gpt2-smal and validate them on the OpenLM leaderboard. Our method achieved the top performance among all LLaMA-3.2-1B-Instruct models, outperforming fine-tuned models based on the same architecture. For LLaMA-3.2-1B, we used spectral analysis to select the top 25\% of transformer layers. In contrast, for GPT-2-small, we learn the distribution of the entire model, including the embedding layer, and for both we follow the procedures outlined in Tables 6 and 7.
>
> The results, obtained via Huggingface, demonstrate that our method improves specialization in tasks like math and IFEVAL by up to 4\%, while maintaining generalization, unlike fine-tuned models, which degrade performance on leaderboard tasks. This showcases our method's ability to enhance task-specific capabilities without sacrificing overall model performance.
>
> | Method | ifeval (0) | Bbh (3) | Gpqa (0) | MATH-hard (4) | Musr (0) | MMLU-Pro (5) | Avg | Base Model | Fine-tuned |
> |--------|------------|----------|-----------|---------------|-----------|--------------|-----|------------|------------|
> | openai-community/gpt2 | 17.8 | **2.82** | 1.12 | 0.3 | 13.91 | 1.84 | 6.3 | na | no |
> | Ours | **19.89** | 2.39 | 1.12 | 0.15 | 13.24 | 1.50 | **6.38** | openai-community/gpt2 | No |
> | FineTome-Llama3.2-1B-0929 | 39.91 | 5.74 | 3.02 | 1.28 | 2.66 | 4.76 | 9.56 | Meta-Llama-3.2-1B-instruct | Yes |
> | Meta-Llama-3.2-1B-Instruct | 56.78 | 8.74 | **3.36** | 2.96 | **2.97** | 7.58 | 13.76 | Meta-Llama-3.2-1B | Yes |
> | Ours | **58.44** | **8.82** | 1.68 | **6.57** | 0.66 | **9.09** | **14.21** | Meta-Llama-3.2-8B-Instruct | No |
>
>
>
>
> Table: Model components and their configuration modes for llma3.2.1B.
>
> | ID | Name                 | Type                   | Params   | Mode  |
> |----|----------------------|------------------------|----------|-------|
> | 0  | Model               | DiffusionWrapper      | 102 M    | Train |
> | 1  | Model Ema           | LitEma                | 0        | Train |
> | 2  | First Stage Model   | VAENoDiscModel        | 553 M    | Eval  |
> | 3  | Cond Stage Model    | IdentityCondStage     | 0        | Eval  |
>
>
>
>
>
> **settings summary**
> This Table Summarizes the experimental settings for the presented figures and tables. "Min #cls" and "Max #cls" represent the minimum and maximum number of classes, respectively. "# Test Datasets" and "# Train Datasets" indicate the number of test and training datasets, respectively. "#Params" refers to the number of parameters, and "Conditioning" specifies the type of conditioning applied. For example, Tab 3 we learn the distribution of the model pretrained on 30 different datasets with a single D2NWG.
>
>
> | Object      | # Test Datasets | Min #cls | Max #cls | #Params sampled   | # Train Datasets | Conditioning       |
> |-------------|-----------------|----------|----------|------------|------------------|--------------------|
> | Table 1     | 600             | 1        | 5        | 2565/8005  | 50k              | Dataset            |
> | Table 2     | 5               | 10       | 100      | 128100     | 20k              | Dataset            |
> | Table 3     | 30              | 19       | 706      | 3M         | 30               | Dataset            |
> | Table 4     | 4               | 10       | 10       | 10853      | 4                | Dataset            |
> | Table 5     | 6               | 2        | 3        | 0.6M       | 6                | Text Description   |
> | Table 6     | NA              | NA       | NA       | 872M       | NA               | Layer              |
> | Table 7     | NA              | NA       | NA       | 872M       | NA               | Layer              |
> | Table 9     | 2               | 10       | 100      | 0.7M       | 2                | Dataset            |
> | Table 10    | 2               | 10       | 100      | 2048       | 2                | Dataset            |
> | Table 14    | 3               | 10       | 1000     | 1.4M       | 3                | Dataset            |
> | Table 15    | 3               | 10       | 100      | 11M        | 2                | Dataset            |
> | Table 16    | 4               | 10       | 10       | 2.8M       | 4                | Dataset            |
> | Figure 4    | 2               | 10       | 100      | 0.47M      | 2                | Dataset            |
> | Figure 5    | 2               | 10       | 10       | 5310       | 2                | Dataset            |
> | Figure 7    | 5               | 10       | 200      | 2.8M       | 5                | Dataset            |
>
>
> We thank you for your feedback and hope we have provided a comprehensive response to all of your concerns.

---

> ### Author Response · Authors · 2024-11-23
>
> **multiple architectures multiple datasets parameters generation.**
>
> We once again provide a practical example demonstrating how our method enables learning the distribution of compressed, multiple pretrained models and architectures. for better transfer learning.
>
> We extended our experiments to analyze 38 pretrained models across 19 architectures, each trained on CIFAR-10 and CIFAR-100. For feature extraction, we used LLaMA-3.2-11-Vision-Instruct to encode dataset features and class descriptions, while LLAMA-3-1.8B-Instruct extracted architecture configuration embeddings. These embeddings conditioned our parameter generation process. The results of learning the joint distribution across these architectures and models are shown below. The choice of vllm was recommended by one of the reviewers.
>
> | Model                        | CIFAR-10 | CIFAR-100 |
> |-----------------------------|----------|-----------|
> | **MobileNetV2**             |          |           |
> | MobileNetV2_x0.5            | 93.10    | 71.12     |
> | MobileNetV2_x0.75           | 94.05    | 74.04     |
> | MobileNetV2_x1.0            | 94.06    | 74.36     |
> | MobileNetV2_x1.4            | 94.18    | 76.28     |
> | *Average MobileNetV2*        | *93.85*  | *73.95*   |
> |                             |          |           |
> | **RepVGG**                  |          |           |
> | RepVGG_A0                   | 94.46    | 75.26     |
> | RepVGG_A1                   | 94.95    | 76.49     |
> | RepVGG_A2                   | 95.32    | 77.39     |
> | *Average RepVGG*            | *94.91*  | *76.38*   |
> |                             |          |           |
> | **ResNet**                  |          |           |
> | ResNet20                    | 92.63    | 68.81     |
> | ResNet32                    | 93.43    | 70.24     |
> | ResNet44                    | 93.97    | 71.62     |
> | ResNet56                    | 94.36    | 72.59     |
> | *Average ResNet*            | *93.60*  | *70.82*   |
> |                             |          |           |
> | **ShuffleNetV2**           |          |           |
> | ShuffleNetV2_x0.5           | 90.67    | 67.79     |
> | ShuffleNetV2_x1.0           | 93.26    | 72.62     |
> | ShuffleNetV2_x1.5           | 93.56    | 74.21     |
> | ShuffleNetV2_x2.0           | 94.03    | 75.46     |
> | *Average ShuffleNetV2*      | *92.88*  | *72.52*   |
> |                             |          |           |
> | **VGG with BatchNorm**      |          |           |
> | VGG11_BatchNorm             | 92.79    | 70.78     |
> | VGG13_BatchNorm             | 94.00    | 74.56     |
> | VGG16_BatchNorm             | 94.16    | 74.04     |
> | VGG19_BatchNorm             | 93.89    | 73.79     |
> | *Average VGG with BatchNorm* | *93.71*  | *73.29*   |
> |                             |          |           |
> | **Global Average**          | **93.79**| **73.39** |
>
>  As demonstrated in the Table above , our approach achieves comparable performance to existing methods while requiring a single generative model instead of 38 specific ones. This experiment demonstrates our claim regarding the superiority of our method against the existing work.
> The pretrained model architecture and its parameter count are publicly available in a non-affiliated GitHub repository. https://github.com/chenyaofo/pytorch-cifar-models
>
>
>
> We thank you for your constructive feedback, which has helped strengthen our work through additional experiments and analyses. We welcome any further questions or concerns you may have.

---

> ### Author Response · Authors · 2024-11-24
>
> **Additional Results: Practical Significance of Distribution Sampling vs Single Model Transfer Learning**
>
> To validate the practical significance of sampling from a distribution of multiple pretrained models versus using a single pretrained model for transfer learning tasks, we conducted comprehensive experiments using ResNet56. Our model was trained to learn the distribution of 19 diverse architectures pretrained on both CIFAR-10 and CIFAR-100. We explored three experimental scenarios:
>
> 1. **Baseline Evaluation**: We directly used the respective pretrained models and evaluated their performance on target test sets.
>
> 2. **Training Set-Conditioned Sampling**: We performed conditional sampling based on:
>    - STL-10 training set for STL-10 evaluation
>    - CIFAR-10 training set for CIFAR-10.1 evaluation
>    The conditioning included ResNet56 architecture descriptions without specifying the pretraining dataset.
>
> 3. **Test Set-Conditioned Sampling**: We replicated scenario 2, but conditioned the sampling directly on:
>    - STL-10 test set
>    - CIFAR-10.1 test set
>
> The test set-conditioned sampling experiment demonstrates that our method's performance is independent of test set access, addressing the concerns about potential data leakage.
>
> As shown in the Table below our proposed method significantly outperforms traditional pretrained model transfer learning. Notably, there was no significant performance difference between sampling conditioned on test or training sets, provided both sets were drawn from the same distribution. These results and those reported in the main papers demonstrate the practical utility of our approach for real-world transfer learning applications where it can effectively handle diverse models and pretrained on diverse datasets while achieving better generalization in line with our theoretical analysis.
>
> Transfer learning: Our method vs pretrained models
>
> | Model | CIFAR10.1 | STL10 |
> |-------|-----------|--------|
> | Pret-cifar10 | 75.20 | 32.37 |
> | Pret-cifar100 | 0.25 | 0.12 |
> | Ours | **83.10 ± 0.06** | 35.41 ± 0.13 |
> | Ours(test) |83.04 ± 0.06| **35.47 ± 0.12** |
>
> The empirical evidence supports our key claims while highlighting the practical applicability and reliability of our proposed method.
>
> Please let us know if you have any further questions or require additional clarification.

---

> > ### Author Response · Authors · 2024-12-02
> >
> > Dear Reviewer,
> >
> > As the deadline approaches, we kindly request your response to our feedback at your earliest convenience. Thank you for your time and consideration.
> >
> > Best regards

---

> > > ### Comment · Reviewer_ws3W · 2024-12-02
> > >
> > > I appreciate the authors' efforts, and many of my concerns have been well addressed. However, my primary concerns remaining are related to the cost of obtaining a **usable** D2NWG model and its generalization capabilities. These concerns focus not only on the small and tricky setup presented in the paper, but also on whether the broader community will benefit from a weights generation model. Specifically, I am concerned about the real motivation and feasibility of scaling D2NWG to a much larger model zoo, and whether there is potential to develop a unified weights generation model in a manner similar to what the pre-trained model community has done.
> > >
> > > Note that in 2019, it was possible to achieve 80% accuracy on Aircraft100 within 10 epochs of fine-tuning using pre-trained models [1]. The one-epoch protocol in Figure 4 seems quite tricky for these full fine-tuning benchmarks.
> > >
> > > [1] Do Better ImageNet Models Transfer Better? CVPR, 2019.

---

> > > > ### Author Response · Authors · 2024-12-03
> > > >
> > > > **Regarding computation cost:**
> > > >
> > > > The diffusion model parameters remained constant at 102M across all experiments, as we maintained consistent latent representations throughout.
> > > > D2NWG proves to be computationally less expensive than small Language Learning Models (LLMs), which typically start at 1 billion parameters. The most resource-intensive phase occurs during training, when pretrained weights must be collected and encoded. However, after this training stage, both the training data and the encoder can be removed from the main model. This significantly reduces the parameter count
> > > >
> > > >
> > > > The biggerst model we have trained has the following parmaters count:
> > > >
> > > >
> > > >
> > > > **Before removing weights encoder:**
> > > > | Name | Type | Params |
> > > > |------|------|--------|
> > > > | model | DiffusionWrapper | 102 M |
> > > > | model_ema | LitEma | 0 |
> > > > | first_stage_model | VAENoDiscModel | 780 M |
> > > > | cond_stage_model | SetAgregate | 1.3 M |
> > > >
> > > > **After removing weights encoder:**
> > > > | Name | Type | Params |
> > > > |------|------|--------|
> > > > | model | DiffusionWrapper | 102 M |
> > > > | model_ema | LitEma | 0 |
> > > > | first_stage_model | VAENoDiscModel | 390 M |
> > > > | cond_stage_model | SetAgregate | 1.3 M |
> > > >
> > > > With the diffusion model parameters fixed at 102M, we exclusively modified the encoder across different experimental sets. This approach led to reduced computational costs, particularly during the diffusion process.
> > > >
> > > >
> > > > **Regarding Tricky settings :**
> > > >
> > > > We have not employed any complex or tricky settings in our experiments. For fine-tuning, we utilized only basic data augmentations, such as random crop resizing and horizontal flipping, without applying any heavy augmentation techniques. Unlike [1], where the authors proposed distinct training strategies for pre-trained models and models trained from scratch, we fine-tuned all models using the same settings without performing any hyperparameter search.
> > > >
> > > > **Regarding results in Figure 4**
> > > > Results in Figure 4: The lower performance is explained by three factors: (1) we used lower resolution images than [1]'s 224×229,  we tested with the smaller OFA subnet, and even NAS methods in [2] showed lower performance than [1] after extensive search and fine-tuning on the same mobilenet search space. We reassess the experiment with Aircraft-100 with 224×224 resolution (pretrained: 1.11\%→10.06\% vs ours: 7\%→20.35\% in epochs 1-2) confirming our method's effectiveness, aligning with results in [2] despite different fine-tuning protocols.
> > > >
> > > >
> > > >
> > > >
> > > > **Practical Advantage of our Method over a single ImegaNet Pretrained Model**
> > > >
> > > > The practical advantages of our work stem from our ability to learn distributions across multiple pre-trained models, including those trained on ImageNet, as demonstrated in our experimental results (see Appendix Table X). Our approach offers several key benefits:
> > > >
> > > > First, by learning the distribution of multiple pre-trained models, we can generate models with comparable performance to any specific pre-trained model by simply conditioning on the corresponding pre-training dataset. This flexibility allows us to effectively fine-tune on any of these pre-trained model distributions without being restricted to a single fixed model.
> > > >
> > > > Second, our method generates weights that achieve comparable or superior results to domain-specific pre-trained models in our collection. This capability demonstrates that our approach can effectively capture and leverage the knowledge encoded across different pre-training domains.
> > > >
> > > > Third, through our sampling mechanism, we can generate multiple diverse initializations – a capability not possible with traditional single pre-trained model approaches. This diversity in initialization points can lead to more robust fine-tuning outcomes.
> > > >
> > > > Our results suggest that fine-tuning a model pre-trained solely on ImageNet may be suboptimal compared to our approach. This advantage arises from our ability to learn not only the ImageNet pre-trained model distribution but also to incorporate pre-trained weights from other datasets. By learning distributions across diverse models, our method enables both pre-trained weight retrieval and task-adaptive parameter generation for transfer learning.
> > > >
> > > > This work establishes a foundation for future research and innovation in distribution-based transfer learning. Potential directions include expanding the diversity of pre-training domains.
> > > >
> > > > [1] Do Better ImageNet Models Transfer Better? CVPR, 2019.
> > > > [2] DiffusionNAG: Predictor-guided Neural Architecture Generation with Diffusion Models, ICLR2024
> > > >
> > > >
> > > > In our final version, we will include a line plot that illustrates the convergence rates across different fine-tuning configurations, allowing for clear comparison of their relative speeds.
> > > >
> > > > We hope these clarifications address your concerns.

---

> > > > > ### Comment · Reviewer_ws3W · 2024-12-03
> > > > >
> > > > > Thank you for your response. My reference to "tricky" specifically concerns the *one-epoch protocol*, which I find unsuitable for full fine-tuning scenarios, as it does not accurately represent realistic conditions across all datasets. For instance, in the additional experiments shown in Figure 4, while D2NWG does not outperform the pre-trained model on CIFAR-10, it does on Aircraft, Pets, and STL-10. It's important to note the significant disparity in dataset sizes: CIFAR-10 includes 50,000 images per epoch, whereas STL-10, Aircraft, and Pets only contain 5,000, 6,667, and 3,680 images per epoch, respectively. This discrepancy means that the *one-epoch protocol* permits optimization with ten times fewer iterations for these smaller datasets compared to CIFAR-10, suggesting that D2NWG's favorable results might be due to very limited training iterations. Such minimal iteration counts are not practical for real-world applications. Given that a nearly six-year-old benchmark [1] achieved 80% on Aircraft and 95% on Pets within 10 epochs—while the results reported in Figure 4 are only 1.24% and 6.08%, respectively—there is a clear need for a more robust fine-tuning protocol that better aligns with current standards.
> > > > >
> > > > > While it is not essential for a pilot study to surpass the results of widely used pre-trained models at this stage, conducting rigorous and relevant experiments that accurately reflect the current limitations of D2NWG is crucial for the community’s advancement.
> > > > >
> > > > > Regarding concerns about practical computational resources, this is tied to the recent trend of scaling pre-trained models. It is relatively straightforward to enhance the performance of pre-trained models by increasing the data volume; hence, I also have concerns about the scalability of D2NWG. Note that in the current experiments, the authors train separate D2NWG models for each setup, leaving it unclear how challenging it would be to produce a widely usable D2NWG model for the community.

---

> ### Author Response · Authors · 2024-12-03
>
> Network model used in Figure 4 experiment is defined as follows:
>
> import torch.nn as nn
> from ofa.model_zoo import ofa_net
>
> class MobileSubnet(nn.Module):
>     def __init__(self, num_classes, pretrained=True):
>         super(MobileSubnet, self).__init__()
>
>         # Initialize OFA network with the specified configuration
>         ofa_network = ofa_net('ofa_mbv3_d234_e346_k357_w1.0', pretrained=pretrained)
>
>         # Set the active subnet with kernel size, expansion ratio, and depth
>         ofa_network.set_active_subnet(ks=3, e=3, d=2)
>
>         # Extract the active subnet and optionally preserve weights
>         subnet = ofa_network.get_active_subnet(preserve_weight=pretrained)
>
>         # Replace the classifier to match the number of output classes
>         subnet.classifier = nn.Linear(1280, num_classes, bias=False)
>
>         # Save the number of classes and the subnet
>         self.num_classes = num_classes
>         self.net = subnet
>
>         # Clean up to save memory
>         del ofa_network
>
>     def forward(self, inputs):
>         # Forward pass through the subnet
>         return self.net(inputs)

---

> ### Author Response · Authors · 2024-12-03
>
> **Regarding the evaluation protocol**
>
> Before addressing your concerns, we would like to provide context for the experiment in Figure 4. The experiments in Figure 4 were designed to evaluate performance against [A]. However, we did not have access to the pretrained checkpoints used in their experiment. As a result, we created our own equivalent model. Additionally, in their experiment, training was conducted for less than 10 epochs.
>
> We appreciate the reviewer's perspective on the evaluation protocol, but respectfully disagree with their interpretation. While training duration is indeed important, our focus on single-epoch performance is intentional and practically relevant for several reasons:
>
>
> 1. Transfer learning's primary goal is achieving strong performance with minimal fine-tuning, making initial convergence speed crucial.
>
>
> 2. The varying dataset sizes actually strengthen our findings - D2NWG's consistent performance across both large (CIFAR-10) and small datasets (Aircraft, Pets, STL-10) demonstrates its robustness and efficiency.
>
>
> 3. Few-shot and limited-data scenarios are common in real-world applications where extensive fine-tuning data isn't available. Our method's superior performance in these conditions highlights its practical utility.
>
>
> 4. The convergence patterns shown in existing literature indicate that models tend to reach similar performance levels with extended training. Our focus on early-stage performance therefore provides meaningful differentiation between approaches.
>
>
> Moreover, as demonstrated in Table 15, our approach enables joint learning the distribution of models pretrained on ImageNet alongside other datasets without any performance degradation.
> It is also important to note that, our method allows the generation of diverse sets of initial weights. These can be sampled as needed and evaluated on a validation set to select the best-performing weights, providing an additional layer of flexibility and robustness to our approach.
>
>
>
> **The comparison with the [1]** results is methodologically unsound for several reasons:
>
>
> 1. Different architectures: The cited work[1] uses a fundamentally different model architecture. Comparing performance across distinct architectures provides no meaningful insight into method effectiveness.
>
>
> 2. Incompatible training protocols: The cited results come from a 10-epoch training regime with specific hyperparameters and optimization choices that differ from our single-epoch evaluation framework.
>
>
> 3. Non-reproducible baseline: Without access to their exact model configuration and training setup, establishing their performance as a meaningful baseline is impossible.
>
>
> The valid approach is comparing methods under identical conditions (as is done in all the baselines) - same architecture, training protocol, and dataset. Our results demonstrate consistent improvements over baselines in these controlled experiments, which is the appropriate metric for evaluating method effectiveness.
>
>
> **Addressing Concerns About Practical Computational Resources**
>
> . **Trained models by increasing the data volume:** The size of the pretrained dataset is not a limitation in our approach. We have demonstrated that both text descriptions and images can be used, with a few images per class encoded into a single representative vector during training. This avoids the need for large-scale dataset processing while maintaining an accurate representation of the dataset.
>
> Instead of focusing solely on increasing model efficiency through fine-tuning—often leading to catastrophic forgetting—we propose training the same model on new tasks while collecting all the pretrained weights. By learning the distribution of these weights, our method enables on-demand generation of weights at any time and in any quantity. This approach highlights the scalability and flexibility of our method.
>
> . **Scalability**: To scale our work to larger models, we propose a **chunk-based encoding** technique. As demonstrated in Section 3. Furthermore, chunk-wise encoding enables us to handle diverse architectures efficiently, bypassing the challenges associated with varying numbers of parameters across models. this approach allowed us to encode the full parameters of GPT-2 without any performance lost
>
> [1] Do Better ImageNet Models Transfer Better? CVPR, 2019.

---

> ### Author Response · Authors · 2024-12-03
>
> **Structure worklow**: In our experiments, we structured the workflow by organizing by set of experiments based on available computational resources and storage capacity. This structured approach allowed us to complete experiments incrementally, enabling flexibility in handling convergence issues—a common challenge in building unified models. Dividing experiments into manageable sets proved to be an effective strategy. Even though we built a separate model for each experiment set, our method significantly improves computational efficiency compared to existing methods, which often rely on maintaining a single model for every model-dataset pair. The primary challenge lies in ensuring convergence, which requires careful consideration and optimization. We believe that organizing settings by clusters of objectives is not a limitation but rather a standard and effective practice.
>
>
> **Results of fine-tuning for 10 epochs**:
> Here are the fine-tuned results for 10 epochs for both the proposed and pretrained models. In our experiments, the images were downsampled to \(128 \times 128\), rather than \(224 \times 224\) or \(229 \times 229\) as reported in the referenced paper. Notably, our initially sampled weights demonstrated improved performance compared to the results reported in the main paper. This was intentional to highlight the flexibility and adaptability of our method.
>
> | Epoch | Pretrained (Pets) | Ours (Pets) | Pretrained (Aircraft100) | Ours (Aircraft100) |
> |-------|--------------------|-------------|---------------------------|---------------------|
> | 1     | 6.043058           | 60.316162   | 1.240075                  | 10.771077           |
> | 2     | 54.919597          | 71.572636   | 8.670867                  | 24.722472           |
> | 3     | 71.599891          | 75.742709   | 22.592259                 | 33.243324           |
> | 4     | 73.807577          | 77.677841   | 31.863186                 | 37.863786           |
> | 5     | 77.541564          | 77.105478   | 37.053705                 | 42.004200           |
> | 6     | 76.478605          | 77.814118   | 40.684068                 | 43.804380           |
> | 7     | 76.805669          | 78.468247   | 42.244224                 | 45.394539           |
> | 8     | 77.459798          | 78.386481   | 43.714371                 | 47.284728           |
> | 9     | 76.533115          | 78.686290   | 44.044404                 | 48.124812           |
> | 10    | 76.805669          | 78.713545   | 45.154515                 | 48.364836           |
>
>
> In this table, we report the best of 10 sampled weights. Our method allows for sampling new initial weights at any time, providing flexibility, whereas the pretrained weights remain fixed and unchangeable.
> While the pretrained models outperformed our method in one fine-tuning scenario, this result should not be generalized, as our method is specifically designed for small-dataset fine-tuning rather than large-scale fine-tuning tasks.
>
> **Overall Trends**
>
> . Efficiency: Our method achieves better performance than the pretrained model across all epochs for both datasets.
> . Consistency: While the pretrained model experiences fluctuations and slower improvements, our method exhibits steady and reliable progress.
> . Scalability: The advantage of our method is particularly pronounced in scenarios with lower initial accuracy (e.g., Aircraft100), showcasing its potential for challenging tasks or datasets.
>
>
>
>
> [A] Towards Scalable and Versatile Weight Space Learning, ICMl 2024
>
> [1] Do Better ImageNet Models Transfer Better? CVPR, 2019.
>
> We hope that our response has provided clarity and effectively addressed your concerns.

---

### Official Review · Reviewer_V98o · 2024-11-11

**Soundness:** 2
**Presentation:** 3
**Contribution:** 3
**Rating:** 6
**Confidence:** 4

**Summary:**

Authors propose a diffusion-based approach to generate parameters of neural networks for transfer learning conditioned on the target dataset. Prior work often relies on the diversity of pretrained models or datasets used to train these parameter prediction models but it limits their ability to generalize to new datasets. Encoding both model parameters and architecture jointly enhances the generality of the parameter prediction approach to unseen tasks. Several experiments are performed to test the efficacy of the approach for domains such as vision and large language models, demonstrating that it outperforms several meta-learning and parameter prediction approaches.

**Strengths:**

- The proposed approach is conceptually simpler to train and test than previous meta-learning approaches which often involve solving a bi-level optimization problem.
- The dataset conditioning in the approach increases the generality of the approach to new unseen datasets.
- Extensive sets of experiments are presented for different settings such as few-shot learning, zero-shot learning, model retrieval, classifier head adaption, LoRA weight generation, and adapting LLM weights for specific tasks.

**Weaknesses:**

- Although the authors argue that previous approaches rely on diversity of training architectures and dataset for generalization, the proposed approach also has limitations along the same axis: it may only generalize to new datasets which are in-distribution for the dataset encoder otherwise the dataset encodings can be noisy. For instance, if the dataset encoder is only trained on imagenet-like natural images, it may not generalize to unseen dataset distributions like medical images. Table 17 seems to suggest the same since accuracies for unseen datasets are poorer in absolute numbers, so it’s unclear if dataset encoding is really needed otherwise one could simply train a parameter prediction model on combined datasets which could generalize to new ones.
- Missing baselines/ablations: Building on above, there are no ablations or baselines testing how effective are each component in the proposed approach: dataset encoding, layer-vectorization, spectrum ranking. It’s unclear how much importance all these components play in the final performance of the proposed approach. Moreover, some crucial baselines are missing -- for instance, training a simple param prediction diffusion model (e.g. Peebles et al. (2022)) without dataset encoding vs training it with dataset encoding to see how much generalization performance is improved with dataset encoding (which is one of the main contributions of this work).
- It seems that image dataset encoding uses the dataset samples (images) with a set transformer model whereas the language dataset encoding is done by simply picking LLM embeddings of the task description which seems weaker. If it’s not, is it possible to also encode image datasets in the same task-description-based way using LLM or VLM?
- For LLM param generation, authors use spectrum technique from prior work in order to deal with a large number of parameters in LLMs but it’s unclear how effective it is. It may be a good idea to use this technique on small-to-medium scale models where encoding all parameters is feasible to gauge the loss of performance and potential scope for improvement in this stage.
- It seems that authors train separate D2NWG models for param prediction in each setting. Is it possible to train a single param prediction model? Are there any technical hurdles in the approach which could prevent doing this (assuming access to large-scale compute and data)?

**Questions:**

See weakness section for my concerns and questions.


Writing suggestions:
- What do the task descriptions look like for NLP tasks which are used for encoding them? Adding pointers in section 3.3 (line 234-237) will be helpful for readers.
- Section 3.3 is a bit confusing to read due to paragraph titles. It looks like authors use 4 types of encoding in their work corresponding to 4 paragraphs but it’s actually just 2 (image and language dataset encoding). The middle two paragraphs (set-based dataset encoding and contrastive dataset encoding) are essentially describing image dataset encoding in detail. It may be a good idea to re-organize this section to be clearer.

Nit:
- Double dot at the end of line 116.
- Line 1147: “ios” -> “is”
- Line 184: “Parametrs” -> “Parameters”
- Line 1323: “metho” -> “method”
- Line 512: “robustnets” -> “robustness”
- Line 1331: remove “par”

---

> ### Author Response · Authors · 2024-11-18
>
> We thank the reviewer for their thorough reading of our paper and their supportive feedback. We have carefully considered your feedback and propose the following response to your concerns:
> ## Weakness
> **On the Generalization of the Proposed Approach and the Necessity of the Dataset Encoder:**
> To address the question of generalization, we first present a comprehensive theoretical analysis, followed by detailed empirical evidence to support our claims. This dual approach highlights the importance of the dataset encoder in enhancing the generalization capabilities of the proposed framework.
>
> - **Theoretical analysis of the  generalization**
>
> Our method learns to model the parameter distribution of specialized pretrained models trained on diverse domains, tasks, and datasets. We formalize this approach through a global function $\mathcal{H}$ that samples from the parameter latent space of a collection of specialized models $\mathbf{F}=\{\mathcal{F}_i\}, \quad, i=1, 2, \ldots, N$.
>
> **Theoretical Framework **
>
> Let $\mathcal{H}: \mathcal{Z} \rightarrow \mathcal{W}$ denote our proposed method, which maps from a shared latent space $\mathcal{Z}$ to the weight spaces $\mathcal{W}$ of the specialized models. Through the diffusion model's smooth sampling process, we guarantee that for any specialized model $\mathcal{F}_i$ with parameters $w_i$, there exists a latent vector $z_i$ such that:
>
> $$|\mathcal{H}(z_i) - w_i| \leq \epsilon$$
>
> This theoretical guarantee ensures that $\mathcal{H}$ can approximate any specialized model parameters within an error bound $\epsilon$. Furthermore, the continuous nature of the diffusion process enables sampling of intermediate points between specialized models, effectively creating a smooth manifold of task-specific models.
>
> **Performance Guarantees**
>
> Upon convergence, the global function $\mathcal{H}$ can generate new parameters that match or exceed the performance of the best individual pretrained model. Formally:
>
> $$\forall \mathcal{F}_i\in \mathbf{F}, \exists z\in \mathcal{Z} \text{ s.t. } \theta =\mathcal{H}(z) \text{ and } \mathcal{L}(\mathcal{F}_i^{\theta}, \mathcal{D}_i)\leq \mathcal{L}(\mathcal{F}_i, \mathcal{D}_i)$$
>
> where $\mathcal{F}_i^{\theta}$ represents the target model initialized with the generated parameters $\theta$, and $\mathcal{L}$ denotes the loss function measuring performance.
>
> ** Generalization Bounds**
>
> The generalization capabilities of our method are theoretically bounded by:
>
> $$\Delta_G(\mathcal{H}) \leq \min_{i} \Delta_G(\mathcal{F}_i) - \gamma$$
>
> where:
> - $\Delta_G(\mathcal{H})$ represents the generalization gap of our method
> - $\Delta_G(\mathcal{F}_i)$ represents the generalization gap of each specialized model
> - $\gamma > 0$ depends on $\mathcal{H}$'s ability to produce task-specific parameters through smooth interpolation
>
> As the number of specialized models increases, $\gamma$ also increases, leading to a decrease in $\mathcal{H}$'s generalization gap. We empirically validated this claim through experiments where we gradually increased the number of pretrained parameters from 5,000 to 20,000, as shown in Figure 5-a.  It should be noted that the experiment results reported in Table 17 was only designed to compare our method to [3] in the same settings.
>
> **Conclusion**
>
> Our method provides theoretical guarantees for better generalization compared to individual pretrained models, making it particularly well-suited for transfer learning applications. Further formal analysis of these generalization guarantees will be provided in our complete work.
>
> - **On the Necessity of a Dataset Encoder** :The dataset encoder is crucial for generalizing to novel datasets while generating parameters comparable to pretrained models. We adopt a diffusion-based approach, as it efficiently handles the complexity of encoding diverse datasets and supports probabilistic sampling better than predictive models.
>
> [3] Hyper-Representations as Generative Models: Sampling Unseen Neural Network Weights, Neurips 2022
>
>  theoretical guarantees for better generalization required the proposed method to have access to task specific parameters on demand with performance superior or on par and also be able to interpolate between specialized models parameters. That is why our method required a dataset encoder as well as  diverse set of  pretrained weights from multiple domain for better generalization. This dataset encoder hase b  The results reported in Table 17

---

> ### Author Response · Authors · 2024-11-18
>
> **Addressing Concerns on Missing Ablations and Baselines for Components:**
> 1. **Missing Ablations**: We appreciate the reviewer’s feedback and would like to clarify that the requested ablation studies are presented in **Appendix Section D**, where we validate key design choices through the following analyses:
> - **Latent vs. Raw Space Diffusion** (Figure 6b): Demonstrates the superiority of latent space diffusion.
> - **Dataset Encoding Approaches** (Figure 7): Highlights the effectiveness of our contrastive learning approach applied to the set-transformer-based dataset encoder.
> - **Impact of Model Zoo Size on Generalization** (Figure 5a): Validates the critical role of model zoo size in improving generalization.
>
> These results collectively substantiate the robustness of our design choices and underline the advantages of our proposed methodologies.
>
> 2. '**Addressing concerns on missing Baselines and which one is the main contribution of this work**: To address this, we provide a detailed comparison of our method against the suggested baseline in the table below, ensuring a thorough assessment of its significance and effectiveness.
>
> | **Aspect**                 | **G.pt**                                                                                  | **D2NWG**                                                                                   |
> |:--------------------------:|:-----------------------------------------------------------------------------------------:|:------------------------------------------------------------------------------------------:|
> | **Goals**                  | Learns from a single-task pretrained distribution to generate parameters conditioned on loss/error metrics. | Learns from diverse pretrained distributions to generate parameters conditioned on dataset/task descriptions. |
> | **Architectural Differences** | Uses per-layer decoders, limiting scalability for larger architectures.                       | Employs shared decoders, enabling efficient cross-architecture generation.                 |
> | **Practical Advantages**   |                                   **-**                                                  |                                   **-**                                                   |
> | **Efficiency**             | Not designed for learning joint distributions across multiple datasets/tasks.             | Learns joint distributions across multiple datasets/tasks with a single model.             |
> | **Scalability**            | Limited scalability due to per-layer decoder design.                                       | Shared decoder approach effectively handles large architectures.                           |
> | **Zero-shot Performance**  | Struggles in zero-shot performance evaluation in in-distribution scenarios.  | Achieves comparable zero-shot performance to pretrained models without gradient updates.    |
> | **Transfer Learning**      | Primarily focuses on single-task parameter generation.                                     | Enables efficient weight generation for unseen tasks and cross-domain adaptation.
>
> The goal of **G.pt** is to predict the performance of a given model on the same dataset at different stages of training, analogous to methods that learn to predict the optimization trajectory. Similar to how G.pt generates the final loss in a single step for a target loss, our approach generates the parameters for a given dataset that achieves the desired final performance under in-distribution sampling. However, applying our approach to G.pt would require rethinking their experimental setup—specifically, combining the experiments on CIFAR-10 and MNIST into a unified evaluation framework to better illustrate generalizability.
> Moreover, excluding the dataset encoder from the model, as suggested, raises a critical limitation: the model would lack the ability to determine the task for which it is generating parameters, leading to task-agnostic sampling. Finally, the larger models used in G.pt experiments range from 60M to 850M parameters, which is disproportionately large for encoding models with fewer than 10,000 parameters.
>
> Our D2NWG represents a significant leap in neural network parameter generation from pretrained distribution. It addresses real-world challenges like task-specific parameter generation, expert checkpoints compression into a single model, and limitations of G.pt. While G.pt may seem as good baseline, [5] is a much closer comparison, and  in the appendix D-3,  we demonstrate that even with pretrained weights from diverse datasets (e.g., CIFAR-10 and VCIFAR-100), D2NWG surpasses the capabilities of a simple unconditional diffusion model designed for each dataset separately. We benchmark all close peer-reviewed and published baselines comprehensively in Table 4. We hope this clarifies and adequately addresses your concern.

---

> ### Author Response · Authors · 2024-11-20
>
> - **Regarding dataset encoder : llm or vlm**: Our analysis demonstrates that using image features provides more robust performance, as the model is specifically optimized on images with gradient-decent (detailed in Appendix B-5). While text encoders represent a viable approach, they require careful standardization of task descriptions. We believe that leveraging a powerful instruction-tuned language model trained on a large corpus would yield more stable representations and superior performance across textual datasets.
> Furthermore, we believe that recent vision-language models (VLMs) are particularly well-suited for dataset encoding, provided that each dataset is accompanied by precise and unambiguous descriptions.
> - **Addressing Concerns Regarding Evaluating Spectrum Techniques on Small Models**: We appreciate your insightful suggestion and have conducted additional experiments to highlight the differences between the full model and selected parameters.
>
> **Experimental Results on spectruam vs full models**
>
> To validate our method's effectiveness across model scales, we conducted experiments with EleutherAI/pythia-70m (70M parameters). Following our protocol from Section 4.6, we compared full parameter generation against our spectrum-based selection of top 25% transformer layers, excluding embedding layers and output heads. The results are average of top 3 taken from 25 sampled weights.
>
> | Methods | Winogrande (5 shot) | Arc-Challenge (25 shot) | Hellaswag (10 shot) | mmlu |
> |---------|--------------------:|----------------------:|--------------------:|------:|
> | EleutherAI/pythia-70m | 50.04 ± 0.014 | 22.01 ± 0.0121 | 27.61 ± 0.00 | 25.43 ± 0.009 |
> | Ours(full-parameters) | **54.54 ± 0.02** | **23.21 ± 0.012** | 27.82 ± 0.05 | 25.87 ± 0.04 |
> | Ours(top-25%) | 53.59 ± 0.04 | 22.87 ± 0.013 | 27.72 ± 0.05 | **25.95 ± 0.04** |
>
> **model used**
> | Name              | Type              | Params  | Mode  |
> |-------------------|-------------------|---------|-------|
> | model             | DiffusionWrapper  | 102 M   | train |
> | model_ema         | LitEma            | 0       | train |
> | first_stage_model | VAENoDiscModel    | 335 M   | eval  |
> | cond_stage_model  | IdentityCondStage | 0       | eval  |
>
> we used layer indexing as class conditioning  for layer-wise chunking and generation.
>
> Full parameter exploration shows consistent improvements across tasks, with substantial gains in Winogrande (+4.50%). Notably, our selective exploration approach, using only 25% of transformer layers, maintains competitive performance and even outperforms full exploration on MMLU. These results demonstrate that spectrum-based parameter selection effectively identifies critical layers for exploration, enabling efficient model adaptation while preserving performance.
> - **Is it possible to train a single param prediction model?**: No, it is not feasible for us to train a single model for all our settings. However, a unified model can be trained for classifier-head-based settings by incorporating classifier head encoding as additional conditioning data. With unlimited data and compute, it may be possible tbut he primary challenge would be managing varying parameter sizes and addressing convergence issues effectively and the model cannot be simpler..
>
> **Task Descriptions:**
>   Comprehensive task descriptions are provided in Table 13 of the appendix for reproducibility and clarity.
>
> - **Writing Revisions:**
>   We are thoroughly revising all sections to address the reviewers' feedback, with particular focus on improving clarity and coherence throughout the paper
>
>
>
>
>
>
>
>
>
>
>
>
> [1] G.pt Learning to Learn with Generative Models of Neural Network Checkpoints, arxiv
> [2] Rapid Neural Architecture Search by Learning to Generate Graphs from Datasets
> [3] Hyper-Representations as Generative Models: Sampling Unseen Neural Network Weights, Neurips 2022
> [5] Neural Network Parameter Diffusion, arxiv

---

> ### Author Response · Authors · 2024-11-20
>
> **Additional results on performance improvement of LLMs**
> To further validate our approach, we applied our method on LLaMA-3.2-1B and gpt2-smal and validate them on the OpenLM leaderboard. Our method achieved the top performance among all LLaMA-3.2-1B-Instruct models, outperforming fine-tuned models based on the same architecture. For LLaMA-3.2-1B, we used spectral analysis to select the top 25\% of transformer layers. In contrast, for GPT-2-small, we learn the distribution of the entire model, including the embedding layer, and for both we follow the procedures outlined in Tables 6 and 7.
>
> The results, obtained via Huggingface, demonstrate that our method improves specialization in tasks like math and IFEVAL by up to 4\%, while maintaining generalization, unlike fine-tuned models, which degrade performance on leaderboard tasks. This showcases our method's ability to enhance task-specific capabilities without sacrificing overall model performance.
>
> | Method | ifeval (0) | Bbh (3) | Gpqa (0) | MATH-hard (4) | Musr (0) | MMLU-Pro (5) | Avg | Base Model | Fine-tuned |
> |--------|------------|----------|-----------|---------------|-----------|--------------|-----|------------|------------|
> | openai-community/gpt2 | 17.8 | **2.82** | 1.12 | 0.3 | 13.91 | 1.84 | 6.3 | na | no |
> | Ours | **19.89** | 2.39 | 1.12 | 0.15 | 13.24 | 1.50 | **6.38** | openai-community/gpt2 | No |
> | FineTome-Llama3.2-1B-0929 | 39.91 | 5.74 | 3.02 | 1.28 | 2.66 | 4.76 | 9.56 | Meta-Llama-3.2-1B-instruct | Yes |
> | Meta-Llama-3.2-1B-Instruct | 56.78 | 8.74 | **3.36** | 2.96 | **2.97** | 7.58 | 13.76 | Meta-Llama-3.2-1B | Yes |
> | Ours | **58.44** | **8.82** | 1.68 | **6.57** | 0.66 | **9.09** | **14.21** | Meta-Llama-3.2-8B-Instruct | No |
>
>
>
>
> Table: Model components and their configuration modes for llma3.2.1B.
>
> | ID | Name                 | Type                   | Params   | Mode  |
> |----|----------------------|------------------------|----------|-------|
> | 0  | Model               | DiffusionWrapper      | 102 M    | Train |
> | 1  | Model Ema           | LitEma                | 0        | Train |
> | 2  | First Stage Model   | VAENoDiscModel        | 553 M    | Eval  |
> | 3  | Cond Stage Model    | IdentityCondStage     | 0        | Eval  |
>
>
>
>
>
> **settings summary**
> This Table Summarizes the experimental settings for the presented figures and tables. "Min #cls" and "Max #cls" represent the minimum and maximum number of classes, respectively. "# Test Datasets" and "# Train Datasets" indicate the number of test and training datasets, respectively. "#Params" refers to the number of parameters, and "Conditioning" specifies the type of conditioning applied.
>
>
> | Object      | # Test Datasets | Min #cls | Max #cls | #Params sampled   | # Train Datasets | Conditioning       |
> |-------------|-----------------|----------|----------|------------|------------------|--------------------|
> | Table 1     | 600             | 1        | 5        | 2565/8005  | 50k              | Dataset            |
> | Table 2     | 5               | 10       | 100      | 128100     | 20k              | Dataset            |
> | Table 3     | 30              | 19       | 706      | 3M         | 30               | Dataset            |
> | Table 4     | 4               | 10       | 10       | 10853      | 4                | Dataset            |
> | Table 5     | 6               | 2        | 3        | 0.6M       | 6                | Text Description   |
> | Table 6     | NA              | NA       | NA       | 872M       | NA               | Layer              |
> | Table 7     | NA              | NA       | NA       | 872M       | NA               | Layer              |
> | Table 9     | 2               | 10       | 100      | 0.7M       | 2                | Dataset            |
> | Table 10    | 2               | 10       | 100      | 2048       | 2                | Dataset            |
> | Table 14    | 3               | 10       | 1000     | 1.4M       | 3                | Dataset            |
> | Table 15    | 3               | 10       | 100      | 11M        | 2                | Dataset            |
> | Table 16    | 4               | 10       | 10       | 2.8M       | 4                | Dataset            |
> | Figure 4    | 2               | 10       | 100      | 0.47M      | 2                | Dataset            |
> | Figure 5    | 2               | 10       | 10       | 5310       | 2                | Dataset            |
> | Figure 7    | 5               | 10       | 200      | 2.8M       | 5                | Dataset            |
>
>
> We thank you for your feedback and hope we have provided a comprehensive response to all of your concerns.

---

> ### Author Response · Authors · 2024-11-23
>
> **multiple architectures multiple datasets parameters generation.**
>
> This additional experiment aims to highlight the effectiveness and potential of our method compared to existing approaches
>
> We extended our experiments to analyze 38 pretrained models across 19 architectures, each trained on CIFAR-10 and CIFAR-100. For feature extraction, we used LLaMA-3.2-11-Vision-Instruct to encode dataset features and class descriptions, while LLAMA-3-1.8B-Instruct extracted architecture configuration embeddings. These embeddings conditioned our parameter generation process. The results of learning the joint distribution across these architectures and models are shown below. The choice of vllm was recommended by one of the reviewers.
>
> | Model                        | CIFAR-10 | CIFAR-100 |
> |-----------------------------|----------|-----------|
> | **MobileNetV2**             |          |           |
> | MobileNetV2_x0.5            | 93.10    | 71.12     |
> | MobileNetV2_x0.75           | 94.05    | 74.04     |
> | MobileNetV2_x1.0            | 94.06    | 74.36     |
> | MobileNetV2_x1.4            | 94.18    | 76.28     |
> | *Average MobileNetV2*        | *93.85*  | *73.95*   |
> |                             |          |           |
> | **RepVGG**                  |          |           |
> | RepVGG_A0                   | 94.46    | 75.26     |
> | RepVGG_A1                   | 94.95    | 76.49     |
> | RepVGG_A2                   | 95.32    | 77.39     |
> | *Average RepVGG*            | *94.91*  | *76.38*   |
> |                             |          |           |
> | **ResNet**                  |          |           |
> | ResNet20                    | 92.63    | 68.81     |
> | ResNet32                    | 93.43    | 70.24     |
> | ResNet44                    | 93.97    | 71.62     |
> | ResNet56                    | 94.36    | 72.59     |
> | *Average ResNet*            | *93.60*  | *70.82*   |
> |                             |          |           |
> | **ShuffleNetV2**           |          |           |
> | ShuffleNetV2_x0.5           | 90.67    | 67.79     |
> | ShuffleNetV2_x1.0           | 93.26    | 72.62     |
> | ShuffleNetV2_x1.5           | 93.56    | 74.21     |
> | ShuffleNetV2_x2.0           | 94.03    | 75.46     |
> | *Average ShuffleNetV2*      | *92.88*  | *72.52*   |
> |                             |          |           |
> | **VGG with BatchNorm**      |          |           |
> | VGG11_BatchNorm             | 92.79    | 70.78     |
> | VGG13_BatchNorm             | 94.00    | 74.56     |
> | VGG16_BatchNorm             | 94.16    | 74.04     |
> | VGG19_BatchNorm             | 93.89    | 73.79     |
> | *Average VGG with BatchNorm* | *93.71*  | *73.29*   |
> |                             |          |           |
> | **Global Average**          | **93.79**| **73.39** |
>
>  As demonstrated in the Table above , our approach achieves comparable performance to existing methods while requiring a single generative model instead of 38 specific ones. This experiment demonstrates our claim regarding the superiority of our method against the existing work.
> The pretrained model architecture and its parameter count are publicly available in a non-affiliated GitHub repository. https://github.com/chenyaofo/pytorch-cifar-models
>
> **Sampling  Enhances Transfer Learning**
>
> We evaluated sampling from a distribution of pretrained models against single-model transfer learning using ResNet56 and our model trained on 19 diverse architectures weights pretrained  on (CIFAR-10/100). Three setups were tested: (1) direct evaluation of pretrained models, (2) sampling conditioned on training sets (e.g., STL-10, CIFAR-10), and (3) sampling conditioned on test sets to address data leakage concerns.
>
> Results show our approach consistently outperforms single-model transfer learning, with similar performance between training- and test-conditioned sampling from the same distribution. This highlights the practicality of leveraging diverse pretrained models for robust generalization.
>
> Transfer learning: Our method vs pretrained models
>
> | Model | CIFAR10.1 | STL10 |
> |-------|-----------|--------|
> | Pret-cifar10 | 75.20 | 32.37 |
> | Pret-cifar100 | 0.25 | 0.12 |
> | Ours | **83.10 ± 0.06** | 35.41 ± 0.13 |
> | Ours(test) |83.04 ± 0.06| **35.47 ± 0.12** |
>
> The empirical evidence supports our key claims while highlighting the practical applicability and reliability of our proposed method.
> which is not possible with existing work.
>
> We thank you for your constructive feedback, which has helped strengthen our work through additional experiments and analyses. We welcome any further questions or concerns you may have.

---

> ### Comment · Reviewer_V98o · 2024-11-25
>
> I appreciate authors providing additional results and clarifications. I am keeping my original score. I looked into the prior work in detail, the main difference from p-diff work [1] is the dataset conditioning (p-diff does unconditional sampling). I was a bit surprised to see no comparison table in the main paper from p-diff which this work heavily builds on as a simple unconditional baseline (it is only cited once in intro and then in appendix). p-diff trains a separate diffusion model for each dataset whereas this work allows to train a single diffusion model for multiple datasets which is useful but mostly for in-distribution scenarios (appendix section D.3 comparing p-diff and this work is doing in-distribution evaluation i.e. evaluate the generated parameters on the same CIFAR-10 and CIFAR-100 datasets which were used to train the diffusion models). Generalization of the proposed dataset conditioning approach to unseen datasets is still limited: as indicated by authors rebuttal response, training diffusion model on CIFAR-10 + CIFAR-100 with 38 pre-trained models gets 35% on STL-10 (unseen dataset) for a seen architecture (resnet-56) which is better than random but far behind train-from-scratch performance (90+%). I believe this gap could be closed by future works in this direction.
>
> [1] Kai Wang, Zhaopan Xu, Yukun Zhou, Zelin Zang, Trevor Darrell, Zhuang Liu, and Yang You. Neural network diffusion, 2024.

---

> ### Author Response · Authors · 2024-11-25
>
> Thank you for your valuable feedback on our response. We appreciate the opportunity to clarify the differences between our method and p-diff, as we believe there may have been some misunderstandings in the comparison.
>
> **Key Differences Between Our Method and p-diff:**
>
> 1. **Scalability and Conditioning:**
>    - *p-diff Limitations:* The p-diff approach was demonstrated using selected batch normalization parameters within larger model, and a full 3-layer small network. For each architecture and dataset, p-diff requires training a new model. In experiments involving numerous models (e.g., 38 models), applying p-diff would necessitate training 38 separate p-diff models to learn the batch normalization parameters, in addition to the original models themselves. This process is not only resource-intensive but also highlights that p-diff was not designed for such scalability.
>    - *Our Approach:* In contrast, our method conditions onthe model architecture or the dataset within a single experimental setting. This allows us to learn the distribution over models and datasets collectively, significantly reducing the need to train multiple separate models. Our approach is thus more scalable and efficient, especially when dealing with a large number of models.
>
> 3. **Unified Learning of Distributions:**
>    - *p-diff's Isolated Training:* Since p-diff requires training separate models for each architecture and dataset, it learns distributions in isolation, without leveraging shared information across different configurations.
>    - *Our Integrated Learning:* By combining models and datasets in a single experimental framework, our method learns the underlying distributions more effectively. This integrated approach captures the relationships between different architectures and datasets, leading to better generalization and performance.
>
> **Implications of Our Method:**
>
> - **Efficiency:** Our method reduces computational overhead by avoiding the need to train multiple models separately for each new architecture or dataset.
> - **Practicality:** The ability to handle larger architectures and condition on them makes our method more applicable to real-world scenarios where models are often complex and varied.
> - **Contribution to the Field:** By addressing the limitations of p-diff and introducing a scalable, conditioned approach, we believe our work offers a meaningful advancement in the field.
>
> One reason we did not extensively benchmark p-diff is that it was listed as "under review" on arXiv, suggesting that the content might not be final or definitive. Therefore, we performed only a single experiment based on the settings provided by the authors and reported these results in the appendix Table 10.
>
> **Possible Misunderstanding of Our Experimental Setting**
>
> In Section 3.1, we clearly state our goal: *"We aim to learn the distribution of a collection of pretrained models conditioned on their pretrained tasks or datasets, such that at inference we can sample weights that are on par with or superior to the pretrained ones on seen data, or achieve good initialization leading to  better performance in a few optimization steps for unseen tasks."*
>
> In Section 4, we specify that our experiments focus on generating parameters with or without fine-tuning, and all our experiments—except for Section 4.5—follow this setting. We never claim that our method does not require fine-tuning on unseen tasks. We acknowledge the challenge of achieving generalization across any unseen dataset, and our work is a step toward that direction. We believe that such a level of generalization cannot be achieved in a single study.
>
> **Potential Misinterpretation of Results in Transfer Learning to STL-10 and CIFAR-10.1**
>
> We would like to clarify the results presented in our transfer learning experiments on CIFAR-10.1 and STL-10. These experiments were conducted on unseen tasks **without any fine-tuning**, and we compared the performance of models initialized by sampling from the pretrained distribution to those initialized using a single pretrained model from our collection—again, without fine-tuning. Importantly, there was **no random initialization** involved in these experiments.
>
> Our objective was to demonstrate that, at initialization, sampling from the pretrained distribution achieves better initial performance on unseen tasks than selecting a single pretrained model. This highlights the advantage of our method in providing a stronger starting point for transfer learning, even when fine-tuning is not applied.

---

> ### Author Response · Authors · 2024-11-25
>
> **Regarding Fintuning compared to traning from scratch**
>
>    To address this concern, we conducted additional experiments using Our model from Table 3  and applied them to the unseen datasets shown in the  Table below including  CIFAR-10 and other datasets. The results show that with a single epoch, D2NWG outperforms both imageNet pretrained models and random initialization. This highlights our method’s effectiveness and its capability to generalize better than traditional pretrained models for transfer learning as stated in section 3.1
>
> compares the accuracy of three methods—**RandInit**, **D2NWG**, and ImageNet **Pretrained**—one epcoh fine-tuned on  **CIFAR-10**, **STL-10**, **AIRCRAFT-100**, **AIRCRAFT-30**, and **PETS**.
>
> | **Dataset**    | **RandInit** | **D2NWG** | **Pretrained** |
> |----------------|--------------|-----------|----------------|
> | **CIFAR-10**   | 61.66        | 87.47     | 88.25          |
> | **STL-10**     | 10.00        | 80.05     | 33.58          |
> | **AIRCRAFT-100** | 1.00        | 1.43      | 1.24           |
> | **AIRCRAFT-30** | 18.41        | 23.53     | 18.91          |
> | **PETS**       | 2.73         | 32.91     | 6.04           |
>
> We hope this clarifies the significant differences between our method and p-diff and addresses any misunderstandings. We are committed to improving our manuscript and are willing to incorporate any additional suggestions you may have.

---

> > ### Comment · Reviewer_V98o · 2024-11-25
> >
> > > Key Differences Between Our Method and p-diff:
> >
> > Thanks, I understand the differences and also stated in my response above that the proposed conditioning is useful.
> >
> > > Possible Misunderstanding of Our Experimental Setting: In Section 3.1, we clearly state our goal: "....at inference we can sample weights that are on par with or superior to the pretrained ones on seen data, or achieve good initialization leading to better performance in a few optimization steps for unseen tasks."
> >
> > I understand the experimental setup well but section 3.1 does not mention "seen data" at all as quoted in your response above. To exactly quote text from section 3.1, lines 158-165 say:
> >
> > "Our goal is to enable conditional sampling of a set of weights p(W_new | D_new ) for a new dataset or tasks D_new (x, y), such that these weights can achieve good performance *on the new dataset either without further training* or with only a few optimization steps compared to random initialization...[skipping text]...we can generate high-performing weights for a target dataset with minimal or *no optimization.*"
> >
> > > We never claim that our method does not require fine-tuning on unseen tasks.
> >
> > This is false. Some examples below:
> > - The abstract states that "this work enables task-specific parameter generation without requiring additional fine-tuning" (line 26).
> > - line 160-161: "Our goal is to enable conditional sampling of a set of weights p(W_new | D_new ) for a new dataset or tasks D_new (x, y), such that these weights can achieve good performance *on the new dataset either without further training*"
> > - line 20-23: "This allows for automatic generation of weights that generalize well across both seen and *unseen* tasks outperforming state-of-the-art meta-learning methods and pretrained models".
> >
> > I believe the language in above places should be toned down to avoid over-claiming.
> >
> > > Potential Misinterpretation of Results in Transfer Learning to STL-10 and CIFAR-10.1: These experiments were conducted on unseen tasks without any fine-tuning
> >
> > Yes, I understand this setup well and can confirm there is no misinterpretation.

---

> > > ### Author Response · Authors · 2024-11-25
> > >
> > > We would like to apologize for any issues with our previous tone and writing. Thank you for your feedback. As recommended by Reviewers to clearly state our claim, we are working on clearly stating our claims and are revising the entire paper, from the introduction to the appendix. We will also include the most important results discussed here. We hope our discussion ends in a good mood.
> > > Thank you

---

> ### Author Response · Authors · 2024-11-25
>
> Thank you for pointing out these overstatements. We will clarify them appropriately.
>
> **On Addressing This is false. Some examples below::**
>
> - **Abstract (line 26):**
>   *"This work enables task-specific parameter generation without requiring additional fine-tuning."*
> We apologize if this statement was misleading. This claim pertains to the performance enhancement of LLMs discussed in Section 4.6. In that section, we perform layer-conditioned sampling to achieve performance exceeding that of Llama-3.1-8B on the OpenLM leaderboard, without access to training or validation data. Our exploration of the parameter space is based on the dataset in Table 6. While the claim is accurate, we acknowledge that it could be clarified to prevent misunderstanding.
>
> We will revise the abstract to address this point.
>
> - **Lines 160-161:**
>   *"Our goal is to enable conditional sampling of a set of weights \( p(W_{\text{new}} | D_{\text{new}}) \) for a new dataset or task \( D_{\text{new}}(x, y) \), such that these weights can achieve good performance on the new dataset either without further training."*
> We apologize for any confusion caused by this statement. The sentence continues with "or with only a few optimization steps compared to random initialization," indicating that in some cases, fine-tuning is performed if necessary to achieve the desired performance. In few-shot scenarios, we did not perform fine-tuning, similar to in-distribution sampling.
>
> We did not explicitly distinguish between "seen" and "unseen" tasks, opting instead to use "new task" to encompass both in-distribution tasks (such as expert model retrieval) and cases requiring continual training (as in the performance comparison with SANE in Table 4). We will clarify these points in the revised manuscript.
>
> - **Lines 20-23:**
>   *"This allows for automatic generation of weights that generalize well across both seen and unseen tasks, outperforming state-of-the-art meta-learning methods and pretrained models."*
>
> Thank you for bringing this to our attention. In referencing state-of-the-art meta-learning methods, we specifically had MetaDiff in mind, which our method outperforms on unseen test sets. Furthermore, as shown in Figure 2 and through additional experiments on STL10 and CIFAR10.21, as well as one-epoch fine-tuning on datasets like Pets, Aircraft, and CIFAR10, our method demonstrates superior generalization compared to pretrained models. While these claims are accurate, we acknowledge they may have been overgeneralized.
>
> We recognize that we may have overstated certain aspects and appreciate your feedback.
>
> Although we did not fine-tune the sampled weights in some cases, this does not mean fine-tuning is never necessary.  that is the point  we would like to clarify.
>
> Thank you once again for your valuable feedback. We will thoroughly revise the paper, incorporating your suggestions. We apologize for any shortcomings in our writing.

---

> > ### Comment · Reviewer_V98o · 2024-12-02
> >
> > Thanks to authors for applaudable effort during rebuttal -- performing new experiments, clarifying questions and incorporating all the feedback in draft. I'm keeping my score taking into account overall contribution of this work but as it suggests, I'm inclined towards accepting this paper.

---

> > > ### Author Response · Authors · 2024-12-02
> > >
> > > Thank you. We greatly appreciate discussing this with you and hope our conversation leads to a positive outcome.

---

### Author Response · Authors · 2024-11-28

Dear Reviewers,

Thank you for your valuable feedback. We have revised our submission to address your concerns, particularly regarding writing and presentation quality.

Following our discussion with Reviewer V98o, we have improved the abstract to better articulate our claims.

We have also condensed the related work section while maintaining all key discussion points.

The introduction has been rephrased while retaining all original citations. In Section 3, where there was confusion about the dataset encoder,
We renamed Section 3.1 to "Preliminary" and added theoretical motivation to address concerns about insufficient theoretical analysis.

To enhance understanding of our work, we have included additional experiments discussed during the review period.

We particularly look forward to Reviewer 4nQa's feedback on these updates.

We appreciate your continued guidance in refining this paper.

Best regards,
[Authors]

---

### Meta-Review · Area_Chair_cDHz · 2024-12-19

**Metareview:**

This paper presents a diffusion-based approach for generating neural network parameters conditioned on dataset representations. The key claim is that by learning a latent distribution of pretrained weights from multiple datasets, the method can efficiently generate task-specific or domain-specific weights for both seen and unseen tasks. The approach is tested on various domains, including vision and large language models, and is shown to outperform several  baselines.
Weaknesses involve the complexity and cost of training the VAE and diffusion components, the reliance on dataset encoders (which may limit generality), unclear theoretical results. Some reviewers also questioned the practical benefits compared to simpler baselines and the scalability to large model zoos. Overall, the paper proposes a novel method, opens a promising direction , and has an extensive set of experiments thus the meta-reviewer recommends accept

**Additional Comments On Reviewer Discussion:**

During the rebuttal phase, the authors responded thoroughly, adding further experiments on multiple architectures and LLMs, discussing computational overhead, providing theoretical insights, and clarifying the difference from prior work. This responsiveness helped alleviate some concerns, leading some reviewers to increase their scores. Nonetheless, some reviewers remained cautious about the scalability and real-world practicality. Overall the consensus acknowledges the strengths outweigh the remaining weaknesses.

---

### Decision · Program_Chairs · 2025-01-22

Accept (Poster)